# Time-resolved interactome profiling deconvolutes secretory protein quality control dynamics

Madison T Wright [1], Bibek Timalsina[1], Valeria Garcia Lopez [2], Jake N Hermanson [2], Sarah Garcia [1] & Lars Plate [1,2,3] ✉

## Abstract

**Many cellular processes are governed by protein–protein interactions that require tight spatial and temporal regulation. Accordingly, it is necessary to understand the dynamics of these interactions to fully comprehend and elucidate cellular processes and pathological disease states. To map de novo protein–protein interactions with time resolution at an organelle-wide scale, we developed a quantitative mass spectrometry method, time-resolved interactome profiling (TRIP). We apply TRIP to elucidate aberrant protein interaction dynamics that lead to the protein misfolding disease congenital hypothyroidism. We deconvolute altered temporal interactions of the thyroid hormone precursor thyroglobulin with pathways implicated in hypothyroidism pathophysiology, such as Hsp70-/90-assisted folding, disulfide/redox processing, and N-glycosylation. Functional siRNA screening identified VCP and TEX264 as key protein degradation components whose inhibition selectively rescues mutant prohormone secretion. Ultimately, our results provide novel insight into the temporal coordination of protein homeostasis, and our TRIP method should find broad applications in investigating protein-folding diseases and cellular processes.**

**Keywords** Temporal Proteomics; Thyroglobulin; Hypothyroidism; Proteostasis; Bioorthogonal Protein Labeling
**Subject Categories** Biotechnology & Synthetic Biology; Proteomics

## Introduction

Protein–protein interactions drive functional diversity within cells and are often closely connected to the observed phenotypes for cellular processes and disease states (Bludau and Aebersold, 2020). Protein homeostasis (proteostasis) is a critical cellular process that relies on tightly regulated protein interactions. The proteostasis network (PN), consisting of protein-folding chaperones, trafficking, and degradation components, maintains the integrity of the proteome by ensuring the appropriate trafficking and localization

of properly folded proteins while recognizing misfolded, potentially detrimental states and routing them for degradation (Karagöz et al, 2019; Needham et al, 2019; Behnke et al, 2016; Pohl and Dikic, 2019). The concerted action of hundreds of proteostasis factors is referred to as protein quality control (PQC). Perturbations to the PN through genetic, age-related, or environmental factors manifest in several disease states, including amyloidosis, neurodegeneration, cancer, and others (Taldone et al, 2020; McDonald et al, 2022; Wright et al, 2021; Kuo et al, 2021; Marinko et al, 2021).

Identifying and quantifying protein–protein interactions has been critical for comprehending the pathogenesis of these disease states. Approaches include yeast two-hybrid systems, co-immunoprecipitation coupled with western blot analysis, the Luminescence-based Mammalian IntERactome (LUMIER) assay, as well as affinity purification—mass spectrometry (AP-MS) (Taipale et al, 2012, 2014; Piette et al, 2021; Rizzolo et al, 2017; Rizzolo and Houry, 2019; Wright and Plate, 2021). These methods have been powerful for mapping steady-state proteostasis interactions to disease states, yet most lack the ability to measure interaction dynamics over time. Proximity labeling mass spectrometry (BioID and APEX-MS) has had limited use to spatiotemporally resolve protein–protein interactions only following protein maturation, as synchronization of newly synthesized protein populations is challenging to achieve (Lobingier et al, 2017; Perez Verdaguer et al, 2022). Unnatural amino acid incorporation methods, such as biorthogonal non-canonical amino acid tagging (BONCAT), pulsed azidohomoalanine (PALM), or heavy isotope labeled azidohomoalanine (HILAQ) can identify newly synthesized proteins but have not focused on a single endogenously expressed protein or group of proteins in the context of disease (Dieterich et al, 2006; Bagert et al, 2014; Ma et al, 2018; McClatchy et al, 2015; Ma et al, 2017; Howden et al, 2013; van Bergen et al, 2022).

Similar DNA and RNA labeling approaches are used to temporally resolve nascent DNA–protein and RNA–protein interactions in cell culture and whole organisms. Isolation of proteins on nascent DNA (iPOND) has revealed the timing of DNA–protein interactions during replication and chromatin assembly (Cortez, 2017; Sirbu et al, 2011; Munden et al, 2022). Thiouracil cross-linking mass spectrometry (TUX-MS) and viral cross-linking and solid-phase purification (VIR-CLASP) are used to study the timing of RNA–protein interactions during viral infection (Kim et al, 2020; Phillips et al, 2016). In contrast, no methods were previously

[1]Department of Chemistry, Vanderbilt University, Nashville, TN 37240, USA. [2]Department of Biological Sciences, Vanderbilt University, Nashville, TN 37240, USA. [3]Department of Pathology, Microbiology and Immunology, Vanderbilt University Medical Center, Nashville, TN 37232, USA. ✉E-mail: lars.plate@vanderbilt.edu

available to identify de novo protein–protein interactions of newly synthesized proteins in a client-specific manner with temporal resolution, which has motivated our efforts here.

In earlier work, we mapped the interactome of the secreted thyroid prohormone thyroglobulin (Tg), comparing the WT protein to secretion-defective mutations implicated in congenital hypothyroidism (CH) (Wright et al, 2021). Tg is a heavily post-translationally modified 330 kDa prohormone that is necessary to produce triiodothyronine (T3) and thyroxine (T4) thyroid-specific hormones (Citterio et al, 2019; Coscia et al, 2020). Tg biogenesis relies extensively on distinct interactions with the PN to facilitate folding and eventual secretion. Our previous results identified topological changes in Tg–PN interactions among CH-associated Tg mutants compared to WT. Nonetheless, the lack of temporal resolution precludes more mechanistic discernment of these changes in PN interactions. While some changes may simply correlate with disease pathogenesis, others may be directly responsible for the aberrant PQC and secretion defect of the mutant Tg variants.

To address this shortcoming, we developed time-resolved interactome profiling (TRIP) to capture and quantify interactions between Tg and interacting partners throughout the life cycle of the protein. We found that Tg mutants are characterized by both discrete changes with select PN components and broad temporal alterations across Hsp70/90, N-glycosylation, and disulfide/redox-processing pathways. Moreover, we find that these perturbations are correlated with alterations in interactions with degradation components. We coupled our TRIP method with functional siRNA screening and uncovered that VCP (p97) and TEX264 are two key regulators of Tg processing. VCP and TEX264 inhibition or silencing in thyroid cells rescued the secretion of mutant Tg, representing—to our knowledge—the first restorative approach based upon proteostasis modulation to increase mutant Tg secretion.

## Results

### TRIP temporally resolves Tg interactions with PQC components

To develop the time-resolved interactome profiling method, we envisioned a two-stage enrichment strategy utilizing epitope-tagged immunoprecipitation coupled with pulsed biorthogonal unnatural amino acid labeling and functionalization (Fig. 1A). Cells are pulse-labeled with homopropargylglycine (Hpg) to synchronize newly synthesized protein populations. After the Hpg pulse, samples are collected across timepoints throughout a chase period (Fig. 1A, Box 1) (Kiick et al, 2001; Beatty et al, 2006). The Hpg alkyne incorporated into the newly synthesized population of protein is then conjugated to biotin using copper-catalyzed azide-alkyne cycloaddition (CuAAC) (Fig. 1A, Box 2). Subsequently, the first stage of the enrichment strategy globally captures the client protein and binding partners using epitope-tagged immunoprecipitation, followed by elution (Fig. 1A, Box 3). The second enrichment step then utilizes a biotin–streptavidin pulldown to capture the Hpg pulse-labeled, and CuAAC-conjugated population, enriching samples into time-resolved fractions (Fig. 1A, Box 4) (Li et al, 2020; Thompson et al, 2019).

Thyroglobulin was chosen as the model secretory client protein. We generated isogenic Fischer rat thyroid (FRT) cells that stably expressed FLAG-tagged Tg (Tg-FT), including WT or mutant variants (A2234D and C1264R) (Appendix Fig. S1). WT Tg is readily secreted from the FRT cells, while C1264R Tg shows only minimal residual secretion (~2% of WT when detected after immunoprecipitation) (Appendix Fig. S1D). A2234D is fully secretion deficient, consistent with the prior characterization of these variants (Hishinuma et al, 1999; Pardo et al, 2009). We first set out to determine if Tg could tolerate immunoprecipitation after pulse Hpg labeling, dithiobis(succinimidyl propionate) (DSP) cross-linking, and CuAAC conjugation with a trifunctional tetramethylrhodamine (TAMRA)-Azide-Polyethylene Glycol (PEG)-Desthiobiotin probe. We showed previously that a C-terminal FLAG-tag is tolerated by Tg and allows efficient immunoprecipitation, while the DSP crosslinker aids in capturing transient protein–protein interactions that take place during Tg processing (Wright et al, 2021). We pulse-labeled FRT cells expressing WT or C1264R Tg for 4 h with Hpg, performed DSP cross-linking, CuAAC, and tested our two-stage enrichment strategy via western blot analysis (Fig. 1B–D). Pulsed Hpg labeling, DSP cross-linking, and CuAAC did not significantly affect immunoprecipitation efficiency and allowed for robust two-stage enrichment of WT and C1264R Tg-FT with well-validated Tg interactors HSPA5 (BiP), HSP90B1 (Grp94), and PDIA4 (ERp72) (Fig. 1B,C) (Menon et al, 2007; Baryshev et al, 2004; Wright et al, 2021). Furthermore, the C-terminal FLAG-tag and Hpg labeling are necessary for this two-stage enrichment strategy, and DSP cross-linking is necessary to capture these interactions after stringent wash steps (Fig. 1D; Appendix Fig. S2).

Next, we investigated whether TRIP could temporally resolve interactions with these PN components. We pulse-labeled WT Tg-FRT cells with Hpg for 1 h, followed by a 3 h chase in regular media capturing timepoints in 30-min intervals and analyzing via western blot or TMTpro LC-MS/MS (Fig. 2A). Our previous study indicated that ~70% of WT Tg-FT was secreted after 4 h, while ~30% of A2234D and 15% of C1264R was degraded after the same time period (Wright et al, 2021). Therefore, we reasoned that a 3-h chase period would be enough time to capture the majority of Tg interactions throughout processing, secretion, cellular retention, and degradation, while still being able to capture an appreciable amount of sample for analysis. For WT-Tg, interactions with HSPA5 peaked within the first 30 min of the chase period and rapidly declined, in line with previous observations, but PDIA4 interactions were not detectable by western blot analysis (Fig. 2B) (Menon et al, 2007; Kim and Arvan, 1995). To test the ability of TRIP to distinguish temporal differences in PQC interactions across mutant Tg variants, we performed TRIP on FRT cells expressing C1264R Tg, a known patient mutation implicated in CH (Hishinuma et al, 1999; Kanou et al, 2007). For C1264R, interactions with HSPA5 were highly abundant at the 0 h timepoint and remained mostly steady throughout the first 1.5 h (Fig. 2C). A similar temporal profile was also observed for HSP90B1. In addition, interactions with PDIA4 were detectable for C1264R and were found to gradually increase throughout the first 1.5 h of the chase period, before rapidly declining (Fig. 2C). We noticed similar temporal profiles for PDIA4 and HSPA5 to our western blot analysis, when measured via TMTpro LC-MS/MS as further outlined below (Fig. 2D,E). In particular, the HSPA5 WT-Tg

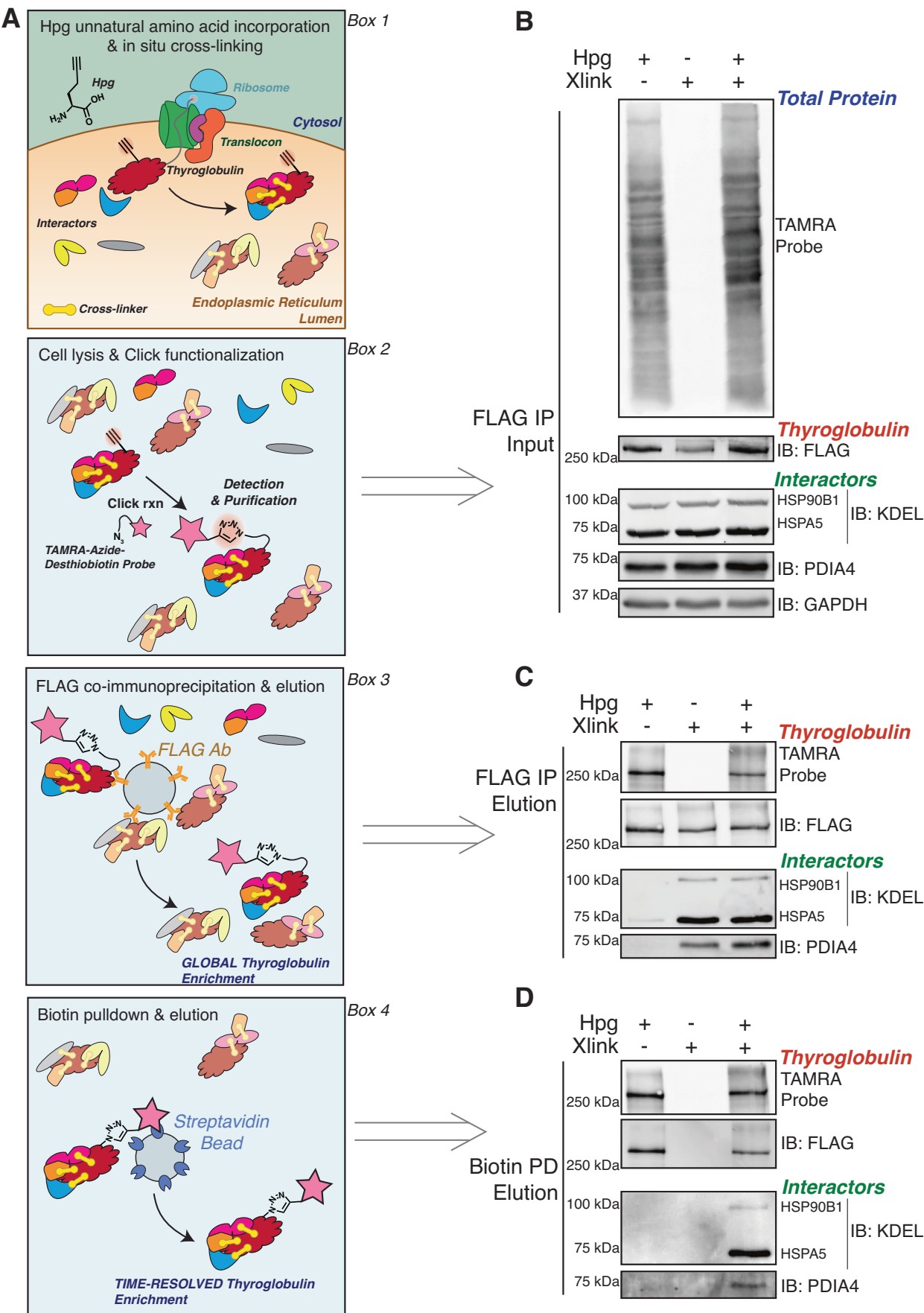

**Figure 1.   Time-resolved interactome profiling to identify interactions with newly synthesized proteins.**

(A) Key steps necessary for the TRIP workflow. Box 1 shows pulsed unnatural amino acid labeling with Hpg to incorporate an alkyne functional group into newly synthesized proteins followed by in situ cross-linking of protein interactions using DSP. Box 2 shows labeled protein functionalization with copper-catalyzed azide-alkyne cycloaddition (CuAAC) click chemistry. Box 3 shows FLAG co-immunoprecipitation to globally enrich labeled and unlabeled Tg. Box 4 shows subsequent streptavidin-biotin enrichment of the pulse-labeled, time-resolved Tg fractions. (B–D) Western blot analysis of the two-stage purification strategy with continuous Hpg labeling. FRT cells stably expressing WT Tg were continuously labeled with Hpg (200 µM) for 4 h and cross-linked with DSP (0.5 mM) for 10 min to capture transient proteostasis network interactions ( + Xlink). Lysates were functionalized with a TAMRA-Azide-PEG-Desthiobiotin probe using CuAAC and subjected to the dual affinity purification scheme as described. (B) FLAG IP inputs as described in Box 2 in (A). Top shows a fluorescence image of TAMRA-labeled proteins, and the bottom shows immunoblots of Tg (IB: FLAG) and interactors HSP90B1, HSPA5, PDIA4, and loading control GAPDH. (C) FLAG IP elutions as described in Box 3 in (A). The top shows a fluorescence image of TAMRA-labeled Tg and total Tg (IB: FLAG). The bottom shows immunoblots of interactors (HSP90B1, HSPA5, and PDIA4). Samples were then subjected to biotin pulldown. (D) Biotin pulldown elutions as described in Box 4 in (A). The top shows a fluorescence image of TAMRA-labeled Tg and total Tg (IB: FLAG). The bottom shows immunoblots of interactors (HSP90B1, HSPA5, and PDIA4). Source data are available online for this figure.

interaction declined within the first hours, yet for C1264R Tg, the HSPA5 interactions remained mostly steady over the 3-h chase period (Fig. 2E).

These data highlight the utility of TRIP to not only identify changes in protein interactions over time but also monitor how these interactions differ for a given protein of interest across disease states. Moreover, these data corroborate our previous findings that steady-state interactions with HSPA5, HSP90B1, and PDIA4 are elevated for C1264R Tg (Wright et al, 2021).

## TRIP identifies altered temporal dynamics associated with Tg processing

We benchmarked the utility of our TRIP approach to temporally resolve previously identified and novel interactors, as the Tg interactome has not been fully characterized in native tissue. We focused on A2234D and C1264R Tg as they present with distinct defects in Tg processing, and mutations are localized in separate structural domains (Kanou et al, 2007; Hishinuma et al, 1999). Following the Hpg pulse-chase labeling scheme and dual affinity purification, the time-resolved Tg fractions were trypsin/Lys-C digested and labeled individually with isobaric TMTpro tags. Subsequently, two sets of TRIP time course samples (0, 0.5, 1, 1.5, 2, and 3 h) could be pooled using the 16plex TMTpro and analyzed by LC-MS/MS (Fig. 2A). In total, five biological replicates were analyzed for WT and six biological replicates were analyzed for A2234D and C1264R (Dataset EV8).

Aside from the experimental samples, we utilized a (−) biotin pulldown booster (or carrier) channel with cells that were pulse-labeled with Hpg, underwent CuAAC functionalization, and immunoprecipitation, but did not undergo biotin–streptavidin enrichment (Fig. 2A). This booster sample acted to (1) aid in increased peptide/protein identification—compared to the much lower abundant chase samples; and (2) benchmark Tg interactors in FRT cells compared to our previously published dataset (Tsai et al, 2020; Petelski et al, 2021). When comparing the booster channel to our (−)Hpg negative control samples, most previously identified Tg interactors were strongly and significantly enriched (Appendix Fig. S3). Our dataset in this study showed appreciable overlap between our previous results in HEK293T cells identifying 75 of the previous 171 Tg interactors and identifying 198 new interactors (Fig. 3A). Several ribosomal and proteasomal subunits, trafficking factors, and lysosomal components were not identified in our previous dataset (Appendix Fig. S3; Dataset EV2). We then took our list of previously identified interactors and PQC

components found to be enriched in the (−) biotin pulldown samples compared to (−) Hpg and carried these proteins forward to time-resolved analysis utilizing the Hpg-chase samples (Dataset EV6).

To map the time-resolved Tg–PN interactome changes, we considered the Hpg-chase samples (0–3 h) and compared the enrichment of Tg interactors to the (−) Hpg control. The enrichments were normalized to Tg protein levels to account for gradual changes due to Tg secretion or degradation, and positive enrichment values were scaled from 0 to 1. We organized the interactors according to distinct PN pathways known to influence Tg processing (Appendix Fig. S4; Datasets EV3 and EV4).

To benchmark the TRIP methodology, we chose to monitor a set of well-validated Tg interactors and compare the time-resolved PN interactome changes to our previously published steady-state interactomics dataset (Wright et al, 2021). Previously, we found that CALR (Calreticulin), CANX (Calnexin), ERP29 (PDIA9), ERP44, and P4HB (PDIA1) interactions with mutants A2234D or C1264R Tg exhibited little to no change when compared to WT under steady-state conditions (Fig. EV1A). However, in our TRIP dataset we were able to uncover distinct temporal changes in engagement that were previously masked within the steady-state data. Our time-resolved data deconvolutes these aggregate measurements, revealing prolonged CALR, ERP29, and P4HB engagements for both A2234D and C1264R Tg mutants compared to WT (Fig. EV4B–F). We found that these measurements for key interactors and PN pathways exhibited robust reproducibility, as exemplified by the standard error of the mean for the TRIP data (Fig. EV1B–I; Appendix Fig. S4B).

Next, we monitored temporal changes more broadly across proteostasis network pathways. We found that both A2234D and C1264R exhibited prolonged interactions with components of Hsp70/90 and disulfide/redox-processing pathways (Fig. 3B–E). Particularly, A2234D and C1264R showed increased interactions with HSPA5, HSP90B1, HYOU1 (Grp170), DNAJC3 (ERdj6), and SDF2L1 throughout the chase period (Fig. 3D). Conversely, WT interactions peaked at the 0 h timepoint and consistently tapered off for many of these components (Fig. 3B,D). Similarly, divergent temporal interactions were observed in the case of disulfide/redox-processing components. PDIA3 (ERp57), PDIA4 (ERp72), PDIA6 (ERp5), and ERP29 have all been heavily implicated in Tg processing (Fig. 3E) (di Jeso et al, 2005; Menon et al, 2007; di Jeso et al, 2014; Baryshev et al, 2006). While WT interactions with these components showed similar trends as Hsp70/90 chaperoning components— peaking at the 0 h timepoint and consistently

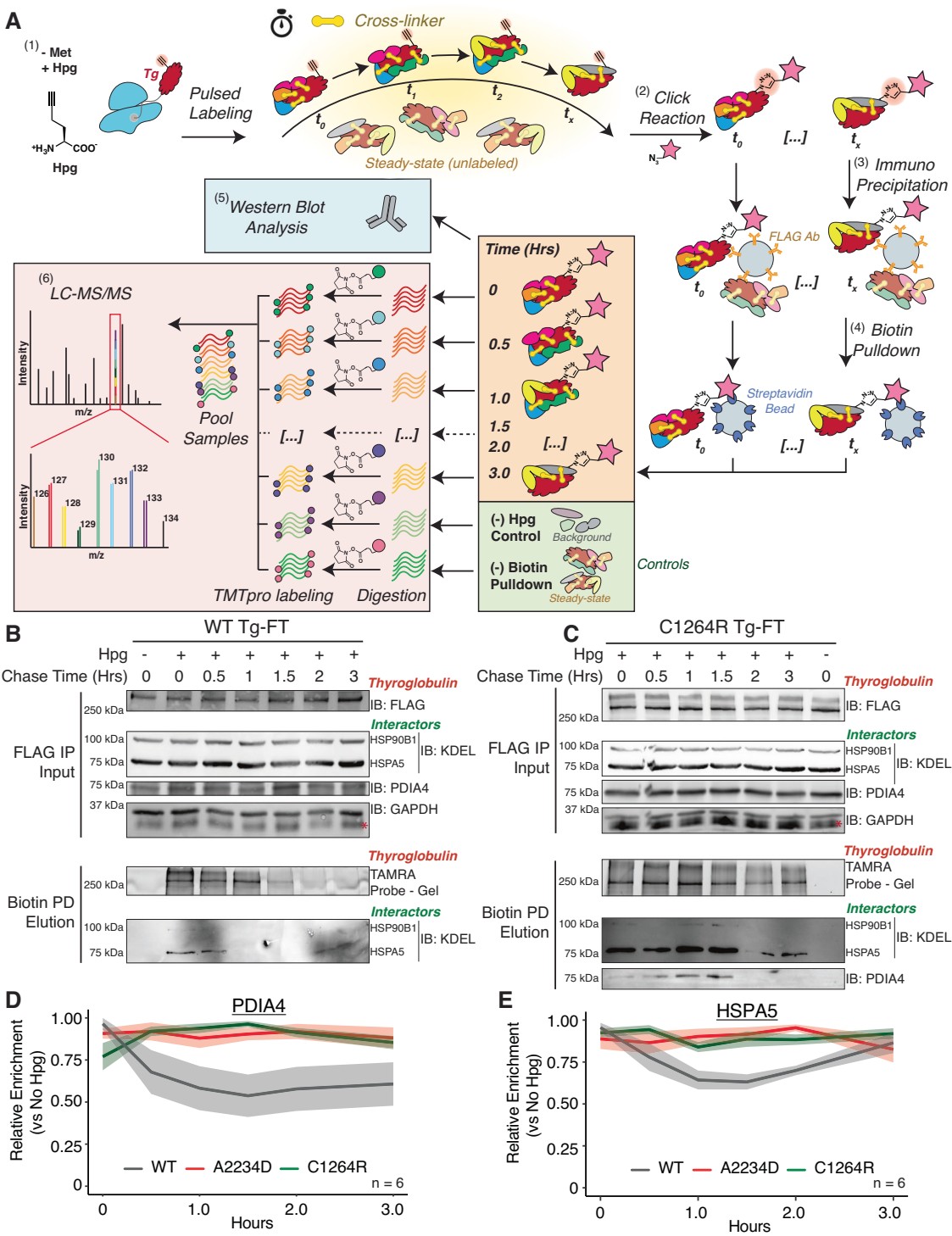

**Figure legend area (panels A–E)**

decreasing—prolonged mutant interactions peaked later at 1–1.5 h for both A2234D and C1264R (Fig. 3C,E). Furthermore, individual protein disulfide isomerase interactions peaked at distinct times, thereby revealing an order to their engagement (Fig. EV1G–I).

To assess temporal interaction changes in an unbiased fashion and identify protein groups exhibiting comparative behavior, we carried out k-means clustering of the temporal profiles for WT and C1264R. This analysis revealed a large divergence in the interaction

profiles. For WT Tg, only one cluster exhibited steadily decreasing interactions (cluster 4), while others increased with time, or showed peaks at intermediate timepoints (Figs. 3F and EV2A). On the other hand, C1264R largely exhibited clusters with decreasing interactions over time (Figs. 3F and EV2B). Cluster 2 for WT with bimodal interactions at early and late timepoints contains many Hsp70/90 chaperoning components. For C1264R Tg, many Hsp70/90 chaperoning components and disulfide/redox-processing

**Figure 2.  TRIP temporally resolves Tg interactions with protein quality control components.**

(A) Workflow for TRIP protocol utilizing western blot or mass spectrometric analysis of time-resolved interactomes. (1) Cells are pulse-labeled with Hpg (200 μM final concentration) for 1 h, chased in regular media for specified timepoints, and cross-linked with DSP (0.5 mM) for 10 min to capture transient proteoastasis network interactions; (2) lysates are functionalized with a TAMRA-Azide-PEG-Desthiobiotin probe using copper CuAAC Click reaction; (3) lysates undergo the first stage of the enrichment strategy where Tg-FT is globally captured and enriched using immunoprecipitation; (4) eluted Tg-FT populations from the global immunoprecipitation undergo biotin–streptavidin pulldown to capture the pulse Hpg-labeled, and CuAAC-conjugated population of Tg-FT, enriching samples into time-resolved fractions; (5) time-resolved fraction may then undergo western blot analysis or (6) quantitative liquid chromatography–tandem mass spectrometry (LC-MS/MS) analysis with tandem mass tag (TMTpro) multiplexing for analysis. The (−) Hpg control channel is used to identify enriched interactors and a (−) Biotin pulldown channel to act as a booster (or carrier). (B, C) TRIP western blot analysis of WT Tg-FT (B) and C1264R Tg-FT (C). Samples were processed as described above in (A). FLAG IP Input panel shows immunoblot of Tg (IB: FLAG), and interactors HSP90B1, HSPA5, PDIA4 and loading control GAPDH. Biotin pulldown elution panel shows a fluorescence image of TAMRA-labeled Tg and immunoblot of Tg interactors HSP90B1 and HSPA5. Validated Tg interactors show higher and delayed enrichment with the misfolded C1264R Tg mutant in (C) compared to WT in (B). PDIA4 interactions were not detectable by western blot analysis for biotin pulldown elution with WT Tg. (D, E) Plots showing the scaled enrichment of select Tg interactors HSPA5 (E) and PDIA4 (D) compared across constructs. Samples were processed as described above in (A) and analyzed by mass spectrometry. The solid line corresponds to mean and shading represents the SEM ($N = 5$ for WT Tg; $N = 6$ for A2234D and C1264R Tg). Data in Dataset EV4. Source data are available online for this figure.

components are instead part of cluster 2', which exhibited an initial rise in interactions strength before plateauing (Figs. 3F and EV2A,B). In addition, we carried out k-means clustering of the combined WT and C1264R time series, which revealed similar clusters (Fig. EV2C). This divergent temporal engagement between WT Tg and the destabilized C1264R mutant is aligned with the patterns observed in the manual grouping (Fig. 3B,C), highlighting that the unbiased temporal clustering can reveal broader patterns in the reorganization of the proteostasis dynamics.

## TRIP highlights link between glycan processing and ER-phagy pathways

One area of particular interest was the crosstalk and correlation between interactions with glycan-processing components and degradation pathways. The link between the glycosylation state of ER clients and ER-associated degradation (ERAD) is well established, whereas more recently defined autophagy at the ER (ER-phagy) or ER to lysosome-associated degradation (ERLAD) represents alternative degradation mechanisms for ER clients (Christianson et al, 2008, 2012; Fregno et al, 2021; Chiritoiu et al, 2020; Fregno et al, 2018). Previously, we showed that A2234D and C1264R differ in interactions with N-glycosylation components, particularly the oligosaccharyltransferase (OST) complex. Efficient A2234D degradation required both STT3A and STT3B isoforms of the OST, which mediate co-translational or post-translational N-glycosylation, respectively (Kelleher et al, 2003; Cherepanova and Gilmore, 2016). TRIP revealed differential interactions with glycosylation components that may lead to altered degradation dynamics. Many glycan-processing enzymes, lectin chaperones, and several subunits of the OST complex were identified (Fig. 3G; Appendix Fig. S3A). While STT3A interactions across all constructs showed similar temporal profiles, we observed prolonged interactions for lectin chaperones CALR and CANX with mutant Tg (Fig. EV1B,C). The most striking difference observed was with the "gatekeeping" glycosyltransferase UGGT1 (Fig. 3G). This protein regulates glycoprotein folding through the CANX/CALR cycle by re-glycosylating ER clients, and thus triggering reengagement with the lectin chaperone cycle (Lamriben et al, 2016). UGGT1 interactions with WT remain moderate from 0.5 to 1.5 h and peak at 3 h, while interactions for A2234D and C1264R peaked earlier and were more pronounced throughout the chase period. These differential interactions with UGGT1 may suggest

changes in the monitoring of the Tg folded state. Moreover, CANX has been directly linked to emerging mechanisms of ERLAD for glycosylated ER clients (Forrester et al, 2019; Fregno et al, 2018).

Proteasomal and autophagic degradation pathways exhibited broad differences in interaction dynamics for WT and mutant Tg (Fig. 3H–J). The ERAD-associated lectin OS9 (Erlec2), and ATPase VCP both peaked at the 3 h chase timepoint for WT Tg (Fig. 3H,K). Conversely, A2234D exhibited more prolonged VCP interactions, and OS9 interactions peaked at the 2 h timepoint (Fig. 3H,K). For C1264R, we observed much stronger and prolonged VCP interactions, as well as additional interactions with the ERAD-associated mannosidase EDEM3 and E3 ubiquitin ligase adaptor SEL1L (Hrd3) (Christianson et al, 2008, 2012) (Fig. 3H). Most notably, our previous aggregate steady-state data showed no significant difference for VCP interactions between WT and mutant Tg, yet our TRIP workflow was able to resolve these temporal dynamics (Fig. 3K) (Wright et al, 2021), further highlighting the utility of this novel TRIP methodology.

Our original dataset identified several lysosomal components, yet it was unclear how Tg might be delivered to the lysosome. Recent work has identified several selective ER-phagy receptors, and highlighted ER-phagy mechanisms for the clearance of mutant prohormones and other destabilized clients from the ER (Chen et al, 2021; Cunningham et al, 2019). It was intriguing then to identify several lysosomal and autophagy-related components and observe differential temporal profiles across WT and C1264R Tg constructs (Fig. 3I; Appendix Fig. S4). For A2234D, interactions with these components were more sparse. The most intriguing observation was the enrichment of three different ER-phagy receptors, ATL3 (Atlastin-3), CCPG1 (Cpr8), and RTN3 (Reticulon 3) between WT and C1264R, along with the RTN3 adaptor protein PGRMC1 (Fig. 3I) (Chen et al, 2019; Liang et al, 2018; Smith et al, 2018; Grumati et al, 2017; Chen et al, 2021). CCPG1 and RTN3 were found to specifically interact with C1264R, with RTN3 interactions peaking at 0 h and then decreasing, while CCPG1 interactions peaked later (Fig. 3I). In the C1264R k-means clustered profiles, autophagy interactions largely group together in the same cluster, showing stronger interactions at earlier timepoints. In the same cluster are glycosylation components (UGGT1, STT3B, and MLEC), further supporting a possible coordination for C1264R Tg between lectin-dependent protein quality control and targeting to autophagy (Fig. EV2B,D).

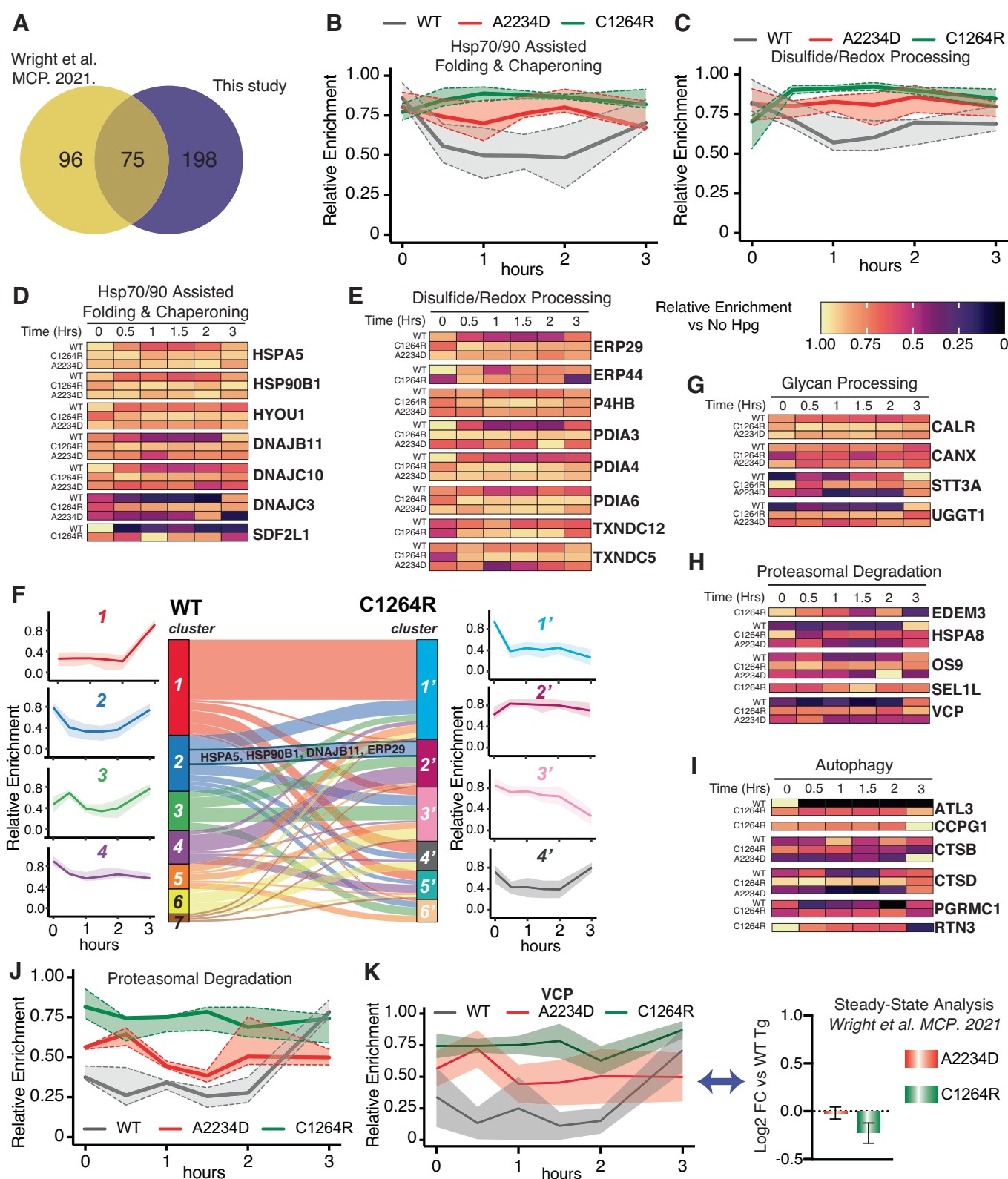

**Figure 3. TRIP identifies altered temporal dynamics associated with Tg processing.**

(A) Venn diagram showing the overlap in proteostasis components identified as Tg interactors here compared to our previous dataset (Wright et al, 2021). (−) Biotin pulldown vs (−) Hpg samples were used to identify Tg interactors in FRT cells. Data available in Dataset EV3. (B, C) Plot showing the relative pathway enrichment for Hsp70-/90-assisted folding & chaperoning interactors (B) and disulfide/redox-processing interactors (C) with WT, A2234D, or C1264R Tg constructs. For this and all subsequent panels, the average log2 fold change enrichment value across timepoints for a given interactor were used to scale data. Positive enrichment was scaled from 0 to 1. Lines represent the median scaled enrichment for the group of interactors and shades correspond to the first and third quartile cutoff. All source data for this and subsequent panels can be found in Dataset EV4. (D, E) Heatmap showing the relative enrichment for individual Hsp70-/90-assisted folding & chaperoning interactors (D) and disulfide/redox-processing interactors (E) with WT, A2234D, or C1264R Tg constructs. Relative enrichment scaled as described above in (C). (F) Unbiased k-means clustering of TRIP profiles to determine co-regulated groups of interactors. Aggregate time profiles for the most prominent clusters are shown on the left (WT) and the right (C1264R). The line corresponds to the mean scaled log2 fold enrichment, and the shading represents the 25–75% quartile range within each cluster. The Sankey plot in the center shows the shift of interactors between clusters from WT to C1264R. (G–I) Heatmap showing the relative enrichment for select glycan processing (G), proteasomal degradation (H) and autophagy interactors (I) with WT, A2234D, or C1264R Tg. (J) Plot showing the relative pathway enrichment for proteasomal degradation interactors. (K) Plot comparing the relative enrichment of VCP throughout the time course for WT, C1264R, and A2234D Tg. The TRIP data (left) is contrasted to aggregate (steady-state) interactomics data (right) (Wright et al, 2021). TRIP resolves dynamic VCP interaction changes with mutant Tg, while these changes are muted in the aggregate data. On the right, data are represented as mean ± SEM ($N = 12$ biological replicates for C1264R, and 6 biological replicates for A2234D). On the left, a solid line corresponds to the mean, and shading represents the SEM ($N = 5$ for WT Tg; $N = 6$ biological replicates for A2234D and C1264R Tg).

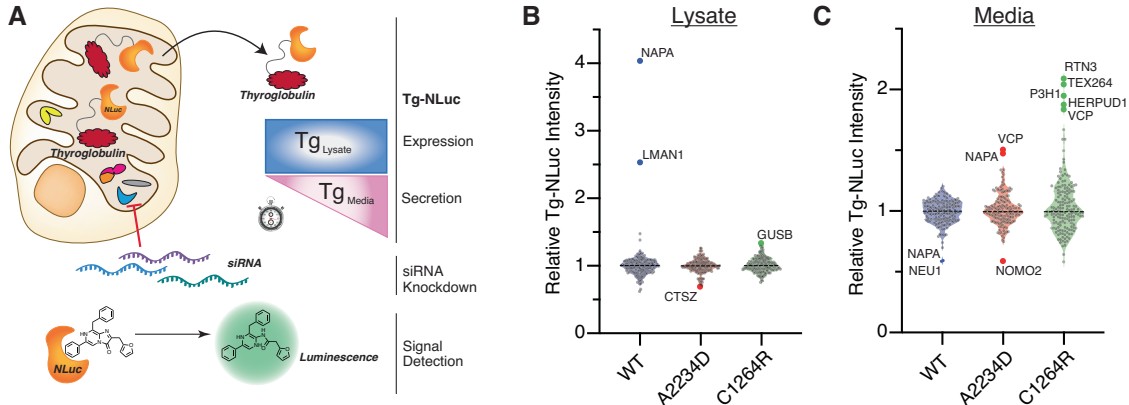

**Figure 4. siRNA screening finds regulators of construct-specific Tg processing.**

(A) siRNA screening workflow utilizing NLuc-tagged Tg to monitor lysate and media abundance. Approximately 36 h after transfection with 25 nM siRNAs cells were replenished with fresh media and Tg-NLuc abundance in lysate and media was measured after 4 h using the nano-glo luciferase assay system. (B) Violin plots showing the relative Tg-NLuc abundance changes in lysate with siRNA knockdown of select genes. Tg-NLuc abundance in lysate was measured 4 h after replenishing with fresh media using the nano-glo luciferase assay system. Hits labeled within the plot were defined as changes in Tg-NLuc abundance by greater than $3\sigma$. $N = 2$ biological replicates for WT and A2234D Tg. $N = 3$ biological replicates for C1264R Tg. (C) Violin plots showing the relative Tg-NLuc abundance changes in media with siRNA knockdown of select genes. Sample processing and cutoff criteria to identify hits are described above in (B). $N = 2$ for WT and A2234D Tg. $N = 3$ for C1264R Tg. Full data for (B, C) is available in Appendix Fig. S6 and Dataset EV5. Source data are available online for this figure.

## siRNA screening discovers key regulators of construct-specific Tg processing

We developed an RNA interference screening platform to investigate whether the temporal interaction changes discovered by TRIP are functionally important for Tg PQC. Moreover, we were interested in identifying factors whose modulation may act to rescue mutant Tg secretion. HEK293 cells were engineered to stably express nanoluciferase-tagged Tg constructs (Tg-NLuc) and screened against 167 Tg interactors and related PN components (see Dataset EV5 for the list of genes). The NLuc tag allowed us to monitor changes to both intracellular Tg abundance in the cell lysates, and Tg levels in the conditioned media to assess secretion rates in a 96-well format (Fig. 4A). Importantly, the NLuc tag did not alter the secretion of WT Tg, and CH-associated mutants maintained the same secretion deficiency (Appendix Fig. S5) (England et al, 2016). Silencing of NAPA (α-SNAP) and LMAN1

(Ergic53) were found to increase WT Tg-NLuc lysate abundance but had no effects on the two mutants (Fig. 4B; Appendix Fig. S6A; Dataset EV5). NAPA is a member of the Soluble N-ethylmaleimide-sensitive factor Attachment Protein (SNAP) family and plays a critical role in vesicle fusion and docking, while LMAN1 is a mannose-specific lectin that functions as a glycoprotein cargo receptor for ER-to-Golgi trafficking (Song et al, 2017; Zhao et al, 2007; Marinko et al, 2021). For mutant Tg-NLuc constructs we found CTSZ (cathepsin Z) silencing decreased A2234D Tg-NLuc lysate abundance, while GUSB (β-glucoronidase) silencing increased C1264R lysate abundance (Fig. 4B).

Remarkably, we identified six genes whose silencing rescued mutant Tg-NLuc secretion in a construct-specific manner. NAPA silencing increased secretion of A2234D Tg-NLuc (Fig. 4C). This contrast to the reduction in WT secretion with NAPA silencing may suggest an alternative role for NAPA in regulating mutant Tg processing as other proteins involved in vesicular fusion and

trafficking have been implicated in ER-phagy (Cui et al, 2019; Liang et al, 2020). Silencing of P3H1 (Lepre1), an ER-resident prolyl hydroxylase, increased C1264R Tg-NLuc secretion but not WT nor A2234D (Fig. 4C) (Vranka et al, 2004). Silencing of several protein degradation genes robustly increased mutant Tg secretion: VCP, HERPUD1, TEX264 and RTN3. VCP silencing increased both A2234D and C1264R Tg-NLuc secretion (Fig. 4C). VCP is associated with ERAD but also aids in several diverse cellular functions including the interplay between proteasomal and autophagic degradation (Hill et al, 2021; Christianson et al, 2008, 2012). VCP silencing exclusively affecting mutant Tg corroborates our TRIP dataset, and suggests a more prominent role for VCP in mutant Tg PQC compared to WT. VCP interactions were sparse for WT Tg, while they remained more steady throughout the chase period for the mutants (Fig. 3H,K). HERPUD1, TEX264 and RTN3 silencing selectively increased C1264R secretion, but did not alter WT nor A2234D secretion (Fig. 4C). HERPUD1 is a ubiquitin-like protein and associates with VCP during ERAD (Christianson et al, 2012; Needham et al, 2019; Okuda-Shimizu and Hendershot, 2007). The ER-phagy receptors RTN3 and TEX264 localize to subdomains of the ER to facilitate degradation of specific ER clients and organellular regions (Chino et al, 2019; Fielden et al, 2022; An et al, 2019; Grumati et al, 2017; Chen et al, 2021; Cunningham et al, 2019). Unfortunately, TEX264 was not identified in our TRIP data, but RTN3 was found to specifically interact with only C1264R (Fig. 3I; Appendix Fig. S4). Of note, while the ER-phagy receptor CCPG1 was identified in our mass spectrometry dataset, siRNA silencing of CCPG1 did not significantly alter Tg-NLuc abundance in lysate or media, nor did silencing of SEC62 or RETREG1 (FAM134B), two additional ER-phagy receptors found to regulate ER dynamics (Appendix Fig. S6) (Fumagalli et al, 2016; Bhaskara et al, 2019).

This is the first study to broadly investigate the functional implications of Tg interactors and other PQC network components on Tg processing. Coupling these data with our TRIP methodology helped to deconvolute PQC dynamics associated with Tg and identify pathways implicated in the aberrant secretion of CH-associated Tg mutations. The discovery of several protein degradation components as hits for rescuing mutant Tg secretion may suggest that the blockage of degradation pathways can broadly rescue the secretion of A2234D and C1264R mutant Tg, a phenomenon similarly found for destabilized CFTR implicated in the protein-folding disease cystic fibrosis (Vij et al, 2006; Pankow et al, 2015; McDonald et al, 2022).

## Trafficking and degradation factors selectively regulate Tg processing in thyroid cells

We examined the hits from the initial siRNA screen in FRT cells stably expressing Tg constructs to test whether their silencing exhibited similar phenotypes in thyroid-specific tissue. Thyroid tissue must synthesize and fold a large amount of Tg as it is the main protein produced and can make up more than 50% of all protein components within the thyroid gland (di Jeso and Arvan, 2016). Silencing of NAPA led to a ~50% increase in WT-Tg lysate abundance, while NAPA and LMAN1 silencing both led to marginal decreases in WT-Tg secretion after 4 h, consistent with the results in HEK293 cells (Figs. 5A,B and EV3A,B). Using $^{35}$S

pulse-chase analysis, we confirmed that NAPA silencing significantly increased lysate retention by 15% over 4 h and decreased secretion by 18% (Fig. EV3H). To understand if these results were directly attributable to NAPA function, we performed complementation experiments where FRT cells treated with NAPA siRNAs were cotransfected with a human NAPA plasmid. WT-Tg lysate abundance decreased when NAPA expression was complemented, confirming that the observed retention phenotype could be attributed to NAPA silencing (Fig. EV3I). These results established that NAPA acts as a pro-secretion factor for WT Tg.

In C1264R Tg-FRT cells, RTN3, TEX264, and HERPUD1 silencing marginally, yet significantly decreased C1264R lysate abundance (Figs. 5C,D and EV3C-G). Moreover, VCP and TEX264 silencing significantly increased C1264R secretion by threefold and twofold, respectively, after 8 h, in line with our results in HEK293 cells (Fig. 5C,D). In contrast, silencing of RTN3 significantly decreased C1264R Tg secretion by fivefold, in opposition to the increased secretion observed in HEK293 cells. Several individual VCP and TEX264 siRNAs were able to partially recapitulate these increased secretion phenotypes on C1264R Tg-FT, confirming that the effect is mediated by the respective gene silencing (Fig. EV3J,K).

## Mutant Tg is selectively enriched with the ER-phagy receptor TEX264

Intrigued by the finding that TEX264 silencing increased C1264R Tg secretion without affecting WT Tg, we asked whether TEX264 exhibited differential interactions with Tg constructs. HEK293 Tg-NLuc cells were transfected with either a fluorescent GFP control, or C-terminal FLAG-tagged ER-phagy receptors, followed by FLAG co-IP. The NLuc assay and western blotting were then used to monitor Tg enrichment (Fig. 6A). The negative control SEC62 did not yield any appreciable enrichment of Tg compared to GFP control (Fig. 6B), consistent with SEC62 not impacting Tg constructs in the siRNA screen. Conversely, co-IP of TEX264 resulted in the enrichment of all Tg variants compared to GFP control when monitored by NLuc assay, with C1264R and A2234D being significantly enriched. C1264R Tg exhibited a threefold increased interaction compared to WT Tg and almost a twofold increase compared to A2234D Tg (Fig. 6B). This increase in C1264R enrichment was also observable by western blot analysis (Fig. 6C). In addition, we monitored Tg enrichment with ER-phagy receptors CCPG1 and RTN3 via Western blot as both were found to be C1264R Tg interactors within our TRIP dataset. RTN3L is found to be the only RTN3 isoform involved in ER turnover via ER-phagy (Grumati et al, 2017). WT and C1264R Tg-NLuc were modestly enriched with RTN3L compared to control samples expressing GFP. Conversely, we found that all Tg variants exhibited modest interactions with CCPG1 compared to control samples expressing GFP, although less than with TEX264 (Appendix Fig. S7).

Together, these data suggest that TEX264, CCPG1, or RTN3L engage with Tg during processing, and CH-associated Tg mutants may be selectively targeted to TEX264. Furthermore, ER-phagy may be considered as a degradative pathway in Tg processing, as other studies have mainly focused on Tg degradation through ERAD (Tokunaga et al, 2000; Menon et al, 2007).

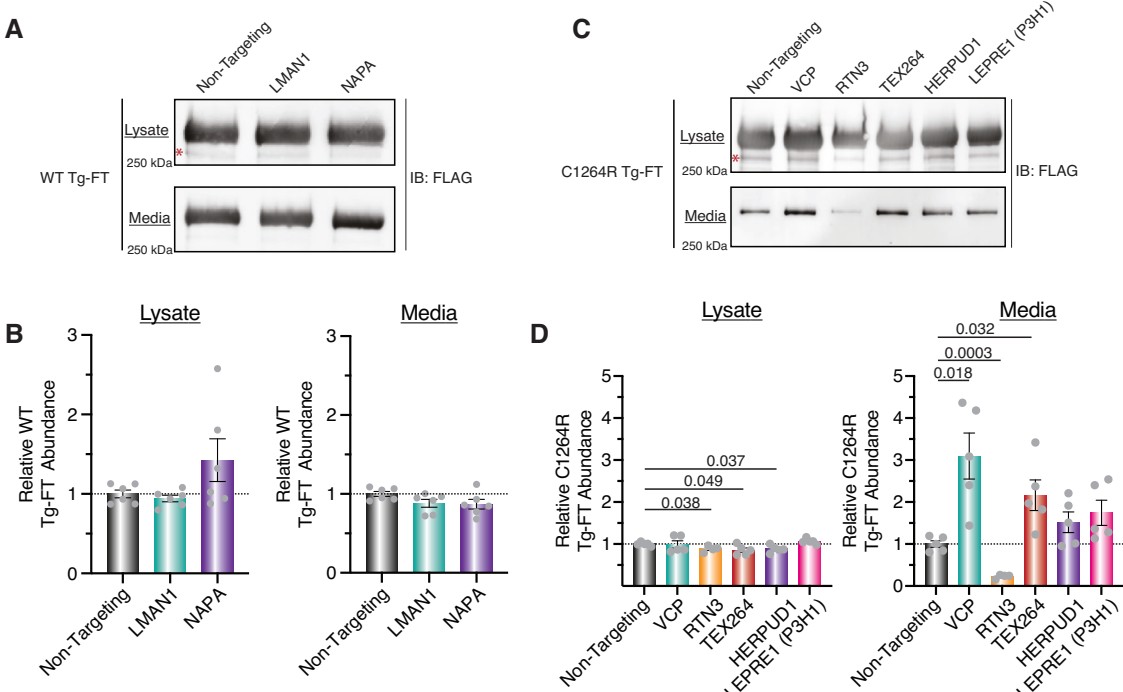

**Figure 5. Trafficking and degradation factors regulate Tg processing in FRT cells.**

(A, B) Western blot analysis (A) and quantification (B) of WT Tg-FT secretion from FRT cells transfected with select siRNAs hits from the initial screening dataset. The red asterisk denotes a non-specific background band within the western blot. Cells were transfected with 25 nM siRNAs for 36 h, media exchanged and conditioned for 4 h, Tg-FT was immunoprecipitated from lysate and media samples, and Tg-FT amounts were analyzed via immunoblotting. Data are represented as mean ± SEM from $N = 6$ biological replicates. (C, D) Western blot analysis (C) and quantification (D) of C1264R Tg-FT secretion from FRT cells transfected with select siRNA hits from the initial screening dataset. Red asterisk denotes a non-specific background band within the western blot. Cells were transfected with 25 nM siRNAs for 36 h, media exchanged and conditioned for 8 h, Tg-FT was immunoprecipitated from lysate and media samples, and Tg-FT amounts were analyzed via immunoblotting. All statistical testing performed using an unpaired Student's $t$ test with Welch's correction with $P$ values as indicated. Data are represented as mean ± SEM from $N = 5$ biological replicates (one RTN3 sample excluded due to sample handling error). Source data are available online for this figure.

## Pharmacological VCP inhibition selectively rescues C1264R Tg secretion

Considering the promising finding that silencing of degradation factors by siRNA rescued C1264R secretion, we sought to investigate whether pharmacological modulation of select Tg processing components could similarly improve Tg secretion. There are no selective inhibitors currently available for TEX264, but several for VCP. We monitored C1264R Tg secretion from FRT cells in the presence of VCP inhibitors ML-240, CB-5083, and NMS-873. ML-240 and CB-5083 are ATP-competitive inhibitors that preferentially target the D2 domain of VCP subunits, whereas NMS-873 is a non-ATP-competitive allosteric inhibitor that binds at the D1-D2 interface of VCP subunits (Chou et al, 2013, 2014; Anderson et al, 2015; le Moigne et al, 2017; Tang et al, 2019). ML-240 and NMS-873 have been shown to decrease both proteasomal degradation and autophagy, in line with VCP playing a role in both processes (Chou et al, 2013, 2014; Her et al, 2016). Conversely, while CB-5083 is known to decrease proteasomal degradation it has been shown to increase autophagy. (Anderson et al, 2015; le Moigne et al, 2017; Tang et al, 2019). We found that treatment with ML-240 was able to significantly increase C1264R Tg secretion without altering C1264R Tg lysate abundance, corroborating the siRNA silencing data (Fig. 7A,B). To investigate whether this rescue

of C1264R Tg secretion with ML-240 treatment was specific to C1264R Tg, we also monitored WT-Tg abundance. In contrast, we observed that ML-240 significantly reduced WT-Tg abundance in both lysate and media 4 and 8 h after treatments (Fig. 7C,D).

We hypothesized that ML-240 treatment may differentially regulate Tg degradation and processing in a construct-specific manner. Hence, we turned to a $^{35}$S pulse-chase assay to fully characterize Tg degradation and processing dynamics and found that lysate abundance of C1264R Tg was not significantly changed at the 4 h timepoint with ML-240 treatment compared to DMSO (Figs. 7E and EV4A,B). This paralleled the previous results with VCP silencing and ML-240 treatment under steady-state conditions in FRT cells. Impressively, C1264R Tg secretion was increased tenfold at the 4 h timepoint with ML-240 treatment compared to DMSO with no significant change in C1264R Tg degradation in the presence of ML-240 (Figs. 7E and EV4A,B). Conversely, for WT-Tg ML-240 treatment led to a significant increase in lysate accumulation at 4 h compared to DMSO (Figs. 7F and EV4C,D). This increase in lysate abundance correlated with a significant decrease in WT-Tg secretion from 67% to 4% in the presence of ML-240 compared to DMSO, without altering WT degradation (Figs. 7F and EV4C,D).

To understand whether this rescue in secretion was uniquely linked to VCP inhibition or could be more broadly attributed to

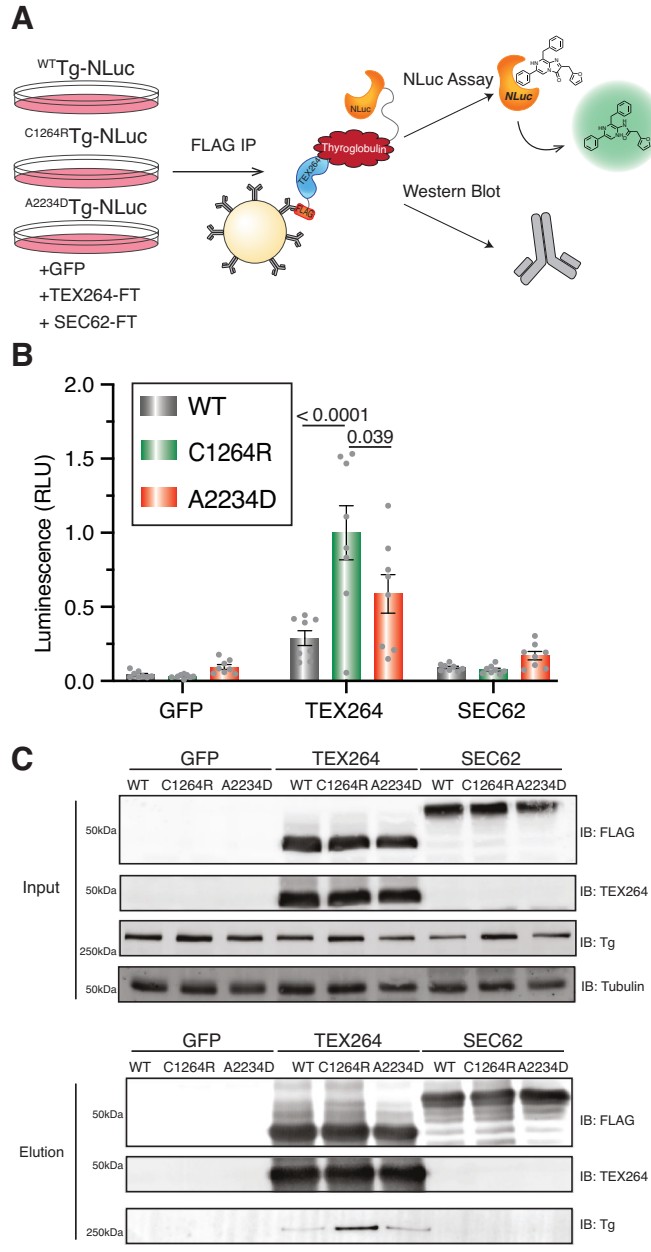

**Figure 6. Mutant Tg is selectively enriched with the ER-phagy receptor TEX264.**

(A) Schematic for western blot and NLuc analysis for identifying TEX264-Tg interactions. HEK293T cells stably expressed WT or mutant Tg-NLuc were transfected with FLAG-tagged ER-phagy receptors TEX264, SEC62, or GFP control. After FLAG Co-IP, samples were analyzed using the using the nano-glo luciferase assay system or immunoblot analysis to monitor Tg enrichment. (B) Luminescence data from FLAG-Co-IP and nano-glo luciferase analysis. Tg is selectively enriched with TEX264 compared to GFP fluorescent control, or SEC62. Mutant Tg exhibits higher enrichment compared to WT. Data represented as mean ± SEM. Statistical testing was performed using a one-way ANOVA with post hoc Tukey's multiple testing corrections with adjusted P values as indicated. N = 8 biological replicates. (C) Western blot analysis of FLAG Co-IP samples. Top panel shows IP inputs and bottom panel IP elutions with Tg (IB: Tg), ER-phagy receptors (IB: FLAG and IB: TEX264) and loading control Tubulin. Tg is selectively enriched with TEX264 compared to GFP fluorescent control, or SEC62, with C1264R Tg exhibiting higher enrichment compared to WT. Source data are available online for this figure.

blocking Tg degradation, we tested the proteasomal inhibitor bortezomib, and lysosomal inhibitor bafilomycin. Bafilomycin increased WT-Tg lysate abundance, and bortezomib significantly increased A2234D lysate abundance, consistent with a role of these degradation processes in Tg PQC (Fig. EV5A). When monitoring Tg-NLuc media abundance, neither bafilomycin nor bortezomib significantly altered WT, A2234D, or C1264R abundance, confirming that general inhibition of proteasomal or lysosomal degradation does not rescue mutant Tg secretion (Fig. EV5B).

Finally, we monitored the activation of the unfolded protein response (UPR) in the presence of ML-240 in FRT cells expressing C1264R Tg-FT. Phosphorylation of eIF2α, an activation marker for the PERK arm of the UPR, was induced within 2 h of ML-240 treatment (Fig. EV5C). We further investigated the induction of UPR targets via qRT-PCR. HSPA5 and ASNS transcripts, markers of ATF6 and PERK UPR activation, respectively, remained unchanged or slightly decreased after 3 h treatment with ML-240 in C1264R Tg cells (Fig. EV5D). Only DNAJB9, a marker of the IRE1 arm of the UPR, showed a significant increase in both WT-Tg and C2164R Tg-FRT cells (Fig. EV5D). Moreover, ML-240 did not significantly alter cell viability after 3 h, as measured by propidium iodide staining (Fig. EV5E). Overall, these results highlight that the short ML-240 treatment induces early UPR markers, but the selective rescue of C1264R Tg secretion via ML-240 treatment is unlikely the results of global remodeling of the ER PN due to UPR activation.

## Secretion rescue of C1264R Tg is associated with temporal remodeling of the interactome

After identifying that VCP inhibition via ML-240 rescued C1264R Tg secretion, we sought to use TRIP to capture PQC interaction changes that correlated with increased secretion. TRIP was carried out in FRT cells expressing C1264R Tg in the presence of ML-240 to monitor temporal changes in PN interactions (Appendix Fig. S8; Dataset EV4). Both glycan-processing and Hsp70-/90-chaperoning pathways exhibited broad decreases in C1264R Tg interactions in ML-240 treated samples compared to untreated samples (Fig. 7G,H). Particularly, CALR, CANX and UGGT1 interactions tapered off more rapidly within 0.5 h compared to untreated C1264R. In contrast, interactions with key Hsp70-/90-chaperoning components remained relatively steady throughout the chase period before peaking at the 3 h timepoint (Fig. 7I,J). Conversely, C1264R interactions with chaperones HSPA5 and HSP90B1, and co-chaperones DNAJB11, DNAJC10, and DNAJC3 decreased and mimicked the WT-Tg temporal profile (Figs. 3D and 7J).

Interactions with disulfide/redox-processing components exhibited milder but marked declines at intermediate timepoints with ML-240 treatment compared to untreated samples (Fig. 7K,L; Appendix Fig. S8). PDIA4 interactions remained much lower before peaking at the 3 h timepoint (Fig. 7L). Conversely, ERP29 and PDIA3 interactions remained largely unchanged in the presence of ML-240 (Fig. 7L). The most striking interaction changes occurred with proteasomal degradation components, which remained steady until 1.5 h, but then abruptly declined with ML-240 treatment at later timepoints (Fig. 7M,N). This decline tracks with changes to the glycan-processing machinery, highlighting that the coordination between N-glycosylation and diverting Tg away from ERAD may be a key to the rescue mechanism.

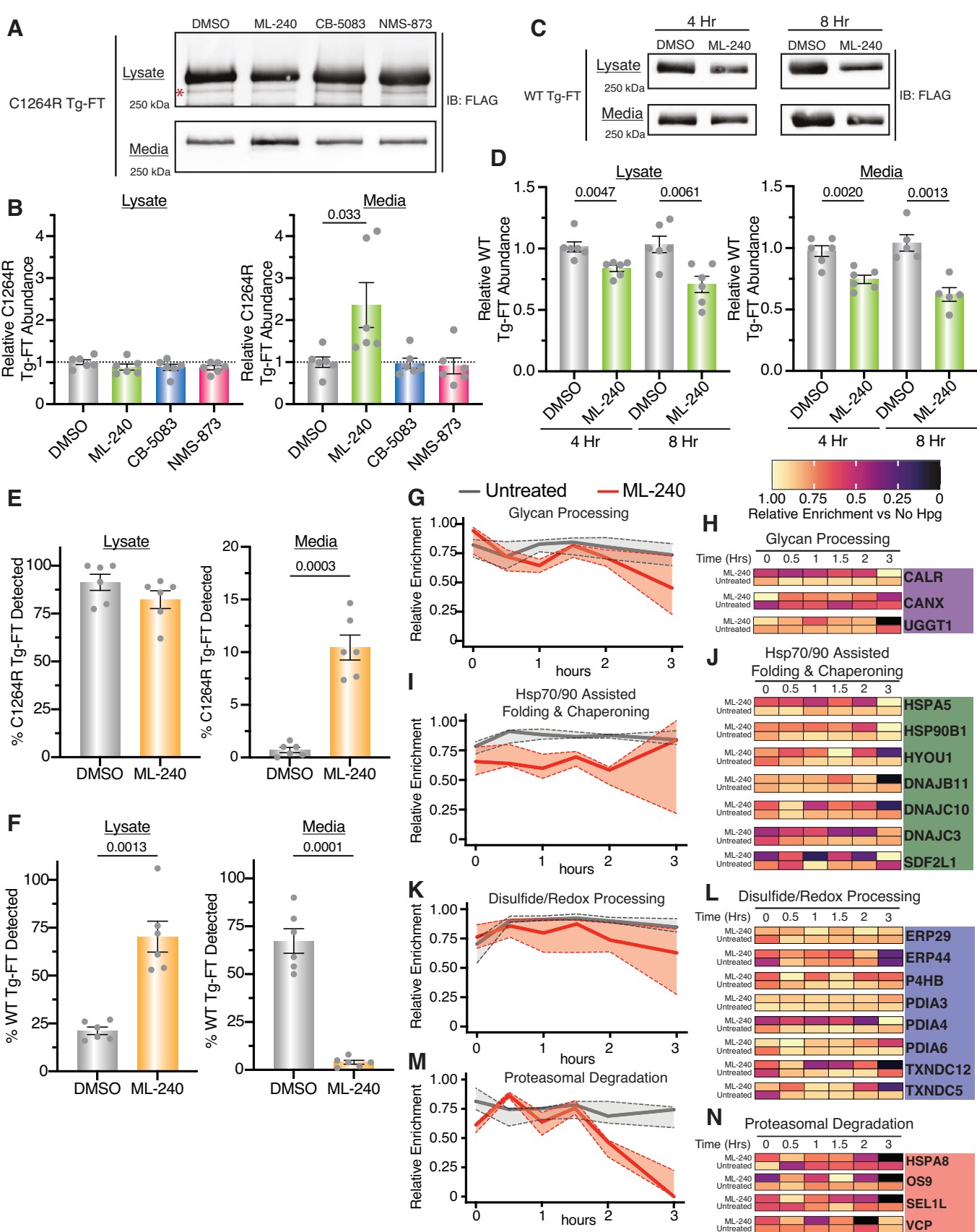

**Figure 7.  Secretion rescue of C1264R Tg is associated with temporal remodeling of the Tg interactome.**

(A, B) Western blots analysis (**A**) and quantification (**B**) of C1264R Tg-FT FRT cells treated with VCP inhibitors ML-240 (10 µM), CB-5083 (5 µM), NMS-873 (10 µM), or vehicle (0.1% DMSO) for 8 h. Lysate and media samples were subjected to immunoprecipitation and analyzed via immunoblotting. Data are normalized to the median C1264R Tg-FT abundance of DMSO-treated samples. Data represented as mean ± SEM. Statistical testing was performed using an unpaired Student's *t* test with Welch's correction with the exact *P* value indicated. $N = 6$ biological replicates. (**C, D**) Western blots analysis (**C**) and quantification (**D**) of WT Tg-FT FRT cells treated with ML-240 for 4 and 8 h. Lysate and media samples were processed and analyzed as described above. Data are normalized to the median WT Tg-FT abundance of DMSO-treated samples. Data representation and statistical testing as in (**B**). Data are represented as mean ± SEM from $N = 6$ biological replicates. (**E**) $^{35}$S Pulse-chase analysis of C1264R Tg-FT FRT cells with ML-240 treatment. Cells were pre-treated with ML-240 or DMSO for 15 min prior to pulse labeling with $^{35}$S for 30 min and chased for 4 h with DMSO or ML-240 treatment. Data are represented as mean ± SEM. Full data panel available in Fig. EV4A,B. Statistical testing was performed using an unpaired Student's *t* test with Welch's correction with the exact *P* value indicated. $N = 6$ biological replicates. (**F**) $^{35}$S Pulse-chase analysis of WT Tg-FT FRT cells with ML-240 (10 µM) treatment. Samples were processed and analyzed as described in (**E**). Data are represented as mean ± SEM. Full data panel available in Fig. EV4C,D. Statistical testing was performed using an unpaired Student's *t* test with Welch's correction with the exact *P* value indicated. $N = 6$ biological replicates. (**G–N**) TRIP analysis showing the relative enrichment for select Tg interactors across proteostasis pathways between untreated and ML-240 (10 µM) treated C1264R Tg. Cells were treated with ML-240 (10 µM) and analyzed utilizing the TRIP workflow (Fig. 2). The average log2 fold change enrichment values across timepoints for a given interactor are used to scale data. Positive enrichment values were scaled from 0 to 1. The enrichment is shown for Glycan Processing (**G, H**), Hsp70/90-Assisted Folding (**I, J**), Disulfide/Redox Processing (**K, L**), and Proteasomal Degradation (**M, N**). Aggregate pathway enrichment is displayed in (**G**), (**I**), (**J**) and (**M**), where lines represent the median scaled enrichment for the group of interactors and shades correspond to the first and third quartile cutoff. Heatmaps with individual enrichments of interactors is shown in (**H**), (**J**), (**L**), and (**N**). Data are available in Dataset EV4. Source data are available online for this figure.

# Discussion

The timing of protein–protein interactions implicated in cellular processes and pathogenic states remains pivotal to our understanding of disease mechanisms. Nonetheless, the temporal measurement of these interactions has remained difficult and has proven to be a bottleneck in elucidating the coordination of complex biological pathways such as the PN. Here, we developed an approach to temporally resolve protein–protein interactions implicated in PQC. Furthermore, we complemented these data with a functional genomic screen to further characterize and investigate the implications of these protein–protein interactions on Tg processing. This combination of our novel TRIP approach coupled with functional screening deconvoluted previously established PQC mechanisms for Tg processing while also providing new paradigms within PQC pathways that are critical for the secretion of the prohormone.

TRIP has allowed the identification and resolution of unique temporal changes in Tg interactions with glycan-processing components including CANX, CALR, and UGGT1, while contrasting WT to mutant Tg variants. These changes subsequently correlate with altered interactions with ERAD components EDEM3, OS9, SEL1L, and VCP. Moreover, we identified a broader scope of Tg interactors in thyroid tissue, including ER-phagy receptors ATL3, CCPG1, RTN3, and the RTN3 adaptor protein PGRMC1. Identification of these receptors establishes a direct link from Tg processing to ER-phagy or ERLAD degradation mechanisms. In addition, glycan-dependent and independent mechanisms have been established for the degradation of ERAD-resistant ER clients (He et al, 2021; Fregno et al, 2021). This overlap in degradation pathways may be similarly at play in the case of Tg biogenesis and processing, especially with transient disulfide-linked aggregation taking place during nascent Tg folding and mutants forming difficult-to-degrade aggregates (Kim et al, 1993; di Jeso et al, 2005; Menon et al, 2007; di Jeso et al, 2014).

An important finding was that TRIP was capable of resolving subtle temporal interaction changes, for example, with CALR, ERP29, ERP44, P4HB, and VCP, which were otherwise masked in steady-state interactomics data (Wright et al, 2021). Nonetheless, there are some limitations within our TRIP methodology. TRIP relies on a two-stage purification strategy which increases sample handling and limits the amount of protein that can be subsequently enriched. Both add inherent variability to the workflow. Furthermore, pulsed labeling with unnatural amino acids has been shown to slow portions of protein translation (Kiick et al, 2001; Dieterich et al, 2006; Bagert et al, 2014; Lang and Chin, 2014). To address this, we utilized a labeling time of 1 h which allows us to generate a large enough labeled population of Tg-FT for TRIP analysis, but some early interactions are likely missed within the TRIP workflow. In the case of mutant Tg, performing the TRIP analysis for much longer chase periods (6–8 h) may provide insightful details to the iterative binding process of PN components that is thought to facilitate protein retention within the secretory pathway. In addition, within our dataset, we noticed that the temporal profiles for ribosomal and proteasomal subunits, trafficking and lysosomal components are inherently difficult to measure across experiments. To efficiently measure these components temporally the TRIP methodology will require further optimization.

The functional implications of protein–protein interactions can be difficult to deduce, especially in the case of PQC mechanisms containing several layers of redundancy across stress response pathways, paralogs, and multiple unique proteins sharing similar functions (Wright and Plate, 2021; Bludau and Aebersold, 2020; Karagöz et al, 2019; Braakman and Hebert, 2013). This led us to establish a siRNA screening platform to complement our TRIP data and broadly investigate the functional implications of PQC components. With this assay, we found the trafficking factor NAPA was heavily implicated in WT-Tg secretion. Most strikingly, we found that VCP and TEX264 were implicated in C1264R processing, and siRNA silencing of either led to rescue of C1264R secretion. Rescue of mutant Tg trafficking from ER to Golgi, but not secretion, with low-temperature correction had been documented previously (Kim et al, 1996). Yet, this is the first study, to our knowledge, that identified a restorative approach to mutant Tg secretion. To expound upon this, our findings that pharmacological VCP inhibition selectively rescues C1264R secretion compared to WT, and mutant Tg was selectively enriched with TEX264 further corroborated the TRIP data establishing that Tg mutants undergo differential interactions with degradation pathways compared to WT Tg. Similar phenomena have been observed for the partitioning

and differential interactions of other prohormone proteins, such as proinsulin and proopiomelanocortin, and proteasome-resistant polymers of alpha1-antitrypsin Z-variant. RTN3 is implicated in these differential interactions and is shown to be selective for these prohormones compared to other ER-phagy receptors (Chen et al, 2021; Cunningham et al, 2019; Fregno et al, 2018). While A2234D and C1264R Tg were preferentially enriched with TEX264 compared to WT, it remains unclear what other accessory proteins may be necessary for the recognition of TEX264 clients (Chino et al, 2019; An et al, 2019). Furthermore, TEX264 function in both protein degradation and DNA damage repair further complicates siRNA-based investigations (Fielden et al, 2022). Further investigation is needed to fully elucidate (1) if Tg degradation takes place via ER-phagy and (2) by which mechanisms this targeting is mediated.

As we discovered that pharmacological VCP inhibition with ML-240 can rescue C1264R Tg secretion yet is detrimental for WT-Tg processing, it is unclear whether VCP may exhibit distinct functions for WT and mutant Tg PQC. Finally, as ML-240 is shown to block both the proteasomal and autophagic functions of VCP it is unclear which of these pathways may be playing a role in the rescue of C1264R, or detrimental WT processing (Chou et al, 2013, 2014).

We used our TRIP method to monitor the changes in interactions associated with the rescue of C1264R Tg with pharmacological VCP inhibition. We found that this rescue is correlated with broad temporal changes in interactions across glycan processing, Hsp70-/90-chaperoning and proteasomal degradation pathways, while exhibiting more discrete changes with select disulfide/redox-processing components. Mapping these temporal changes in response to pharmacological VCP inhibition and C1264R rescue highlights the capabilities of TRIP to not only resolve protein–protein interactions across disease states but also identify compensatory mechanisms that may take place with drug treatment or other modulating techniques like gene inhibition or activation used to study or treat disease states. Consequently, TRIP should find broad applicability for delineating the proteostasis deficiencies that give rise to diverse protein misfolding diseases and elucidating other cellular interactome dynamics.

# Methods

**Reagents and tools table**

| Reagent/resource | Reference or source | Identifier or catalog number |
| --- | --- | --- |
| **Antibodies** | | |
| KDEL | Enzo Life Sciences | ADI-SPA-827 |
| M2-FLAG | Sigma-Aldrich | F1804 |
| PDIA4 | Protein Tech | 14712-1-AP |
| TG | Proteintech | 21714-1-AP |
| GAPDH | GeneTex | GTX627408 |
| Tubulin-Rhodamine | Bio-Rad | 12004165 |
| Goat anti-mouse Star-bright700 | Bio-Rad | 12004158 |
| Goat anti-rabbit IRDye800 | LI-COR | 926-32211 |

| Reagent/resource | Reference or source | Identifier or catalog number |
| --- | --- | --- |
| G1 Anti-DYKDDDDK affinity resin | GenScript | L00432 |
| High-Capacity Streptavidin agarose resin | Pierce | 20357 |
| **Chemical compounds** | | |
| NMS-873 | Cayman Chemical | 17674 |
| CB-5083 | Cayman Chemical | 19311 |
| ML-240 | Cayman Chemical | 17373 |
| Dithiobis(succinimidyl propionate) (DSP) | Thermo Scientific | 22585 |
| 2-(4-((Bis((1-(tert-butyl)-1H-1,2,3-triazol-4-yl)methyl) amino)methyl)-1H-1,2,3-triazol-1-yl)acetic acid (BTTAA) | Click Chemistry Tools | 1236 |
| Carboxytetramethylrhodamine (TAMRA)-Azide-Polyethylene Glycol (PEG)-Desthiobiotin | BroadPharm | BP-22475 |
| DharmaFECT1 Transfection Reagent | Dharmacon | T-2001 |
| Lipofectamine 3000 | Invitrogen | L3000 |
| **Commercial assays** | | |
| Nano-Glo Luciferase Assay System | Promega | N1110 |
| **Cell lines** | | |
| Human HEK293 Flp-In | Thermo Fisher | R75007 |
| Fischer Rat Thyroid (FRT) Flp-In | (Sabusap et al, 2016) | N/A |
| **Recombinant DNA** | | |
| Tg-FLAG - pcDNA3.1 + /C-(K)-DYK | Genscript | OHu20241 |
| pcDNA5/FRT | (Sabusap et al, 2016) | N/A |
| pOG44 | Thermo Fisher | V600520 |
| pcDNA5/FRT-Tg-FLAG | This study | N/A |
| pcDNA5/FRT-Tg-NLuc | This study | N/A |
| pMRX-INU-TEX264-FLAG | Addgene | 128258 |
| pMRX-INU-SEC62-FLAG | Addgene | 128263 |
| **Oligonucleotides** | | |
| PCR primers | This study | Dataset EV7 |
| **Software** | | |
| Bio-Rad Image Lab | Bio-Rad | https://www.bio-rad.com/en-us/product/image-lab-software |
| Prism 9 | GraphPad | https://www.graphpad.com/scientific-software/prism/ |
| Proteome Discoverer 2.4 | Thermo Fisher | https://www.thermofisher.com |

| Reagent/resource | Reference or source | Identifier or catalog number |
|---|---|---|
| CFX Maestro | Bio-Rad | https://www.bio-rad.com/en-us/product/cfx-maestro-software-for-cfx-real-time-pcr-instruments |
| FlowJo | BD Biosciences | https://www.flowjo.com/solutions/flowjo/downloads |

All unique/stable reagents generated, including plasmids and cell lines, are available from the corresponding author (lars.plate@vanderbilt.edu) with a complete Materials Transfer Agreement.

## Methods and protocols

### Plasmid production and antibodies

Tg-FLAG in pcDNA3.1 + /C-(K)-DYK plasmid was purchased from Genscript (Clone ID OHu20241). The Tg-FLAG gene was then amplified and assembled with an empty pcDNA5/FRT expression vector using a HiFi DNA assembly kit (New England BioLabs, E2621). This plasmid then underwent site-directed mutagenesis to produce pcDNA5-C1264R-Tg-FLAG, and pcDNA5-A2234D-Tg-FLAG plasmids.

An oligonucleotide fragment encoding Nanoluciferase (NLuc) was ordered from Genewiz. To generate pcDNA5/FRT-Tg-NLuc plasmids the Tg gene was amplified from the pcDNA5/FRT-Tg-FLAG plasmid and assembled with the NLuc fragment using a HiFi DNA assembly kit (New England BioLabs). To generate the respective mutant construct plasmids for A2234D and C1264R Tg, site-directed mutagenesis was performed using a Q5 polymerase (New England BioLabs). All oligonucleotide sequences used for site-directed mutagenesis can be found in Dataset EV7.

### Cell line engineering

FRT cells were cultured in Ham's F12, Coon's Modification (F12) media (Sigma, cat. No. F6636) supplemented with 5% fetal bovine serum (FBS), and 1% penicillin (10,000 U)/streptomycin (10,000 µg/mL). All cell lines were tested monthly to ensure they were free of mycoplasma contamination. To generate FRT flp-Tg-FT cells, $5 \times 10^5$ cells cultured for 1 day were cotransfected with 2.25 µg of flp recombinase pOG44 plasmid and 0.25 µg of FT-Tg pcDNA5 plasmid using Lipofectamine 3000. Cells were then cultured in media containing Hygromycin B (100 µg/mL) to select site-specific recombinants. Resistant clonal lines were sorted into single-cell colonies using flow cytometry and screened for FT-Tg expression (Appendix Fig. S1).

To generate HEK293 flp-Tg-NLuc cells, $5 \times 10^5$ cells cultured for 1 day were cotransfected with 2.25 µg of flp recombinase pOG44 plasmid and 0.25 µg of NLuc-Tg pcDNA5 plasmid using Lipofectamine 3000. Cells were then cultured in media containing Hygromycin B (100 µg/mL) to select site-specific recombinants. Polyclonal lines were screened for Tg-NLuc expression and furimazine substrate turnover (Appendix Fig. EV5).

### Time-resolved interactome profiling

Fully confluent 15-cm tissue culture plates of FRT cells were used. Cells were washed with phosphate-buffered saline (PBS) and depleted of methionine by incubating with methionine-free

Dulbecco's Modified Eagle Medium (DMEM) supplemented with 5% dialyzed fetal bovine serum (FBS), 1% L-glutamine (2 mM final concentration), 1% L-cysteine (200 µM final concentration), and 1% penicillin (10,000 U)/streptomycin (10,000 µg/mL) for 30 min. Cells were then pulse-labeled with Hpg-enriched DMEM supplemented with 1% Hpg (200 µM final concentration), 5% dialyzed FBS, 1% L-glutamine (200 mM final concentration), 1% L-cysteine (200 µM), and 1% penicillin (10,000 U)/streptomycin (10,000 µg/mL) for 1 h. After pulse labeling, cells were washed with F12 media containing tenfold excess methionine (2 mM final concentration). Cells were then cultured in normal F12 media supplemented with 5% FBS and chased for the specified timepoints. Cells were harvested by washing with PBS and then cross-linked with 0.5 mM DSP in PBS at 37 °C for 10 min. Cross-linking was quenched by the addition of 100 mM Tris pH 7.5 at 37 °C for 5 min. Lysates were prepared by lysing in Radioimmunoprecipitation assay (RIPA) buffer (50 mM Tris pH 7.5, 150 mM NaCl, 0.1% SDS, 1% Triton X-100, 0.5% deoxycholate) with protease inhibitor cocktail (Roche, 4693159001). Protein concentration was normalized to 1 mg/mL using a BCA assay (Thermo Scientific, 23225), and cell lysates underwent click reactions with the addition of 0.8 mM copper sulfate (diluted from a 20 mM stock in water), 1.6 mM BTTAA (diluted from a 40 mM stock in water), 5 mM sodium ascorbate (diluted from a 100 mM stock in water), and 100 µM TAMRA-Azide-PEG-Desthiobiotin ligand (diluted from a 5 mM stock in DMSO). Samples were allowed to react at 37 °C for 1 h while shaking at 750 rpm. Cell lysates were then precleared on 4B sepharose beads (Sigma, 4B200) at 4 °C for 1 h while rocking. Precleared lysates were immunoprecipitated with G1 Anti-DYKDDDDK affinity resin overnight at 4 °C while rocking. The resin was washed four times with RIPA buffer, and proteins were eluted twice in 100 µL immunoprecipitation elution buffer (2% SDS in PBS) by heating at 95 °C for 5 min. Eluted samples from FLAG immunoprecipitations were then diluted with PBS to reduce the final SDS concentration to 0.2%. The solutions then underwent streptavidin enrichment with high-capacity streptavidin agarose resin (Pierce, 20359) for 2 h at room temperature while rotating. The resin was then washed with 1 mL each of 1% SDS, 4 M Urea, 1 M sodium chloride, followed by a final wash with 1% SDS (all wash buffers dissolved in PBS). The resin was frozen overnight at −80 °C and samples were then eluted twice with 100 µL biotin elution buffer (50 mM Biotin in 1% SDS in PBS) by heating at 37 °C and shaking at 750 rpm for 1 h. Eluted streptavidin enrichment samples were precipitated in methanol/chloroform, washed three times with methanol, and air-dried. Protein pellets were then resuspended in 3 µL 1% Rapigest SF Surfactant (Waters, 186002122) followed by the addition of 10 µL of 50 mM HEPES pH 8.0, and 34.5 µL of water. Samples were reduced with 5 mM tris(2-carboxyethyl)phosphine (TCEP) (Sigma, 75259) at room temperature for 30 min and alkylated with 10 mM iodoacetimide (Sigma, I6125) in the dark at room temperature for 30 min. Proteins were digested with 0.5 µg of trypsin/Lys-C protease mix (Pierce, A40007) by incubating for 16-18 h at 37 °C and shaking at 750 rpm. Peptides were reacted with TMTpro 16plex reagents (Thermo Fisher, 44520) in 40% v/v acetonitrile and incubated for 1 h at room temperature. Reactions were quenched by the addition of ammonium bicarbonate (0.4% w/v final concentration) and incubated for 1 h at room temperature. TMT-labeled samples were then pooled and acidified with 5% formic acid (Fisher, A117, v/v).

Samples were concentrated using a speedvac and resuspended in buffer A (97% water, 2.9% acetonitrile, and 0.1% formic acid, v/v/v). Cleaved Rapigest SF surfactant was removed by centrifugation for 30 min at $21{,}100 \times g$.

For TRIP analysis coupled with ML-240 treatment, C1264R Tg-FRT cells were processed as described above with ML-240 (10 µM) supplemented in Hpg pulse media and throughout the chase period.

### Liquid chromatography–tandem mass spectrometry

MudPIT microcolumns were prepared as previously described (Fonslow et al, 2012). Peptide samples were directly loaded onto the columns using a high-pressure chamber. Samples were then desalted for 30 min with buffer A (97% water, 2.9% acetonitrile, 0.1% formic acid v/v/v). LC-MS/MS analysis was performed using an Exploris480 (Thermo Fisher) mass spectrometer equipped with an Ultimate3000 RSLCnano system (Thermo Fisher). MudPIT experiments were performed with 10 µL sequential injections of 0, 10, 30, 60, and 100% buffer C (500 mM ammonium acetate in buffer A), followed by a final injection of 90% buffer C with 10% buffer B (99.9% acetonitrile, 0.1% formic acid v/v) and each step followed by a 140 min gradient from 4 to 80% B with a flow rate of 500 nL/minute on a 20 cm fused silica microcapillary column (ID 100 µm) ending with a laser-pulled tip filled with Aqua C18, 3 µm, 125 Å resin (Phenomenex). Electrospray ionization (ESI) was performed directly from the analytical column by applying a voltage of 2.2 kV with an inlet capillary temperature of 275 °C. Data-dependent acquisition of mass spectra was carried out by performing a full scan from 400 to 1600 $m/z$ at a resolution of 120,000. Top-speed data acquisition was used for acquiring MS/MS spectra using a cycle time of 3 s, with a normalized collision energy of 32, 0.4 $m/z$ isolation window, automatic maximum injection time, and 100% normalized AGC target, at a resolution of 45,000 and a defined first mass ($m/z$) starting at 110. Peptide identification and TMT-based protein quantification was carried out using Proteome Discoverer 2.4. MS/MS spectra were extracted from Thermo Xcalibur.raw file format and searched using SEQUEST against a Uniprot rat proteome database supplemented with the human thyroglobulin gene (accessed 03/2014 and containing 28,860 entries). The database was curated to remove redundant protein and splice-isoforms. Searches were carried out using a decoy database of reversed peptide sequences and the following parameters: 20 ppm peptide precursor tolerance, 0.02-Da fragment mass tolerance, minimum peptide length of 6 amino acids, trypsin cleavage with a maximum of two missed cleavages, dynamic methionine modification of +15.995 Da (oxidation), dynamic protein N-terminus +42.011 Da (acetylation), −131.040 (methionine loss), −89.030 (methionine loss + acetylation), static cysteine modification of +57.0215 Da (carbamidomethylation), and static peptide N-terminal and lysine modifications of +304.2071 Da (TMTpro 16plex).

### Immunoblotting and SDS-PAGE

Cell lysates were prepared by lysing in RIPA buffer with protease inhibitor cocktail (Roche), and protein concentrations were normalized using a BCA assay (Thermo Scientific). Lysates were then denatured with 1× Laemmli buffer + 100 mM DTT and heated at 95 °C for 5 min before being separated by SDS-PAGE. Samples were transferred onto poly-vinylidene difluoride (PVDF) membranes (Millipore, IPFL00010) for immunoblotting and blocked using 5% nonfat dry milk dissolved in tris-buffered saline with 0.1% Tween-20 (Fisher, BP337-100) (TBS-T). Primary antibodies were incubated either at room temperature for 2 h, or overnight at 4 °C. Membranes were then washed three times with TBS-T and incubated with secondary antibody in 5% nonfat dry milk dissolved in TBS-T either at room temperature for 1 h or overnight at 4 °C. Membranes were washed three times with TBS-T and then imaged using a ChemiDoc MP Imaging System (Bio-Rad). Primary antibodies were acquired from commercial sources and used at the indicated dilutions in immunoblotting buffer (5% bovine serum albumin (BSA) in Tris-buffered saline pH 7.5, 0.1% Tween-20, and 0.1% sodium azide): KDEL (1:1000), M2 anti-FLAG (1:1000), PDIA4 (1:1000), thyroglobulin (1:1000). Tubulin-Rhodamine primary antibody was obtained from commercial sources and used at 1:10000 dilution in 5% milk in Tris-buffered saline pH 7.5, 0.1% Tween-20 (TBS-T). Secondary antibodies were obtained from commercial sources and used at the indicated dilutions in 5% milk in TBS-T: Goat anti-mouse Star-bright700 (1:10,000), Goat anti-rabbit IRDye800 (1:10,000).

### qRT-PCR

RNA was prepared from cell pellets using the Quick-RNA miniprep kit (Zymo Research). cDNA was synthesized from 500 ng total cellular RNA using random hexamer primer (IDT), oligo-dT primer (IDT), and M-MLV reverse transcriptase (Promega). qPCR analysis was performed using iTaq Universal SYBR Green Supermix (Bio-Rad) combined with primers for genes of interest and reactions were run in 96-well plates on a Bio-Rad CFX qPCR instrument. Data analysis was then carried out in CFX Maestro (Bio-Rad). All oligonucleotide sequences used for qRT-PCR can be found in Dataset EV7.

### siRNA screening assay

Proteins previously identified as Tg interactors, and other key proteostasis network components were selected and targeted using an siGENOME SMARTpool siRNA library (Dharmacon) (Wright et al, 2021). HEK293 cells stably expressing Tg-NLuc constructs were seeded into 96-well plates at $2.5 \times 10^4$ cells/well and transfected using DharmaFECT1 following the DharmaFECT1 protocol (Dharmacon) with a 25 nM siRNA concentration. Approximately 36 h after transfection, cells were washed with PBS and replenished with fresh DMEM media. After 4 h, Tg-NLuc abundance in the lysate and media were measured using the nano-glo luciferase assay system according to the manufacturer protocol (Promega). Four controls were included for the experiments, including a non-targeting siRNA control, a siGLO fluorescent control to monitor transfection efficiency, a vehicle control containing transfection reagents but lacking any siRNAs, and a lethal TOX control. Data was median normalized across individual 96-well plates (Chung et al, 2008). Data represents two independent experiments for WT-NLuc and A2234D-NLuc, and three independent experiments for C1264R NLuc. Cutoff criteria for hits were set to those genes that increased or decreased Tg-NLuc abundance in lysate or media by 3σ.

### FRT siRNA validation studies

For siRNA silencing follow-up experiments, Tg-FRT cells were seeded into six-well dishes at $6.0 \times 10^5$ cells/well transfected using

DharmaFECT1 following the DharmaFECT protocol (Horizon) with a 25 nM siRNA concentration. For NAPA complementation experiments, siRNA (25 nM) and corresponding plasmid (0.83 μg per six-well dish) were cotransfected with DharmaFECT Duo (Horizon) Approximately 36 h after transfection, cells were washed with PBS and harvested for qRT-PCR for western blotting. For qRT-PCR siRNA target transcript levels were normalized to a GAPDH loading control to monitor siRNA silencing efficiency. For immunoblot analysis ~36 h after transfection cells were washed with PBS and replenished with fresh DMEM media. After 4 h in the case of WT Tg and 8 h in the case of C1264R or A2234D Tg, cells and media samples were harvested. Cells were lysed with 1 mL of RIPA with protease inhibitor cocktail (Roche), and lysate and media samples were subjected to immunoprecipitation with G1 Anti-DYKDDDDK affinity resin overnight at 4 °C while rocking. After three washes with RIPA buffer, protein samples were eluted with 3× Laemmli buffer with 100 mM DTT heating at 95 °C for 5 min. Immunoblot quantification was performed using Image Lab Software (Bio-Rad).

### $^{35}S$ pulse-chase assay

Confluent six-well dishes of FRT cells (approximately $1 \times 10^6$/well) were metabolically labeled in DMEM depleted of methionine and cysteine and supplemented with EasyTag $^{35}S$ Protein Labeling Mix (Perkin Elmer, NEG772007MC), glutamine, penicillin/streptomycin, and 10% dialyzed FBS at 37 °C for 30 min. Afterward, cells were washed twice with F12 media containing 10× methionine and cysteine, followed by a burn-off period of 10 min in normal F12 media. Cells were then chased for the respective time periods with normal F12 media, lysed with 500 μL of RIPA buffer with protease inhibitor cocktail (Roche) and 10 mM DTT. Cell lysates were diluted with 500 μL of RIPA buffer with protease inhibitor cocktail (Roche) and subjected to immunoprecipitation with G1 anti-DYKDDDDK affinity resin overnight at 4 °C. After three washes with RIPA buffer, protein samples were eluted with 3× Laemmli buffer with 100 mM DTT heating at 95 °C for 5 min. Eluted samples were then separated by SDS-PAGE, and gels were dried and exposed on a storage phosphor screen. Radioactive band intensity was then measured using a Typhoon Trio Imager (GE Healthcare) and quantified by densitometry in Image Lab (Bio-Rad).

For pulse-chase analysis coupled with ML-240 treatment, cells were pre-treated with vehicle (0.1% DMSO) or ML-240 (10 μM) for 15 min prior to $^{35}S$ pulse labeling. Vehicle (0.1% DMSO) or ML-240 (10 μM) treatment was then maintained throughout the pulse labeling and chase period.

### VCP pharmacological inhibition studies

For the follow-up experiment with VCP inhibitors, confluent six-well dishes of C1264R Tg-FT FRT cells were washed with PBS and fresh F12 media supplemented with ML-240 (10 μM), CB-5083 (5 μM), NMS-873 (10 μM), or vehicle (0.1% DMSO) was added and incubated for 8 h. For WT Tg-FT cells, confluent six-well dishes were similarly used, washed with PBS, and fresh F12 media supplemented with ML-240 (10 μM) or vehicle (0.1% DMSO) was added and incubated for 4 or 8 h. Cells were lysed with 1 mL of RIPA with protease inhibitor cocktail (Roche), and lysate and media samples were subjected to immunoprecipitation with G1 Anti-DYKDDDDK affinity resin overnight at 4 °C while rocking.

After three washes with RIPA buffer, protein samples were eluted with 3× Laemmli buffer with 100 mM DTT heating at 95 °C for 5 min. Immunoblot quantification was performed using Image Lab Software (Bio-Rad).

Viability with ML-240 was monitored using propidium iodide staining. Briefly, cells were treated with either ML-240 (10 μM) or vehicle (0.1% DMSO) for 4 h before being harvested and stained with propidium iodide (1 μg/mL) at room temperature for 15 min in the dark. Cells permeabilized with 0.2% Triton were used as a positive staining control. Staining intensity of cells was then analyzed by flow cytometry and data analysis was carried out in FlowJo (BD Biosciences).

### TEX264 & Tg-NLuc co-immunoprecipitation studies

HEK293 flp-Tg-NLuc cells were cultured at $1.0 \times 10^5$ cells/well in 12-well tissue culture dishes for 1 day and transfected with either a fluorescent control, C-terminal FLAG-tag TEX264, or C-terminal FLAG-tag SEC62 using a calcium phosphate method. Confluent plates were harvested by lysing with 300 μL of TNI buffer (50 mM Tris pH 7.5, 150 mM NaCl, 0.5% IGEPAL CA-630 (Sigma-Aldrich)) and protease inhibitor (Roche). Lysates were sonicated for 10 min at room temperature and normalized using a BCA Assay (Thermo Scientific). Cell lysates were then precleared on 4B sepharose beads (Sigma, 4B200) at 4 °C for 1 h while rocking, then immunoprecipitated with G1 Anti-DYKDDDDK affinity resin overnight at 4 °C while rocking. The resin was washed four times with TNI buffer and resuspended in 250 μL of TNI buffer. In total, 50 μL aliquots were taken and measured using the nano-glo luciferase assay system according to the manufacturer protocol (Promega). For differential enrichment, statistical analysis was performed in Prism 9 (GraphPad) using a one-way ANOVA with post hoc Tukey's multiple testing corrections.

For western blot analysis HEK293 flp-Tg-NLuc ($1.0 \times 10^6$ cells/dish in 10-cm tissue culture dishes) were cultured for 1 day and transfected with either a fluorescent control, TEX264, or SEC62 using a calcium phosphate method. Cells were collected and lysed using 300 μL of TNI buffer with protease inhibitor (Roche) sonicated at room temperature for 10 min. Lysates were normalized, precleared, and immunoprecipitated as described above. Proteins were eluted using 6× Laemmli buffer + 100 mM DTT, separated by SDS-PAGE and transferred to PVDF membrane (Millipore) for immunoblotting.

### Mass spectrometry interactomics and TMT quantification data analysis

To identify Tg interactors (−) biotin pulldown vs (−) Hpg samples were processed using the DEP pipeline (Zhang et al, 2018). Enriched proteins were determined based on those with a log2 fold change of 2σ and Benjamini–Hochberg adjusted $P$ value (false discovery rate) of 0.05. For pathway enrichment analysis of identified proteins, EnrichR was used and GO Cellular Component and Molecular Function 2018 terms were used to differentiate secretory pathway associated proteins from background (Chen et al, 2013). For time-resolved analysis, data were processed in R with custom scripts. Briefly, TMT abundances across chase samples were normalized to Tg TMT abundance as described previously and compared to (-) Hpg samples for enrichment analysis (Wright et al, 2021). For relative enrichment analysis, the means of log2 interaction differences were scaled to values from 0 to 1, where a

value of 1 represented the timepoint at which the enrichment reached the maximum, and 0 represented the background intensity in the (−) Hpg channel. Negative log2 enrichment values were set to 0 as the enrichment fell below the background.

Inconsistencies in the quantification of Tg bait protein were observed for replicate 5 of WT Tg Hpg-chase samples, likely due to sample loss during the enrichment. Hence, this replicate was excluded from further time-resolved analysis. All analysis scripts are available as described in "Data availability".

## Data availability

Mass spectrometry spectrum and result files are available via ProteomeXchange under identifier PXD035681. All raw and processed TMT quantification data has been provided (Datasets EV1–4). Code used for data analysis and generation of figures is available at https://github.com/Plate-Lab/Wright_TRIP.

The source data of this paper are collected in the following database record: biostudies:S-SCDT-10_1038-S44320-024-00058-1.

## Peer review information

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

## Acknowledgements

This work was supported by the following funding agencies: National Science Foundation—Graduate Research Fellowship Program (MTW), National Institute of General Medical Sciences (R35GM133552) (LP). This content is solely the responsibility of the authors and does not necessarily represent the official views of the National Science Foundation or National Institutes of General Medical Sciences. The authors thank the VUMC Flow Cytometry Shared Resource Core for assistance with cell sorting.

## Author contributions

**Madison T Wright**: Conceptualization; Data curation; Formal analysis; Funding acquisition; Investigation; Visualization; Methodology; Writing—original draft; Writing—review and editing. **Bibek Timalsina**: Data curation; Formal analysis; Investigation; Writing—review and editing. **Valeria Garcia Lopez**: Data curation; Formal analysis; Investigation; Visualization; Writing—review and editing. **Jake N Hermanson**: Data curation; Visualization; Writing—review and editing. **Sarah Garcia**: Data curation; Formal analysis; Investigation; Visualization; Writing—review and editing. **Lars Plate**: Conceptualization; Data curation; Formal analysis; Supervision; Funding acquisition; Investigation; Visualization; Methodology; Writing—original draft; Project administration; Writing—review and editing.

Source data underlying figure panels in this paper may have individual authorship assigned. Where available, figure panel/source data authorship is listed in the following database record: biostudies:S-SCDT-10_1038-S44320-024-00058-1.

## Disclosure and competing interests statement

The authors declare no competing interests.

# Expanded View Figures

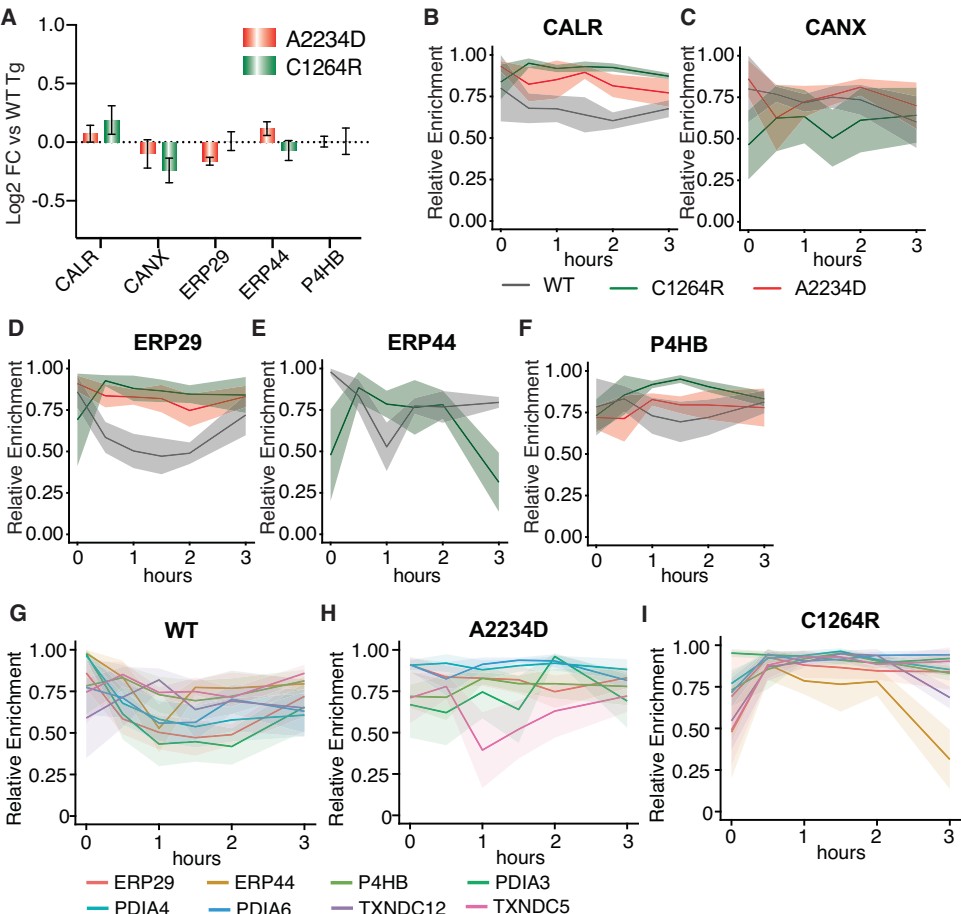

**Figure EV1. TRIP data for individual interactors.**

(A) Aggregate (steady-state) interactomics data comparing the enrichment of Tg interactors for mutant Tg to WT Tg (data from Wright et al, 2021). Interactions are mostly unchanged for mutant Tg relative to WT Tg. Data is represented as mean ± SEM ($N = 12$ biological replicates for C1264R, and 6 biological replicates for A2234D). (B–F) Plots comparing the relative enrichment of interactors CALR (B), CANX (C), ERP29 (D), ERP44 (E), and P4HB (F) throughout the TRIP time course for WT, A2234D, and C1264R Tg. TRIP data can resolve dynamic interaction changes for several mutant Tg interactors, while these changes are muted in the aggregate data. Solid line corresponds to mean and shading represents the SEM ($N = 5$ for WT Tg; $N = 6$ biological replicates for A2234D and C1264R Tg). (G–I) Plots comparing the relative enrichment of individual disulfide/redox-processing interactors throughout the TRIP time course for WT (G), A2234D (H), and C1234R (I). Individual protein disulfide isomerases exhibit distinct peak times when interactions reach maximum, thereby revealing an order to their engagement. For instance, PDIA3, PDIA4, PDIA6, and P4HB peak at 0 h for WT Tg, while TXNDC12 peaks later at 1 h. Moreover, the exact temporal sequence of PDI engagements is shifted for A2234D (H) and C1234R Tg (I). Solid line corresponds to mean and shading represents the SEM ($N = 5$ for WT Tg; $N = 6$ biological replicates for A2234D and C1264R Tg). Data available in Dataset EV4.

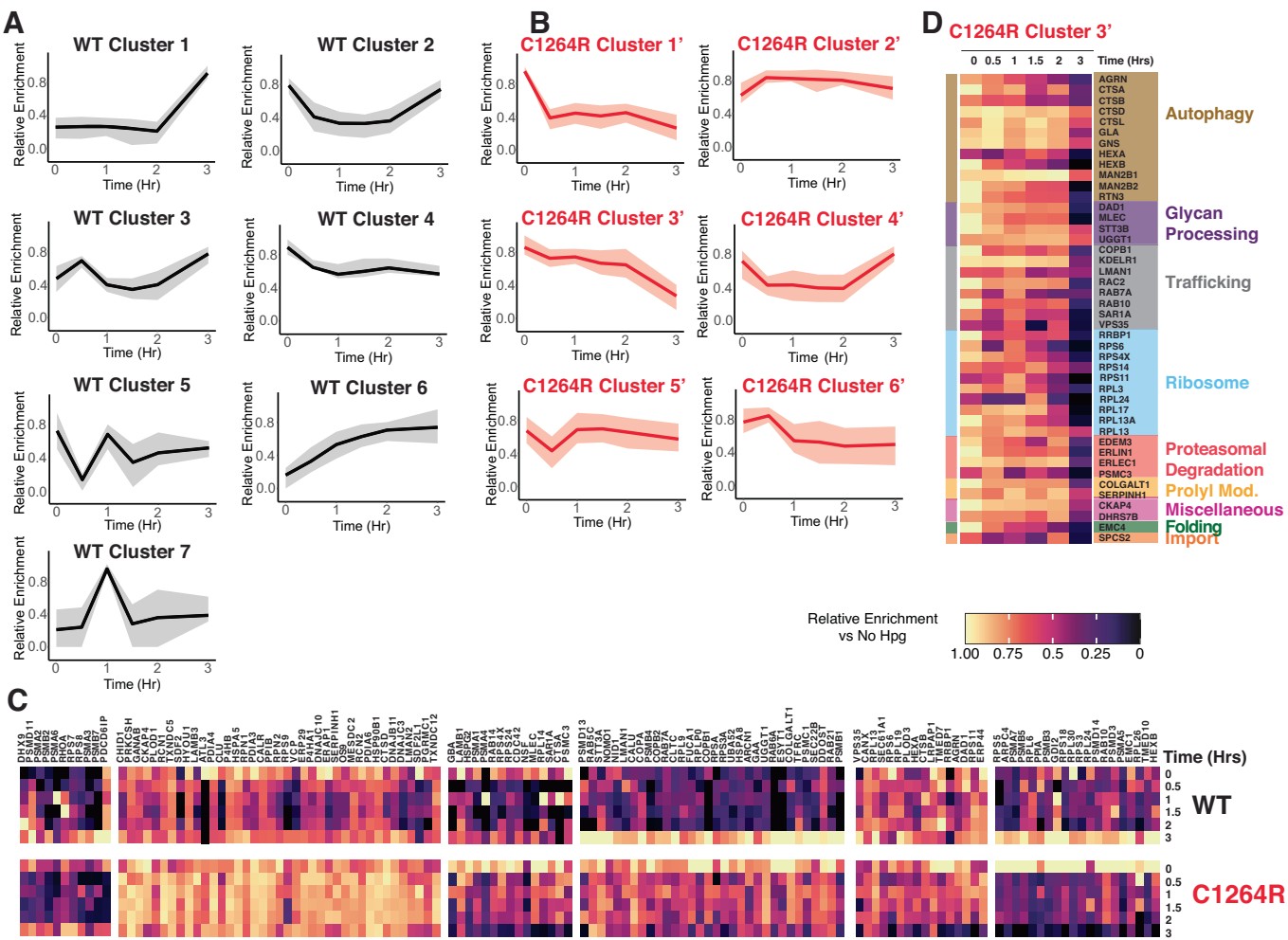

**Figure EV2. Unbiased clustering of TRIP data.**

(A, B) Unbiased k-means clustering of TRIP profiles for WT (A) and C1264R (B) to determine co-regulated groups of interactors. k-means clustering was carried out by using the k-means function from the tslearn python package with the data being normalized using the scaler mean variance function. This analysis resulted in 7 distinct clusters for WT and 6 clusters for C1264R. The line corresponds to the mean scaled log2 fold enrichment and the shading represents the 25–75 % quarter range within each cluster. (C) Heatmap showing unbiased k-means clustering of the combined WT and C1264R Tg TRIP profiles. Only interactors identified in both datasets were included. (D) Heatmap for interactors in C1264R Cluster 3 (from B), which displayed the strongest interactions at the initial 0 h timepoint. The scaled log2 fold change enrichment for individual interactors is shown, and the individual interactors are grouped by pathway. Several interactors related to autophagy (brown) and glycan processing (purple), including the glycoprotein folding sensor UGGT1, are present in this cluster.

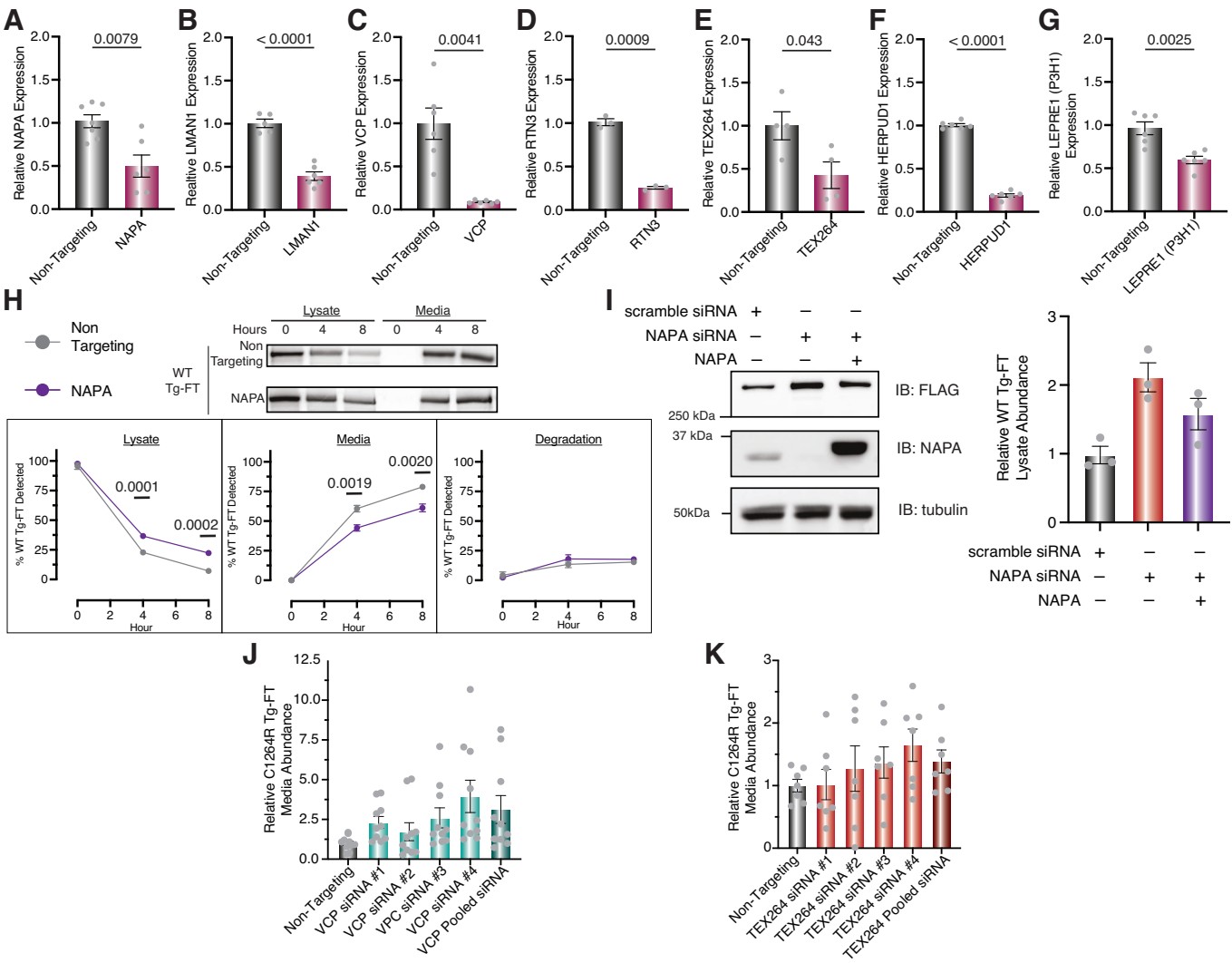

**Figure EV3. Validation of siRNA screening hits in FRT cells.**

(A–G) Relative expression of knockdown targets in engineered WT Tg-FT FRT cells with and without siRNA silencing measured by qRT-PCR. Data was first normalized to a GAPDH loading control followed by normalization to median expression of non-targeting transfected samples and represented as mean ± SEM. (A) NAPA (α-SNAP), (B) LMAN1, (C) VCP, (D) RTN3, (E) TEX264, (F) HERPUD1, (G) LEPRE11 (P3H1). Statistical testing was performed using an unpaired Student's $t$ test with Welch's correction with $P$ values as indicated. $N = 3$–7 biological replicates as shown. Primers for detection are described in Dataset EV7. (H) Pulse-chase analysis of WT Tg-FT in FRT cells with NAPA (α-SNAP) siRNA knockdown. Approximately 36 h after transfection with 25 nM siRNAs cells were pulse-labeled with EasyTag $^{35}$S Protein Labeling Mix (Perkin Elmer, NEG772007MC) for 30 min and chased for 8 h, collecting samples at 0-, 4-, and 8-h time points. Data is normalized to timepoint of maximum Tg recovery and represented as mean ± SEM. % Degradation is defined as $\left[1 - \left(Tg_t^{lysate} + Tg_t^{media}\right)\right] \times 100$. Where $Tg_t^{lysate}$ is the fraction of Tg-FT detected in the lysate at a given timepoint $n$, and $Tg_t^{media}$ is the fraction of Tg-FT detected in the media at a given timepoint n. Statistical testing performed using an unpaired Student's $t$ test with Welch's correction with $P$ values as indicated. $N = 6$ biological replicates. (I) Complementation of NAPA knockdown partially reverses WT-Tg retention. FRT cells stably expressing WT Tg were cotransfected with NAPA siRNA and siRNA resistant NAPA expression plasmid. Cells were harvested 40 h post transfection and lysates were analyzed by Western blot to monitor changes in WT-Tg amounts. Quantification (mean ± SEM) is shown on the right ($N = 3$ biological replicates). (J, K) Individual siRNA knockdown of VCP (J) and TEX265 (K) in C1264R Tg-FRT cells to confirm the increase in Tg secretion. Cells were transfected with 25 nM siRNAs for 36 h, media exchanged and conditioned for 8 h, Tg-FT was immunoprecipitated from media samples, and Tg-FT amounts were analyzed via immunoblotting. Multiple individual siRNAs recapitulated the increase in C1264R secretion. Data is represented as mean ± SEM for $N = 10$ (J) or 7 (K) biological replicates.

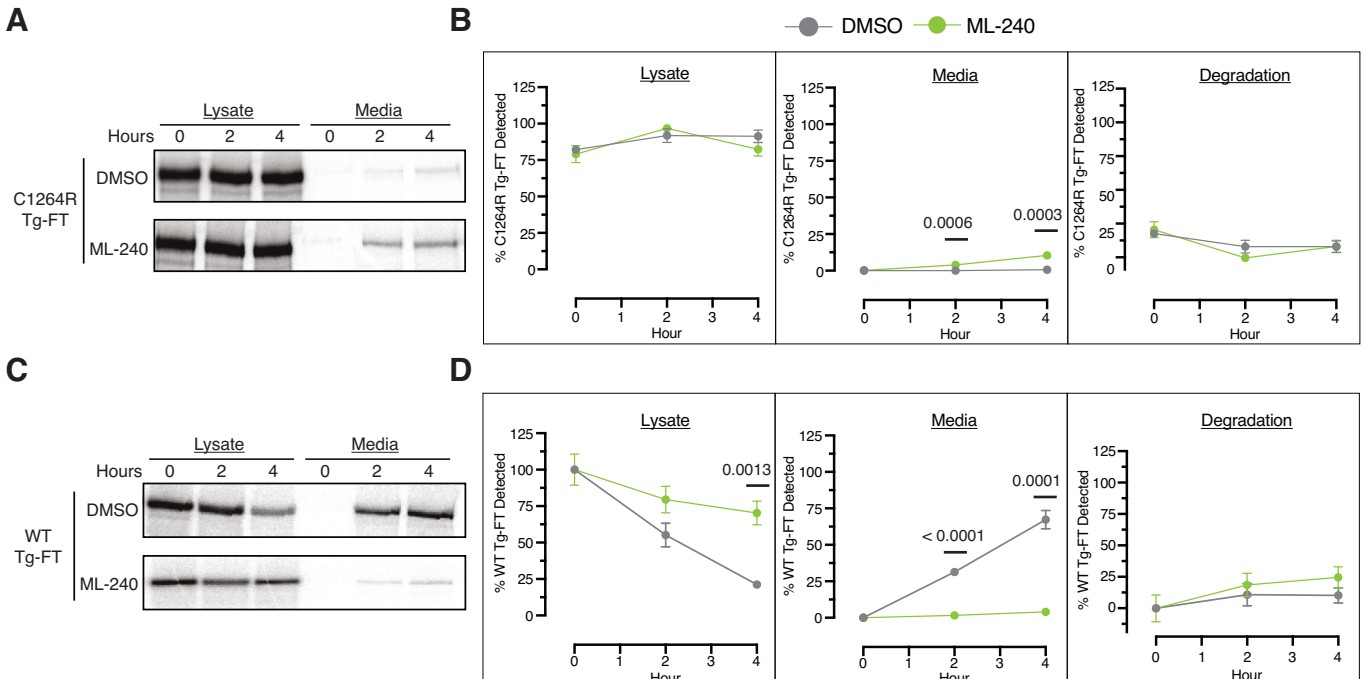

**Figure EV4. Pulse-chase analysis of WT and C1264R Tg with pharmacological VCP inhibition.**

Autoradiographs and quantifications of pulse-chase analysis of C1264R Tg-FT (**A, B**) and WT Tg-FT (**C, D**) in FRT cells with ML-240 treatment. Cells were pre-treated with ML-240 or DMSO for 15 min prior to pulse labeling with EasyTag $^{35}$S Protein Labeling Mix (Perkin Elmer, NEG772007MC) for 30 min and chased for 4 h with DMSO or ML-240 treatment, collecting samples at 0-, 2-, and 4-h time points. Autoradiographs from a representative experiment are shown in (**A, C**). Quantification is shown in (**C, D**). Data is normalized to the timepoint of maximum Tg recovery (C1264R) or 0 h (WT) and represented as mean ± SEM. Statistical testing was performed using an unpaired Student's *t* test with Welch's correction with *P* values as indicated. *N* = 5–6 biological replicates as shown.

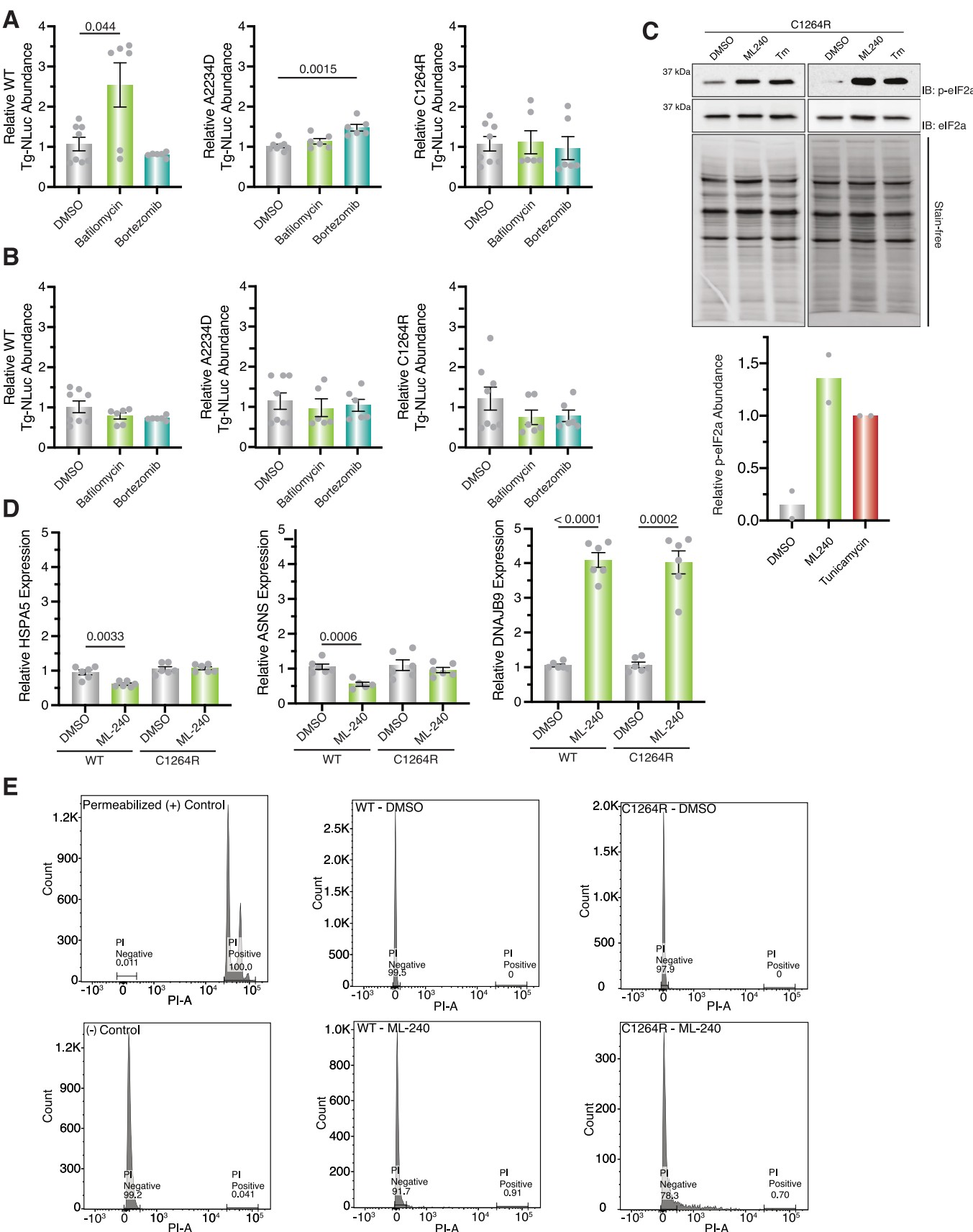

**Figure EV5.  TRIP of C1264R Tg-FT FRT cells with pharmacological VCP inhibition.**

(A, B) Effect of pharmacologic inhibition of protein degradation in Tg retention and secretion. HEK293T cells stably expressed Tg-NanoLuc variants were treated with proteasomal degradation inhibitor Bortezomib (10 µM) or lysosomal inhibitor Bafilomycin A1 (10 µM) for 8 h. Prior to treatment, the media was exchanged to condition secreted Tg. Tg lysate amounts (A) or media amounts (B) were then measured by the nano-glo luciferase assay system. Data is normalized to DMSO condition and represented as mean ± SEM. Statistical testing was performed using an unpaired Student's *t* test with Welch's correction with *P* values as indicated. *N* = 6–8 biological replicates as shown. (C, D) Assessment of UPR upregulation with ML-240 treatment. (C) Western blot analysis to assess phospho-eIF2α levels in of C1264R Tg-FT FRT cells treated with VCP inhibitor ML-240 (10 µM), tunicamycin (1 µg/mL), or vehicle (0.1% DMSO) for 2 h. Lysate samples were analyzed via immunoblotting. Data is normalized to the mean C1264R Tg-FT abundance of tunicamycin-treated samples. Data represented as mean ± SEM from *N* = 2 biological replicates. (D) Activation of UPR markers monitored via qPCR in C1264R Tg-FT FRT cells treated with ML-240 (10 µM) for 3 h. HSPA5 and ASNS expression remained unchanged in C2164R Tg-FT FRT cells but led to a significant decrease in WT Tg-FT FRT cells. Only DNAJB9 showed a significant increase in transcript levels for both C1264R and WT Tg-FT FRT cells. This suggest that ML-240 dependent rescue of C1264R Tg is not due to global remodeling of the ER proteostasis network via UPR activation. Data was first normalized to a GAPDH loading control followed by normalization to median expression of DMSO-treated samples and represented as mean ± SEM. Statistical testing performed using an unpaired Student's *t* test with Welch's correction with *P* values as indicated. *N* = 6 biological replicates. Primers for detection are described in Dataset EV7. (E) Viability analysis using Propidium iodide control samples, WT & C164R Tg-FT FRT cells with ML-240 treatment. Cells were treated with DMSO or ML-240 (10 µM) for 4 h, harvested, and stained with propidium iodide (1 µg/mL). FRT cells permeabilized with 0.2% Triton were used as a positive staining control. Unstained, non-permeabilized FRT cells were used as a negative staining control.

