## [Peer Review File · Molecular Systems Biology]

Time-Resolved Interactome Profiling Deconvolutes Secretory Protein Quality Control Dynamics

Madison Wright, Bibek Timalsina, Valeria Garcia Lopez, Jake Hermanson, Sarah Garcia, and Lars Plate

Corresponding author(s): Lars Plate (lars.plate@vanderbilt.edu)

Review Timeline:

Transfer from Review Commons:	23rd May 24
Editorial Decision:	27th Jun 24
Revision Received:	15th Jul 24
Accepted:	22nd Jul 24

Review
COMMONS

Editor: Jingyi Hou

Transaction Report:

Review #1

1. Evidence, reproducibility and clarity:

Evidence, reproducibility and clarity (Required)

The authors report a mass spectrometry (MS)-based interactomics technique, time-resolved interactome profiling (TRIP), which allows for tracking temporal changes in the interactome of protein of interest. To show that TRIP can successfully deconvolute interactomes over time, they pulsed thyroid cells with homopropargylglycine (Hpg), immunoprecipitated the Hpg incorporated thyroglobulin (Tg) and its interacting proteins at different time points, and subjected the samples to tandem mass tag (TMT)-based quantitative MS analysis. The MS results show that WT and variant Tg proteins indeed associate with different proteostasis network factors in a differential manner over the course of time. In addition, they utilized an siRNA-based luciferase fusion assay to evaluate whether silencing each proteostasis network component changes the levels of Tg in both lysate and media. From the combination of the TRIP and siRNA-based assays, they found many hits, including hits implicated in protein degradation, VCP and TEX264, which they validated with multiple experiments.

I am overall quite positive and think this is an important study. But there are some meaningful points to consider.

****Significant comments:****

1. Only two replicates of the main data (the TRIP-MS experiments) for this paper is problematic. Especially since the manuscript is supposed to be demonstrating and validating the new technique. Consistent with this concern, the relative enrichment profiles for some of the results were surprising. For instance, interaction with CCDC47 was tapering off but then at 3 h it suddenly reaches the maximum level of engagement. Is this a real finding or the variability in the method? Impossible to tell with two replicates. Presenting heat maps based on biological duplicates is also very problematic. It masks the error, which is large as can be seen in some of the panels showing individual proteins. In my view, triplicates and a clear understanding of the error in the technique should be required.
2. The same concern arises for the high-throughput siRNA screen, which was performed only in duplicate for WT and A2234D.
3. There are issues with some of the immunoprecipitation experiments: In Figure 1C, a negative control for FLAG IP is missing. In Figure 2B, I am curious why the band

(Hpg -, chase time 0 h) is so faint for the first WB (IB for FLAG) - is Hpg treatment indeed leading to much more Tg present at 0 h? If so, that is a concern. Also, a negative control must be included (either plain cells or cells expressing fluorescent protein or a different epitope-tagged WT Tg). In this same figure, I am puzzled why the bands for 1.5-3 timepoints in Biotin PD elution, probed for Rhodamine, are very faint especially considering that in Figure 1D, the corresponding bands, which are 4 h after the pulse, look fine. It seems like the IP failed here?

****Suggestion to consider:****

This manuscript, supported by the title and abstract, mainly focuses on the presentation of the development and application of TRIP, which is highly significant. The story becomes less coherent and harder to follow as significant amounts of text/figures are dedicated to siRNA-based high throughput screening and follow-up. In addition, although the discovery of TEX264 as one of the hits is very interesting and exciting, TEX264 apparently was not a hit in the TRIP experiment and is pretty distracting from the main point of the paper highlighted in the abstract and title, therefore. The siRNA-based assay and follow-up studies could be a separate scientific story of their own. Especially considering my concerns on the number of replicates for both the TRIP and siRNA-based assay, it could be beneficial to actually split the manuscript into two and conduct more replicates of the -omic work, which should corroborate the exciting discoveries the authors have made.

****Minor comments:****

Throughout the manuscript, the authors have not defined what FT is; presumably it means FLAG tag.

The authors might discuss their rationale for choosing 0-3 hrs for their TRIP studies. That includes any relevant information about the half-life of WT versus variant Tg, whether the Hpg pulse time is short enough to avoid missing key features of the temporal interactome, and discussion of what would happen if the TRIP were performed at prolonged time points (e.g. 6-10 h).

Lines 68-69: the two citations should probably come one sentence earlier (at least Coscia et al 2020 is a structure paper).

Line 91: "(Figure 1A)" should follow the sentence "To develop the time-resolved..." to help readers better understand the system.

Line 101: Fisher should be Fischer

Line 131: Should be 1.5 hrs instead of 2 hrs.

Lines 135-136: I do not agree with the claim that HSPA5 profile looked similar for MS and WB. I do not see a peak for HSPA5 at 2 hrs in Figure 2D.

Line 186: The cited paper Shurtleff et al 2018 is missing in the reference list.

Line 188: I disagree with the authors' claim here because, at least for CCDC47, interactions with C1264R seem to come back at the 3 hr time point.

Line 203: I am not sure if P4HA1 can be included in the examples for showing distinct patterns for mutants compared to the WT according to their data in Figure 3H.

Line 216: The authors should add citations about the functions of STT3A and STT3B proteins.

Lines 248-251, "We found that interactions with these components...": this sentence should refer to Figure 3 - Figure Supplement 3 instead of Figure 3L and S4.

Lines 258-260, "Another striking observation was that the temporal profile of EMC interactions for C1264R correlated with RTN3, PGRMC1, CTSB, and CTSD interactions.": Please provide more evidence to support the potential correlation between different interaction profiles. Or the authors should move this sentence to the discussion section as it sounds speculative. This highlights the issue of only having duplicates, as well.

Line 340: As written, should cite more than one paper

Line 371: Should be Figure 4 - figure supplement 2

Line 1231: "Zhang et al 2018" needs to be removed

Line 1286: FRTR should be FRT

Figure 3E: Color used to highlight the three proteins (CCDC47, EMC1, EMC4) should match the color used in Figure 3 - Figure Supplement 3

Figure 4A: The bottom figure where lysate signal is inversely proportional to time is misleading because the authors are assessing steady-state level of proteins in this assay.

Figure 4 - Figure Supplement 1 caption: in (C), (F) should be (B). (K) should be (G) and I am not sure what the authors mean when they refer to (J) in caption of (G).

Figure 5 caption for (C and D): Need to specify the time that the samples were collected (8 hrs), as it seems different from A and B according to the main text.

Figure 5 - Figure Supplement 1: Data for HERPUD1 and P3H1 should be included.

Figure 5 - Figure Supplement 2B: Please mention in the caption how degradation is defined.

2. Significance:

Significance (Required)

This manuscript is highly significant because the authors (1) designed and validated a new methodology for time-resolved interactomics study, (2) presented the dynamic changes in Tg interactome for WT and variants, and (3) discovered how proteins implicated in degradation pathways (e.g. VCP, TEX264, RTN3) can change the secretion profile of WT and mutant Tg proteins. With TRIP, the authors demonstrated that they could obtain valuable data that were previously not captured from steady-state interactomics studies (Wright et al. 2021; Figure 3M and Figure 3 - Figure supplement 4D-4I). Furthermore, the authors treated cells with VCP inhibitors and performed both 35S pulse-chase analyses and TRIP. These experiments provide valuable information to the field by (1) presenting a new method to rescue Tg secretion defect, and (2) demonstrating a broader applicability of TRIP. If the major comments above can be addressed I believe this is a tremendous contribution to the field.

3. How much time do you estimate the authors will need to complete the suggested revisions:

Estimated time to Complete Revisions (Required)

(Decision Recommendation)

Between 3 and 6 months

No

Review #2

1. Evidence, reproducibility and clarity:

Evidence, reproducibility and clarity (Required)

In the manuscript 'Time-Resolved Interactome Profiling Deconvolutes Secretory Protein Quality Control Dynamics' Wright et al. developed an approach for time-resolved protein protein interaction mapping relying on pulsed unnatural amino acid incorporation, protein cross linking, sequential affinity purification, and quantitative mass spectrometry named time-resolved interactome profiling (TRIP). The authors applied the TRIP method to compare the interactions of the secreted thyroid prohormone thyroglobulin (Tg) comparing the WT protein to secretion-defective mutations implicated in congenital hypothyroidism. They further employed an RNA interference screening platform (1) to investigate if (1) interactors identified via TRIP are functionally relevant for Tg protein quality control and (2) to identify factors that can rescue mutant Tg secretion. The screen was initially performed in HEK293 cells, but selected hits with a phenotype in HEK cells were then followed up in Fisher rat thyroid cells. Further functional validation was performed by pharmacologic inhibition of VCP, a hit from the RNAi screen with an effect on Tg lysate abundance and Tg secretion. While the authors present a comprehensive study including identification of protein-protein interactions using proteomics followed up by an RNA interference screen for functional validation, major comments need to be addressed for both the proteomics as well as the functional genomics aspects of the study (see comments below).

****Major comments:****

- The authors describe a new method for quantitative, temporal interaction mapping. The protocol involves two enrichment steps as well as several reactions including cross-linking of the samples as well as functionalization of the unnatural amino acids. Given all these steps, the authors should rigorously characterize the quantitative reproducibility of the experiment when performed in independent biological replicates. This is important because in the final quantitative MS experiment, the authors only use two biological replicates, which is too low especially for such an involved sample preparation procedure, which would expect to have a high variability between replicates. Given the low number of replicates and the unknown reproducibility of the quantification for this protocol, it is questionable at this point how reliable the quantification over the time course is.
- Compared to the previous dataset published last year, the authors discover an overlap in interactors, but also a huge discrepancy, with 96 previously identified interactors not detected in the current study, but 198 additional interactors identified. How do the authors explain the big differences between these datasets?
- For the temporal interaction analysis the authors describe differences in the temporal profiles of selected interactions comparing wt and mutant, however no statistical analysis is performed comparing wt and mutant interaction profiles across the time course. Furthermore the variability between the replicates for the temporal profiles is not shown and some of the temporal profiles appear to be noisy. A more rigorous statistical analysis should be performed including additional biological replicates to evaluate the changes over the time course, especially as the temporal interaction analysis is the novelty of this study.
- To functionally validate interactors derived from the TRIP analysis as well as to identify factors that can rescue mutant Tg secretion the authors developed an RNA interference screen. There are a number of aspects that need to be addressed/clarified for this part of the study.
- While the authors validate the stable cell lines expressing the nanoluciferase tagged Tg and the linearity of luminescence signal in lysate and media carefully, they do not validate their platform in combination with the RNAi knockdown strategy. The authors should select genes as positive controls that are expected to modulate Tg secretion and demonstrate that the knockout of these positive controls indeed results in changes in Tg secretion in their system.
- For the screen the authors select 167 Tg interactors and PN (Proteostasis network) related factors. This statement is very vague and the authors should clarify which genes were knocked down and which criteria were applied to narrow down the list of interactors and to select PN factors. The authors should therefore provide a supplementary table including all genes included in the screen, their source (were this derived from the initial study by Wright et al, from the current study or compiled from

prior knowledge about PN), as well as their results from the screen based on luminescence in media and lysate. It is unclear how many of the selected factors are actually coming from the TRIP analysis.

- Only a small number of the 167 selected genes shows an effect on Tg abundance/secretion. How do the authors explain this result? Would we not expect that Tg interactors, especially those from the TRIP method which interact with the newly synthesized are more enriched for functionally relevant genes.

- The authors initially performed the screen in HEK293 cells and as a second step wanted to validate the hits from the HEK cells in more relevant Fisher rat thyroid cells. Indeed they could show that knockdown of NAPA increased WT TG in lysate and decreased WT Tg secretion. Furthermore, they further validated genes to modulate mutant Tg lysate and media abundance. The authors should perform a rescue experiment to demonstrate that the observed phenotype can be reversed through re-introduction of NAPA.

- One hit from this analysis was the ER-phagy receptor TEX264, while TEX264 was not identified in the TRIP data, is selectively increased the C1264R secretion, but not wt and the other Tg mutant. Following Co-IP data however revealed some interaction between the C1264R and to a lesser extent the A2234D mutant. How do the authors explain that TEX264 was missed in the TRIP dataset?

****Minor comments:****

- The workflow needs to be described clearer. For example, it should be better explained why the authors selected a two stage enrichment strategy, I assume that the first based on the Flag affinity tag is to purify the protein of interest and the second step based on the incorporation and functionalization of the unnatural amino acids to enrich for the newly synthesized fraction at specific time points after protein synthesis? These are critical steps for the method but the rationals are not well explained, neither in the text nor the figures captures all these steps of the method very clearly, which makes it really difficult for the reader to understand the individual steps of the method. Moreover, the structures in Figure 1 workflow are not clearly labeled, so that it is confusing which part represents which protein/molecule.

- Except for the general workflow shown in Figure 1, a more detailed workflow showing the experimental steps, such as the sample fractions with the following steps could be added so that the design of the method is clearer. Also the style of the workflows including Figure 1, Figure 2A, and Figure 3A are different. It would be helpful to make them the same style and make the Figure 2A as a zoom in or more detailed illustration on part of Figure 1.

- A summary of proteomics results of time course labeling after all enrichment steps, including the total number of identified proteins at different conditions and control would be helpful for having an overview impression on the proteomics results

- In Figure 2B, the WB for PDIA4 in the Biotin PD elution is missing. Why was the PDIA4 interaction missing for the time course analysis, but the interaction was captured in the initial test for Wt Tg (Figure 1D). Additionally, in this panel the Rhodamine Probe Gel shows inconsistencies at the time points 1.5 - 3h. Does this mean that the labeling did not work well for these conditions? As we would expect a consistent Rhodamine Probe signal at every time point.
 - In Figure 2, why was there no WB results for the A2234D? In Figure 2D and 2E, at which time point are the changes significant compared to WT?
 - All figure legends should indicate how many biological replicates were performed for each experiment represented in the figure.
 - The heatmaps shown in Figure 3, Figure 3 - Figure Supplement 3, and Figure 7 are in the current form incomprehensible. The heatmaps depict the relative enrichment vs the control sample, which was scaled between 1 and -1. The color coding with 5 different colors from 1 to -1 is very confusing and should be changed to just two colors, one for positive and one for negative relative enrichment. I would also suggest changing the visualization of the heatmap showing the wt and mutants side by side, instead of stacked on top of each other for each individual protein.
 - The data analysis method for generating relative enrichment shown in the heatmap is not explained. This should be described in the method section for a better understanding of the data analysis.
- There are no legends of flowcharts in Figure 2A and Figure 3A and it is difficult to understand which are the key components in the complex and what are the differences among different periods of labeling.
- Why did only one of the VCP inhibitors (ML-240) exhibit a phenotype in Tg abundance and secretion, but not the other VCP inhibitors?

2. Significance:

Significance (Required)

The authors describe a novel and elegant method to map time resolved protein interactions of newly synthesized proteins, which allows monitoring of proteins regulating protein quality control.

Authors describe it as a general method, however, they only demonstrate the applicability to one protein and do not systematically evaluate the quantitative nature of their approach by determining quantitative reproducibility, which would be necessary to be able to claim that this is a method with broad applicability.

Given my expertise in quantitative proteomics, I can mainly comment on the technological aspects of the proteomics part of the manuscript, but do not feel qualified to evaluate the significance of this study in terms of novel biology.

Nevertheless, it feels that there is a stronger emphasis on the biology in the current form of the manuscript which will raise interest of scientists with a focus on protein quality control and Tg biology.

3. How much time do you estimate the authors will need to complete the suggested revisions:

Estimated time to Complete Revisions (Required)

(Decision Recommendation)

Between 3 and 6 months

Yes

Review #3

1. Evidence, reproducibility and clarity:

Evidence, reproducibility and clarity (Required)

Provide a short summary of the findings and key conclusions (including methodology and model system(s) where appropriate). Please place your comments about significance in section 2.

In this manuscript, the authors describe their efforts to develop a methodology for determining time-resolved protein-protein interactions using quantitative mass spectrometry. With TRIP (time-resolved interactome profiling), they combine a

pulsed bio-orthogonal unnatural amino acid labelling (homopropargylglycine, Hpg), CuAAC conjugation and biotin-streptavidin pulldowns to enrich at different timepoints and time-resolve by combining TMT labelling and LC-MS/MS (Figure 1). This technique is then applied to the maturation of the secreted WT and mutant thyroglobulin (Tg-WT, Tg-C1264R, Tg-A2234D) expressed in HEK293 and rat thyroid cells (FRT) and linked to hyperthyroidism. There, they identify a collection of ER resident proteins involved in protein folding/processing (e.g. chaperones, redox, glycans, hydroxylation) as well as degradation (e.g. autophagy, ERAD/proteasomes) (Fig. 2). Here the authors effectively use pulse-labelled form of TRIPs to highlight the different interactions formed with Tg-WT vs. Tg-mutants during biogenesis and secretion (or retention). The analysis found ~200 new interactions compared to previous studies along with about 40% of those identified previously. Differences in interactions were observed for mutants, which shown extended interaction with chaperones and redox processing pathways. While many interactions appeared as might be expected, the identification of membrane protein processing elements (e.g. EMC, PAT) was puzzling and raised some questions about the specificity within the protocol. Mutants enriched for CANX CALR and UGGT, suggesting prolonged association with glyco-processing factors. Interaction of C1264R with the ER-phagy factors CCPG1 and RTN3 was greater than WT. The authors note that their interaction correlated with that of EMC1 & 4, but it is not clear why that might be.

With interactors in hand, the authors complemented the TRIP protocol with siRNA KD of identified factors, to investigate any changes to secreted vs intracellular Tg upon loss. KD of NAPA (α -SNAP) and LMAN1 increased WT lysate (intracellular) Tg but not mutants. NAPA also reduced Tg-WT secretion. In contrast, KD of NAPA increased A2234D secretion while LEPRE1 increased C1264R (but not A2234D or WT), suggesting mutants have differential processing paths and requirements. KD of VCP increased secretion of both mutants. Some ER-phagy receptors were found among interactors (e.g. RTN3 in Tg-C1264R only) but often their KD had no impact on secretion (CCPG1, SEC62, FAM134B). NAMA observations were recapitulated in thyroid derived cell line (FRT). KD of TEX264 and VCP increased Tg-C1264 secretion while RTN3 KD in FRTs decreased Tg-C1264 secretion. This was in contrast to data from HEK293s for reasons that are not clear. Co-IP with TEX264 enriched for all Tg forms but more so for C1264R and A2234D - motivating the authors to propose selective targeting of Tg to TEX264 and the consideration of ER-phagy as a "major" degradative pathway during Tg processing.

Given the observations with siRNAs to VCP, the authors next use a selection of VCP inhibitors to ask whether secretion can be rescued upon pharmacological impairment of the AAA ATPase. They observed that ML-240, but interestingly not the more conventionally used CB-5083 or NMS-873, increased secretion of Tg-C1264R but not

lysate. Inhibitors increased lysate but decreased the secreted fraction for Tg-WT (Fig 7). Finally, the authors used TRIP again in ML-240 treated Tg-C1264R expressing cells to look for changes to interactome with treatment - observed decreases to glycan and chaperone interactions, CANX and UGGT1, decreased interaction with DNAJB11 and C10, like that of WT. There was no apparent change to the UPR, although activation was not directly measured.

****Major comments:****

- Are the key conclusions convincing?

The TRIP methodology appears to be quite robust and should be a powerful strategy for this field and others going forward. The drawback will be the length of pulse required will limit the number/type of proteins to be monitored to ones with longer $t_{1/2}$'s. There were interesting interactions found with Tg and the mutants linked to hyperthyroidism, but cut and dry differences did not appear as obvious, even though strong "trends" appear to be present. The path from identifying interactors in a time-resolved manner to then following them up with targeted KD does provides some clarity, which is important.

- Should the authors qualify some of their claims as preliminary or speculative, or remove them altogether?

The data regarding VCP silencing and pharmacological impairment appear clear but leave some questions outstanding in this reviewer's opinion. The lack of effect with the 2 highly selective inhibitors suggests that the underlying mechanism for switching fate of intracellularly retained Tg-C1264R towards secreted forms is not at all clear. ML-240 is an early derivative of DBeQ and reportedly impairs both ERAD and autophagic pathways, similarly to DBeQ. The differences between the VCP inhibitors' mechanism of action were not discussed, but perhaps should be elaborated upon, particularly in the matter of how ERAD and ER-phagy pathways might be being differentially affected. At the risk of asking for too many additional experiments, this reviewer would just prefer to see this fleshed out in a bit more detail.

- Would additional experiments be essential to support the claims of the paper?
Request additional experiments only where necessary for the paper as it is, and do not ask authors to open new lines of experimentation.

Q1. The degree (if any) of Tg-C1264 aggregation during and/or detergent solubility do not appear to have been considered as a potential source of the increase in released secreted material (Figure 4, 5). Do Tg mutants partition into RIPA-insoluble fractions at all? That is to say.. is the total population of synthesized Tg being considered? A

full accounting? Could the authors address this and if biochemical extraction data (via urea or high SDS) is available, include it to answer this concern.

Q2. Along the same lines, what does Tg-WT and mutant expression look like by microscopy? Is Tg-WT uniformly distributed while Tg-mutants appear in puncta... more aggregated - perhaps reflecting the increased engagement of chaperones and redox machinery? Changes in the pattern of Tg-C1264R mutant (e.g. w/ VCP KD or inhibition) would add additional support for the authors interpretation of improved secretion. If this data is at hand, including it might be worth consideration.

Q3. Does the level of Tg mutant expression in the FRT clones impact the profiles obtained by TRIP? (Figure 3). This is a question of gauging the relative saturation of QC machinery and how that might impact profiles from TRIP. Were clones expressing at different levels tested? Perhaps a brief discussion of this.

Q4. For Figure 3, the hour-long labelling period seems a bit long, compared with 3 hr of chase. Perhaps this reviewer missed this but how long does Tg take to mature and/or mutants to misfold and degrade? Is there any possibility to shorten this so that the profiles of labelled Tg could be more synchronized? If not, perhaps this could just be discussed.

Q5. It is curious that only ML-240 and not other well characterized inhibitors of VCP/p97, has an effect, as both are used far more often than ML-240. The authors do not really address this in detail but does it suggest that the ML-240 effect on VCP/p97 could be affecting different pathways, given the nature of this compound. Is this compound acting on Tg-C1264R maturation at the level of translation or post-translationally? If the latter, through what means?

Q6. Continuing from Q5.. At what point and where is VCP/p97 able to affect mutant Tg processing? In line 317, the authors seem to correlate increased VCP association with mutants to their increased secretion. It is not clear how this would result, as engagement with VCP would be in a compartment different to that which supports trafficking and secretion. Could the authors expand on how this might come about. This is also relevant to the ML-240 data in Figure 7. Moreover, VCP is associated with ERAD (as is HerpUD1) rather than ER-phagy and at least in the siRNA raw data, there are also effects from Derlin3 and FAF2 KDs.. both ERAD factors. Some clarity here would be appreciated.

Q7. There does not appear to be a direct demonstration of Tg-C1264R turnover by ER-phagy (via TEX264). Given the inconsistency with it not being detected by TRIP, while another receptor RTN3 was, but has not impact on Tg-C1264R secretion,

perhaps including that data would go some way to demonstrating a fate of ER-phagy (at least partly) for this mutant.

Q9. The authors provide data that the UPR was not induced by ML-240 at 3hrs (10 μ M) (Figure 7, supplemental 1). This is in stark contrast to the results of Chou et al (2013) which the authors reference, reporting that ML-240 induced ATF4 and CHOP by 2 hrs at concentrations lower than used here (albeit a different cell type). While not exclusively UPR, could the authors address the potential activation of the integrated stress response (eIF2a phosphorylation, ATF4 and CHOP) in the FRT cells due to ML-240 treatment? If present, is there some link that could this provide an explanation for increased Tg-C1264R secretion?

- Are the suggested experiments realistic in terms of time and resources? It would help if you could add an estimated cost and time investment for substantial experiments.

Any of the suggested experiments above all use reagents reported in the manuscript and so would presumably incur minimal cost and hopefully time. This reviewer is sympathetic to time and financial constraints and so discussion of the issue could suffice.

- Are the data and the methods presented in such a way that they can be reproduced?

Yes. The methodology is explained in detail.

- Are the experiments adequately replicated and statistical analysis adequate?

Yes. Relevant information is either in the figure legends or is provided in the source data.

****Minor comments:****

- Specific experimental issues that are easily addressable.

- Are prior studies referenced appropriately?

The references are generally appropriate, with a few exceptions of more general references used

- Are the text and figures clear and accurate?

The text is clearly written, and the figures are clear.

- Do you have suggestions that would help the authors improve the presentation of their data and conclusions?

A summary figure comparing the changing profiles of WT and C1264R and the factors implicated for them could be helpful.

Perhaps include common nomenclature for proteins as well (e.g. HSP5A - BiP, HSP90B1 - Grp94, etc..)

Line 317 - our is misspelled

Figure 4 - Supplemental Figure 1 - Legend has text referring to panels J and K, but Figure only goes up to F.

2. Significance:

Significance (Required)

- Describe the nature and significance of the advance (e.g. conceptual, technical, clinical) for the field.
- Place the work in the context of the existing literature (provide references, where appropriate).

Protein-protein interactions are often used to illustrate complexes and functionality, but these provide only snapshots, rather than "movies". There are many datasets out there exploring P-P interactions, but most if not all lack any temporal resolution for the interactions they report. The TRIP method described approaches this from the dynamic perspective - identifying the transient interactions formed by folding nascent chains with proteins that aid in their maturation and trafficking, or degradation. This represents an important technical advance in our ability to dynamically monitor protein interactions. The use of Tg mutants is valuable and perhaps this will lead to new perspectives on how to rescue it or other pathophysiological mutants with loss of function phenotypes.

- State what audience might be interested in and influenced by the reported findings.

This work should appeal to a broad audience within cell biology, particularly as the TRIP technique is attempting to address a fundamental question - what interactions form during the biogenesis/lifetime of a protein. Moreover, the effort to try to understand the different interactions formed with pathologically relevant mutant proteins as a strategy to try to rescue functionality, is a valuable exercise of this approach.

- Define your field of expertise with a few keywords to help the authors contextualize your point of view. Indicate if there are any parts of the paper that you do not have sufficient expertise to evaluate.

ER quality control

3. How much time do you estimate the authors will need to complete the suggested revisions:

Estimated time to Complete Revisions (Required)

(Decision Recommendation)

Between 1 and 3 months

Yes

Review #4

1. Evidence, reproducibility and clarity:

Evidence, reproducibility and clarity (Required)

****Summary****

In this manuscript, Wright et al. developed an approach (termed TRIP) that allowed to map the temporal changes in the interaction landscape of a newly synthesized protein of interest. Using their TRIP approach, the authors found that the extensive interactions of thyroglobulin (Tg) with the proteostasis network (PN) during its passage through the secretory pathway were profoundly altered in response to disease-causing mutations (e.g. C1264R). The authors cross-validated their findings with a focus RNAi screen monitoring the cellular and secreted abundance of Tg variants upon deletion of PN components. In subsequent experiments the authors focused on two hits, VCP and TEX264, for which they confirmed their inhibitory effect on the

secretion of Tg C1264R. Importantly, the authors found that TEX264 increasingly interacts with the Tg mutant and that pharmacological inhibition of VCP yielded the same phenotype than depletion of VCP. Overall, Wright and colleagues established an elegant method to map protein interaction in a time-resolved manner and demonstrated its value by the analysis of disease-related Tg mutants. Hence, this work has the potential to serve as a rich resource for Tg-related research and as a powerful new tool to examine protein interactions. However, several concerns remain.

****Major points****

1. Overall, the TRIP workflow is quite difficult to understand at a first glance - even for a reader with a background in proteomics, biochemistry and cell biology. The authors may want to improve the description of the TRIP methodology and explain in more detail what the individual components and steps are good for. Along the same line, from the main text and the figure legend it was not clear that Tg was actually Flag-tagged. However, without this information it is difficult to follow the workflow. While Figure 1A is certainly helpful, the bulky graphics are deflecting the reader's attention. A more schematic version might be more informative.
2. To what extent do the difference in protein abundance between Tg WT and Tg C1264R contribute to the increase binding of their interactors (e.g., HSP5 and PDIA4). The authors should perform a TRIP coupled immunoblot analysis where WT and Mutant are loaded side-by-side on the SDS-PAGE.
3. While the RNAi screen was done with pooled siRNA, it is not clear what was used for the RNAi validation experiments shown in Figure 5. This should be done by individual siRNA and not the same pooled reagents as used for the screen.
4. In Figure 5A it is not clear which band was used to quantify the effect of NAPA reduction. Also, this analysis lacks normalization to an unrelated protein or loading control. Moreover, the authors should also examine the effect of the siRNA targets shown in Figure 5C for Tg WT and not only the mutant.
5. The authors should also test for the binding of RTN3 to Tg WT and mutant - in particular in comparison to TEX264. This would be important in the context that only RTN3 but not TEX264 was detected in the TRIP approach. Do the authors also detect VCP and LC3B in their pulldowns?
6. The effect of TEX264 depletion on Tg secretion should be confirmed by TEX263 KO experiments. Do the authors observe a similar increase in secreted Tg C1264R in BafA1- or SAR405-treated cells? Moreover, the authors should show that Tg C1264R is actually delivered to lysosomes using biochemical assays such as LysoIP or colocalization experiments.
7. Figure 7A and 7C lack loading controls. The quantification shown in Figure 7B and 7D should be normalized to this control. Since VCP activity is often coupled to the of the proteasome, the authors should check whether blocking the proteasome yields a

similar effect than ML-240.

8. With regard to Figure 7 - Figure supplement 1: The authors should monitor the effect of ML-240 on Tg secretion such that WT and C1264R mutants are directly compared (side-by-side on the same immunoblot). Otherwise, it is difficult to claim that ML-240 rescues the secretion of the mutant.

9. How did ML-240 affect the ER-phagy components (in particular RTN3) in the TRIP analysis of Tg C1264R (Figure 7G-L)?

****Minor points****

10. The candidate labeling in Figure 3 - Figure supplement 2 and 3 is too small and unreadable. The authors should provide a higher resolution of these figures or increase the font.

2. Significance:

Significance (Required)

Please see above

3. How much time do you estimate the authors will need to complete the suggested revisions:

Estimated time to Complete Revisions (Required)

(Decision Recommendation)

Between 3 and 6 months

No

Reviewer Responses

Reviewer #1 (Evidence, reproducibility and clarity (Required)):

The authors report a mass spectrometry (MS)-based interactomics technique, time-resolved interactome profiling (TRIP), which allows for tracking temporal changes in the interactome of protein of interest. To show that TRIP can successfully deconvolute interactomes over time, they pulsed thyroid cells with homopropargylglycine (Hpg), immunoprecipitated the Hpg incorporated thyroglobulin (Tg) and its interacting proteins at different time points, and subjected the samples to tandem mass tag (TMT)-based quantitative MS analysis. The MS results show that WT and variant Tg proteins indeed associate with different proteostasis network factors in a differential manner over the course of time. In addition, they utilized an siRNA-based luciferase fusion assay to evaluate whether silencing each proteostasis network component changes the levels of Tg in both lysate and media. From the combination of the TRIP and siRNA-based assays, they found many hits, including hits implicated in protein degradation, VCP and TEX264, which they validated with multiple experiments.

I am overall quite positive and think this is an important study. But there are some meaningful points to consider.

Our Response: We thank Reviewer #1 for their positive outlook on our manuscript and their constructive feedback. We have addressed the comments below.

Significant comments:

Reviewer #1, Comment #1: *Only two replicates of the main data (the TRIP-MS experiments) for this paper is problematic. Especially since the manuscript is supposed to be demonstrating and validating the new technique. Consistent with this concern, the relative enrichment profiles for some of the results were surprising. For instance, interaction with CCDC47 was tapering off but then at 3 h it suddenly reaches the maximum level of engagement. Is this a real finding or the variability in the method? Impossible to tell with two replicates. Presenting heat maps based on biological duplicates is also very problematic. It masks the error, which is large as can be seen in some of the panels showing individual proteins. In my view, triplicates and a clear understanding of the error in the technique should be required.*

Our Response: The TRIP datasets for WT Tg contains 5 biological replicates, while the A2234D and C1264R Tg contains 6 biological replicates. Two replicates are typically included in a TMTpro 16plex mass spectrometry run, and each analysis consists of 3 MS runs. We apologize that the number of replicates and layout of the MS runs was not clearly explained. Data for individual replicates is found in **Dataset EV1**, **Dataset EV3**, and a newly added **Table EV3** delineates the sample layout across the TMT channels and MS runs. We clarified the text as follows:

“Subsequently, two sets of TRIP time course samples (0, 0.5, 1, 1.5, 2, and 3 hr) could be pooled using the 16plex TMTpro and analyzed by LC-MS/MS (**Fig 2A**). In total, 5 biological replicates were analyzed for WT and 6 biological replicates were analyzed for A2234D and C1264R, respectively (**Table EV3**).”

Reviewer #1, Comment #2: *The same concern arises for the high-throughput siRNA screen, which was performed only in duplicate for WT and A2234D.*

Our Response: While the initial screen was performed in duplicate for WT and A2234D, which is common for larger screens due to resource constraints, we would like to direct the reviewer to the fact that we followed up on observed hits using thyroid cell lines with many more replicates. Furthermore, most hits came from the C1264R Tg variant, which had three replicates in the initial screen. Hits were also extensively followed-up.

Reviewer #1, Comment #3: *There are issues with some of the immunoprecipitation experiments: In Figure 1C, a negative control for FLAG IP is missing.*

-In Figure 2B, I am curious why the band (Hpg -, chase time 0 h) is so faint for the first WB (IB for FLAG) - is Hpg treatment indeed leading to much more Tg present at 0 h? If so, that is a concern.

-Also, a negative control must be included (either plain cells or cells expressing fluorescent protein or a different epitope-tagged WT Tg).

-In this same figure, I am puzzled why the bands for 1.5-3 timepoints in Biotin PD elution, probed for Rhodamine, are very faint especially considering that in Figure 1D, the corresponding bands, which are 4 h after the pulse, look fine. It seems like the IP failed here?

Our Response: In Fig 2B, we have updated this figure with higher-quality images that are more representative of the results found when performing this experiment. Furthermore, to address the missing negative controls in Fig. 1C, we have added a separate figure (**Fig EV2**) where (-) FLAG-tagged Tg is included in this panel. We updated the text as follows:

“Furthermore, the C-terminal FLAG-tag and Hpg labeling are necessary for this two-stage enrichment strategy, and DSP crosslinking is necessary to capture these interactions after stringent wash steps (**Fig 1D**, **Fig EV2**).”

Regarding the Biotin PD rhodamine/TAMRA signal in **Fig 2B**: The blots in this figure panel represent the time-resolved Tg fractions from cell lysate, corresponding only to intracellular thyroglobulin. The decrease in band intensity for 1.5-3 hr time points is expected due to continued secretion and/or degradation dynamics taking place that decrease the intracellular population of labeled thyroglobulin that is able to be captured. For comparison, please note the C1264R panel (**Fig 2C**), where the rhodamine/TAMRA signal in the Biotin PD elutions is more stable compared to WT, indicating the cellular retention of C1264R while WT Tg is efficiently secreted and the signal is lost more rapidly. **Fig 1D** contains samples derived from a 4 hr Hpg pulse (without chase), explaining why the overall fluorescent Tg signal is more intense.

Suggestion to consider:

Reviewer #1, Comment #4: *This manuscript, supported by the title and abstract, mainly focuses on the presentation of the development and application of TRIP, which is highly significant. The story becomes less coherent and harder to follow as significant amounts of text/figures are dedicated to siRNA-based high throughput screening and follow-up. In addition, although the discovery of TEX264 as one of the hits is very interesting and exciting, TEX264 apparently was not a hit in the TRIP experiment and is pretty distracting from the main point of the paper highlighted in the abstract and title, therefore. The siRNA-based assay and follow-up studies could be a separate scientific story of their own. Especially considering my concerns on the number of replicates for both the TRIP and siRNA-based assay, it could be beneficial to actually split the manuscript into two and conduct more replicates of the -omic work, which should corroborate the exciting discoveries the authors have made.*

Our Response: We have edited the manuscript to hopefully provide a more cohesive presentation of all data, findings, and conclusions within the paper. Given the generally positive outlook on the manuscript from other reviewers and our responses to significant comments from Reviewer #1 we opted to keep the manuscript as a single piece and address all reviewer comments.

Minor comments:

Reviewer #1, Comment #5: *Throughout the manuscript, the authors have not defined what FT is; presumably it means FLAG tag.*

Our Response: Reviewer #1 is correct in FT corresponding to FLAG tag. We have now edited the manuscript text to clarify this as follows:

“Thyroglobulin was chosen as model secretory client protein, and we generated isogenic Fischer rat thyroid cells (FRT) cells that stably expressed FLAG-tagged Tg (Tg-FT), including WT or mutant variants (A2234D and C1264R).”

Reviewer #1, Comment #6: *The authors might discuss their rationale for choosing 0-3 hrs for their TRIP studies. That includes any relevant information about the half-life of WT versus variant Tg, whether the Hpg pulse time is short enough to avoid missing key features of the temporal interactome, and discussion of what would happen if the TRIP were performed at prolonged time points (e.g. 6-10 h).*

Our Response: Apologies that we omitted this important point, which is indeed related to the secretion and degradation half-life. We edited the manuscript text to discuss the rationale for 0-3 hr, length of the Hpg pulse and the impact on capturing interactions, and performing TRIP at prolonged time points as follows:

“Our previous study indicated that ~70% of WT Tg-FT was secreted after 4 hours, while approximately 50% of A2234D and 15% of C1264R was degraded after the same time period (Wright *et al*, 2021). Therefore, we reasoned that a 3-hr chase period would be a enough time to capture the majority of Tg interactions throughout processing, secretion, cellular retention, and degradation, while still being able to capture an appreciable amount of sample for analysis.”

We explain the labeling timeline and limitations further in the discussion:

“To address this, we utilized a labeling time of 1 hr which allows us to generate a large enough labeled population of Tg-FT for TRIP analysis, but some early interactions are likely missed within the TRIP workflow. In the case of mutant Tg, performing the TRIP analysis for much longer chase periods (6-8 hrs) may provide insightful details to the iterative binding process of PN components that is thought to facilitate protein retention within the secretory pathway.”

Reviewer #1, Comment #7: *Lines 68-69: the two citations should probably come one sentence earlier (at least Coscia et al 2020 is a structure paper).*

Our Response: We agree. We have edited the manuscript as follows to correct this:

“In earlier work, we mapped the interactome of the secreted thyroid prohormone thyroglobulin (Tg) comparing the WT protein to secretion-defective mutations implicated in congenital hypothyroidism (CH) (Wright *et al*, 2021). Tg is a heavily post-translationally modified, 330 kDa prohormone that is necessary to produce

triiodothyronine (T3) and thyroxine (T4) thyroid specific hormones (Citterio *et al*, 2019; Coscia *et al*, 2020). Tg biogenesis relies extensively on distinct interactions with the PN to facilitate folding and eventual secretion.”

Reviewer #1, Comment #8: *Line 91: "(Figure 1A)" should follow the sentence "To develop the time-resolved..." to help readers better understand the system.*

Our Response: We agree. We have edited the manuscript to add the Fig 1A reference. Furthermore, we redesigned the schematic in Fig 1A to better explain the experimental system. (see also **Reviewer #2, comment 10**)

“To develop the time-resolved interactome profiling method, we envisioned a two-stage enrichment strategy utilizing epitope-tagged immunoprecipitation coupled with pulsed biorthogonal unnatural amino acid labeling and functionalization (**Fig 1A**). Cells can be pulse labeled with homopropargylglycine (Hpg) to synchronize newly synthesized populations of protein. After pulsed labeling with Hpg, samples can then be collected across time points throughout a chase period (**Fig 1A**, Box 1) (Kiick *et al*, 2001; Beatty *et al*, 2006). The Hpg alkyne incorporated into the newly synthesized population of protein can be conjugated to biotin using copper-catalyzed alkyne-azide cycloaddition (CuAAC) (**Fig 1A**, Box 2). Subsequently, the first stage of the enrichment strategy can take place where the client protein of interest is globally captured and enriched using epitope-tagged immunoprecipitation, followed by elution (**Fig 1A**, Box 3).”

Reviewer #1, Comment #9: *Line 101: Fisher should be Fischer*

Our Response: Thank you. We have edited the manuscript text to correct this.

Reviewer #1, Comment #10: *Line 131: Should be 1.5 hrs instead of 2 hrs.*

Our Response: We edited this point (see below in **comment #11**)

Reviewer #1, Comment #11: *Lines 135-136: I do not agree with the claim that HSPA5 profile looked similar for MS and WB. I do not see a peak for HSPA5 at 2 hrs in Figure 2D.*

Our Response: We replaced the mass spectrometry quantification in Fig 2D, E with the scaled, relative enrichments. This provides a more meaningful comparison, as all interactions are scaled in the same way. Unfortunately, it is still difficult to directly compare the Western blot results in Fig. 2B-C to the mass spectrometry quantifications in Fig 2D-E because the WB intensities are not normalized to the Tg bait protein amounts, which is changing over time. At 2-3hrs time points, little WT Tg is pulled down as most of it is secreted. Therefore, the HSPA5 interactions are no longer detectable by Western blot. On the other hand, MS is much more sensitive to capture the interactions. We modified the text as follows:

“For C1264R, interactions with HSPA5 were highly abundant at the 0 hr time point and remained mostly steady throughout the first 1.5 hours (**Fig 2C**). A similar temporal profile was also observed for HSP90B1. Additionally, interactions with PDIA4 were detectable for C1264R and were found to gradually increase throughout the first 1.5 hr of the chase period, before rapidly declining (**Fig 2C**). We noticed similar temporal profiles for PDIA4 and HSPA5 to our western blot analysis, when measured via TMTpro LC-MS/MS as further outlined below (**Fig 2D-E**). In particular, the HSPA5 WT Tg interaction declined within the first hours, yet for C1264R Tg, the HSPA5 interactions remained mostly steady over the 3-hour chase period. (**Fig 2E**).”

Reviewer #1, Comment #12: *Line 186: The cited paper Shurtleff et al 2018 is missing in the reference list.*

Our Response: Thank you. We have corrected this in the citation management system and it is now available in the reference list.

Reviewer #1, Comment #13: *Line 188: I disagree with the authors' claim here because, at least for CCDC47, interactions with C1264R seem to come back at the 3 hr time point.*

Our Response: We have removed the discussion of EMC and PAT complex components from the text. The implications of these interactions for Tg biogenesis remain unclear and were therefore a distraction from the discussion of other core proteostasis network components pertinent to Tg processing. Nonetheless, the full dataset – including these interactions – remains available to readers in **Appendix Fig S1** for further perusal.

Reviewer #1, Comment #14: *Line 203: I am not sure if P4HA1 can be included in the examples for showing distinct patterns for mutants compared to the WT according to their data in Figure 3H.*

Our Response: We agree. We have edited the text to remove the discussion of prolyl hydroxylation and isomerization family members and elected to discuss the new clustering analysis and the robustness of the TRIP method in more detail. The full TRIP data is nonetheless available to interested readers in **Appendix Fig S1**.

Reviewer #1, Comment #15: *Line 216: The authors should add citations about the functions of STT3A and STT3B proteins.*

Our Response: We've edited the manuscript text to include a reference to the primary literature for STT3A and STT3B functions, as follows:

“Previously, we showed that A2234D and C1264R differ in interactions with N glycosylation components, particularly the oligosaccharyltransferase (OST) complex. Efficient A2234D degradation required both STT3A and STT3B isoforms of the OST, which mediate co-translational or post-translational N-glycosylation, respectively (Kelleher *et al*, 2003; Cherepanova & Gilmore, 2016).”

Reviewer #1, Comment #16: *Lines 248-251, "We found that interactions with these components...": this sentence should refer to Figure 3 - Figure Supplement 3 instead of Figure 3L and S4.*

Our Response: Thank you. This section of the manuscript was significantly rewritten and the figure references updated.

Reviewer #1, Comment #17: *Lines 258-260, "Another striking observation was that the temporal profile of EMC interactions for C1264R correlated with RTN3, PGRMC1, CTSB, and CTSD interactions.": Please provide more evidence to support the potential correlation between different interaction profiles. Or the authors should move this sentence to the discussion section as it sounds speculative. This highlights the issue of only having duplicates, as well.*

Our Response: We agree that this point was highly speculative and we removed discussion of the EMC interactions.

To further investigate the correlation of interaction profiles across the dataset, we performed unbiased k-means clustering. This led to the identification of 7 and 6 unique clusters of interactors for WT and C1264R Tg-FT, respectively. These data are represented in **Fig 3F** and **Fig EV5**. Unique clusters

highlight similar temporal interaction profiles for Tg-FT interactors, and provide a quantitative representation of correlative interactions that take place during Tg-FT processing.

“To assess temporal interaction changes in an unbiased fashion and identify protein groups exhibiting comparative behavior, we carried out k-means clustering of the temporal profiles for WT and C1264R. This analysis revealed a large divergence in the interaction profiles. For WT Tg, only one cluster exhibited steadily decreasing interactions (cluster 4), while others increased with time, or showed peaks at intermediate times (**Fig 3F, Fig EV5A**). On the other hand, C1264R largely exhibited clusters with decreasing interactions over time (**Fig 3F, Fig EV5B**). Cluster 2 for WT with bimodal interactions at early and late time points contains many Hsp70/90 chaperoning components. For C1264R Tg, many Hsp70/90 chaperoning components and disulfide/redox-processing components are instead part of cluster 2', which exhibited an initial rise in interactions strength before plateauing (**Fig 3F, Fig EV5A,B**). This divergent temporal engagement between WT Tg and the destabilized C1264R mutant is aligned with the patterns observed in the manual grouping (**Fig 3B,C**), highlighting that the unbiased temporal clustering can reveal broader patterns in the reorganization of the proteostasis dynamics.”

One of the clusters of the C1264R Tg interactions contained autophagy interactors along with glycosylation components. We therefore postulate that this could point to a coordination of these processes. We discuss this new point in the updated manuscript:

“In the k-means clustered profiles, autophagy interactions largely group together in the same cluster, showing stronger interactions at earlier time points. In the same cluster are glycosylation components (UGGT1 and STT3B, MLEC), further supporting a possible coordination for C1264R Tg between lectin-dependent protein quality control and targeting to autophagy (**Fig EV5B,C**).”

Reviewer #1, Comment #18: *Line 340: As written, should cite more than one paper*

Our Response: Thank you. We reworded the manuscript to correct this, as follows:

“The discovery of several protein degradation components as hits for rescuing mutant Tg secretion may suggest that the blockage of degradation pathways can broadly rescue the secretion of A2234D and C1264R mutant Tg, a phenomenon similarly found for destabilized CFTR implicated in the protein folding disease cystic fibrosis (Vij *et al*, 2006; Pankow *et al*, 2015; McDonald *et al*, 2022).”

Reviewer #1, Comment #19: *Line 371: Should be Figure 4 - figure supplement 2*

Our Response: We edited the manuscript to correct this error.

Reviewer #1, Comment #20: *Line 1231: "Zhang et al 2018" needs to be removed*

Our Response: We have removed this citation.

Reviewer #1, Comment #21: *Line 1286: FRTR should be FRT*

Our Response: Thank you. We have corrected this within the text.

Reviewer #1, Comment #22: *Figure 3E: Color used to highlight the three proteins (CCDC47, EMC1, EMC4) should match the color used in Figure 3 - Figure Supplement 3*

Our Response: We have edited Figure 3 to remove the section related to membrane protein biogenesis. This data is still available in **Appendix Fig S1** with consistent color coding.

Reviewer #1, Comment #23: *Figure 4A: The bottom figure where lysate signal is inversely proportional to time is misleading because the authors are assessing steady-state level of proteins in this assay.*

Our Response: We agree. We updated the schematic in **Fig 4A** to better explain the workflow and differentiate the steady-state protein level being measured within the lysate.

Reviewer #1, Comment #24: *Figure 4 - Figure Supplement 1 caption: in (C), (F) should be (B). (K) should be (G) and I am not sure what the authors mean when they refer to (J) in caption of (G).*

Our Response: We have corrected this lettering mistake to match the figure properly. Please note that this figure is now **Fig EV6**, and it includes some new and reorganized panels.

Reviewer #1, Comment #25: *Figure 5 caption for (C and D): Need to specify the time that the samples were collected (8 hrs), as it seems different from A and B according to the main text.*

Our Response: We have specified the collection time within the caption for these data in **Fig 5C** and **5D**.

Reviewer #1, Comment #26: *Figure 5 - Figure Supplement 1: Data for HERPUD1 and P3H1 should be included.*

Our Response: We have now included data to confirm the knockdown for *HERPUD1* and *LEPRE1* (*P3H1*) in **Fig EV7F-G**.

Reviewer #1, Comment #27: *Figure 5 - Figure Supplement 2B: Please mention in the caption how degradation is defined.*

Our Response: We have updated the **Fig EV7H** caption to include how “degradation” is defined within these experiments:

“% Degradation is defined as $[1 - (Tg_t^{lysate} + Tg_t^{media})] \times 100$. Where Tg_t^{lysate} is the fraction of Tg-FT detected in the lysate at a given timepoint n , and Tg_t^{media} is the fraction of Tg-FT detected in the media at a given timepoint n .”

Reviewer #1 (Significance (Required)):

Reviewer #1, Comment #28: *This manuscript is **highly significant** because the authors (1) designed and validated a new methodology for time-resolved interactomics study, (2) presented the dynamic changes in Tg interactome for WT and variants, and (3) discovered how proteins implicated in degradation pathways (e.g. VCP, TEX264, RTN3) can change the secretion profile of WT and mutant Tg proteins. With TRIP, the authors demonstrated that they could obtain valuable data that were previously not captured from steady-state interactomics studies (Wright et al. 2021; Figure 3M and Figure 3 - Figure supplement 4D-4I). Furthermore, the authors treated cells with VCP inhibitors and*

performed both 35S pulse-chase analyses and TRIP. These experiments provide valuable information to the field by (1) presenting a new method to rescue Tg secretion defect, and (2) demonstrating a broader applicability of TRIP. **If the major comments above can be addressed I believe this is a tremendous contribution to the field.**

Our Response: We thank Reviewer #1 for their review comments and praise for the work presented within this manuscript.

Reviewer #2 (Evidence, reproducibility and clarity (Required)):

Reviewer #2: *In the manuscript 'Time-Resolved Interactome Profiling Deconvolutes Secretory Protein Quality Control Dynamics' Wright et al. developed an approach for time-resolved protein protein interaction mapping relying on pulsed unnatural amino acid incorporation, protein cross linking, sequential affinity purification, and quantitative mass spectrometry named time-resolved interactome profiling (TRIP). The authors applied the TRIP method to compare the interactions of the secreted thyroid prohormone thyroglobulin (Tg) comparing the WT protein to secretion-defective mutations implicated in congenital hypothyroidism. They further employed an RNA interference screening platform (1) to investigate if (1) interactors identified via TRIP are functionally relevant for Tg protein quality control and (2) to identify factors that can rescue mutant Tg secretion. The screen was initially performed in HEK293 cells, but selected hits with a phenotype in HEK cells were then followed up in Fisher rat thyroid cells. Further functional validation was performed by pharmacologic inhibition of VCP, a hit from the RNAi screen with an effect on Tg lysate abundance and Tg secretion. While the authors present a comprehensive study including identification of protein-protein interactions using proteomics followed up by an RNA interference screen for functional validation, major comments need to be addressed for both the proteomics as well as the functional genomics aspects of the study (see comments below).*

Our response: Thank you to reviewer 2 for their constructive feedback. We addressed all comments in detail below.

Major comments:

Reviewer #2, Comment #1: *The authors describe a new method for quantitative, temporal interaction mapping. The protocol involves two enrichment steps as well as several reactions including cross-linking of the samples as well as functionalization of the unnatural amino acids. Given all these steps, the authors should rigorously characterize the quantitative reproducibility of the experiment when performed in independent biological replicates. This is important because in the final quantitative MS experiment, the authors only use two biological replicates, which is too low especially for such an involved sample preparation procedure, which would expect to have a high variability between replicates. Given the low number of replicates and the unknown reproducibility of the quantification for this protocol, it is questionable at this point how reliable the quantification over the time course is.*

Our Response: We apologize that the number of replicates and robustness of the analysis was not entirely clear in our manuscript. We thank the reviewer for the feedback, as this is important point to clarify. We included several additional analyses to further explain the robustness and quantitative reproducibility of our results:

- 1) We clarified the **number of replicates** analyzed. For quantitative MS experiments five biological replicates were analyzed for WT, while six biological replicates were analyzed for A2234D and C1264R Tg-FT, respectively not two as mistakenly presumed by Reviewer #2. These data are available in **Dataset EV1** and **Table EV3**. There is only one place where two biological replicates are included, C1264R Tg-FT FRT cells treated with ML-240 treatment for TRIP analysis.

We have further clarified the number of biological replicates within the manuscript text as follows (see also **reviewer #1, comment 1**):

“Subsequently, two sets of TRIP time course samples (0, 0.5, 1, 1.5, 2, and 3 hr) could be pooled using the 16plex TMTpro and analyzed by LC-MS/MS (Fig 2A). In total, 5 biological replicates were analyzed for WT and 6 biological replicates were analyzed for A2234D and C1264R, respectively (**Table EV3**).”

- 2) We **displayed the reproducibility** of TRIP time profiles for several individual proteins in **Fig EV3** and in **Fig 3K** (VCP). We included shading to indicate the standard error of the mean (SEM) for the individual protein time courses to provide further assessment of the quantitative reproducibility. We updated the text as follows:

“To benchmark the TRIP methodology, we chose to monitor a set of well-validated Tg interactors and compare the time-resolved PN interactome changes to our previously published steady-state interactomics dataset (Wright *et al*, 2021). Previously, we found that CALR, CANX, ERP29 (PDIA9), ERP44, and P4HB interactions with mutants A2234D or C1264R Tg exhibited little to no change when compared to WT under steady state conditions (**Fig EV4A**). However, in our TRIP dataset we were able to uncover distinct temporal changes in engagement that were previously masked within the steady-state data. Our time-resolved data deconvolutes these aggregate measurements, revealing prolonged CALR, ERP29, and P4HB engagements for both A2234D and C1264R Tg mutants compared to WT (**Fig EV4B-F**). We found that these measurements for key interactors and PN pathways exhibited robust reproducibility, as exemplified by the standard error of the mean for the TRIP data (**Fig EV4B-I, Appendix Figure S1B**).”

- 3) For full transparency, we also include the SEM of all TRIP profiles in the heatmap in **Appendix Fig S1B**.
- 4) Furthermore, we included **25-75% quartile ranges for the pathway aggregated time courses** (**Fig 3B,C,J,K**) and the k-means hierarchical clustering analysis (**Fig 3F, Fig EV5**). Especially these clustering data allow for the visualization and analysis of temporal protein interactions that are correlated with one another, while the accompanying quartile ranges provide further context for the reproducibility of these measurements and cluster profiles (see **Reviewer #1, Comment 17** above for further explanation about the k-means clustering).

Reviewer #2, Comment #2: *Compared to the previous dataset published last year, the authors discover an overlap in interactors, but also a huge discrepancy, with 96 previously identified interactors not detected in the current study, but 198 additional interactors identified. How do the authors explain the big differences between these datasets?*

Our Response: We can only speculate here but this difference in overlapping interactors may stem from several different factors, including but not limited to cell line, instrumentation, LC-MS/MS methodology, and sample processing workflows. Our previous dataset was published using transiently transfected HEK293 cell lines expressed FLAG-tagged constructs of Tg. The HEK293 cell line makes for a robust cell line used throughout several biological investigations, but it is not representative of the native cellular environment in which Tg is expressed. Moreover, transiently transfected cells can lead to high protein expression that may not always represent what is found within the native cellular environment and proteome. Here, we used Fischer rat thyroid (FRT) cells engineered to stably express FLAG-tagged constructs of Tg. This cell line model should more accurately represent the native cellular environment Tg is expressed as it is exclusively found within thyroid tissue. Our previous dataset was collected across two different instruments with similar LC-MS/MS methodology. Here, this dataset was collected on a single instrument after performing further method optimization from our methodology used to acquire the first dataset. In line with our LC-MS/MS methodology development, the sample processing workflows here are quite different. Our previous dataset utilized 6plex TMT labeling with globally immunoprecipitated samples from various Tg constructs. Global immunoprecipitation of Tg leads to much larger protein sample amounts than the TRIP methodology presented here, which we coupled with 16plex TMTpro labeling. This is also one of the reasons we chose to deploy a booster/carrier channel within our experimental labeling schemes.

Reviewer #2, Comment #3: *For the temporal interaction analysis the authors describe differences in the temporal profiles of selected interactions comparing wt and mutant, however no statistical analysis is performed comparing wt and mutant interaction profiles across the time course. Furthermore the variability between the replicates for the temporal profiles is not shown and some of the temporal profiles appear to be noisy. A more rigorous statistical analysis should be performed including additional biological replicates to evaluate the changes over the time course, especially as the temporal interaction analysis is the novelty of this study.*

Our Response: Please also see our response to **Reviewer #2, comment 1** above. We previously presented an analysis of the variability of the TRIP measurements (SEM) (now in **Appendix Fig S1B**). We have since provided further statistical analysis found in the updated **Fig 2B,C,J**, which include 25-75% quartile ranges for respective proteostasis network pathways. We also included SEM for the time profiles of individual interactors in **Fig EV4**.

To assess the divergence in time profiles in an unbiased way, we added a k-means hierarchical clustering analysis (**Fig 3F, Fig. EV5**). These clustering data allow for the visualization and analysis of temporal protein interaction profiles that are similar to one another and how groups of interactors shift between different clusters for WT Tg and the C1264R mutant.

Reviewer #2, Comment #4: *To functionally validate interactors derived from the TRIP analysis as well as to identify factors that can rescue mutant Tg secretion the authors developed an RNA interference screen. There are a number of aspects that need to be addressed/clarified for this part of the study.*

Our Response: We have added some clarifying changes to the text and the figure panels associated with the siRNA screening and follow-up experiments on the trafficking and degradation factors that rescue Tg secretion. We have addressed other comments from **Reviewers #3 and #4** related to these portions of the paper and hope that Reviewer #2 finds them satisfactory.

Reviewer #2, Comment #5: *While the authors validate the stable cell lines expressing the nanoluciferase tagged Tg and the linearity of luminescence signal in lysate and media carefully, they do*

not validate their platform in combination with the RNAi knockdown strategy. The authors should select genes as positive controls that are expected to modulate Tg secretion and demonstrate that the knockout of these positive controls indeed results in changes in Tg secretion in their system.

Our Response: This is an excellent suggestion and certainly something we would have done given any prior knowledge on known control genes that would positively or negatively regulate Tg secretion. The purpose for developing the siRNA screening platform was to investigate and hopefully discover genes that are able to positively or negatively regulate Tg processing. We have done so to the best of our ability, identifying for example NAPA which positively regulates WT Tg secretion, as seen by the decrease in WT Tg secretion when treated with NAPA siRNA. Conversely, we found that VCP may negatively regulate C1264R Tg secretion, as discovered by the increase in secretion with VCP siRNA or ML-240 treatment. We included a standard “TOX” siRNA control, which we knew would likely negatively affect WT Tg secretion and this was indeed the case. As we stated within the manuscript:

“This is the first study to broadly investigate the functional implications of Tg in-teractors and other PQC network components on Tg processing.”

Reviewer #2, Comment #6: *For the screen the authors select 167 Tg interactors and PN (Proteostasis network) related factors. This statement is very vague and the authors should clarify which genes were knocked down and which criteria were applied to narrow down the list of interactors and to select PN factors. The authors should therefore provide a supplementary table including all genes included in the screen, their source (were this derived from the initial study by Wright et al, from the current study or compiled from prior knowledge about PN), as well as their results from the screen based on luminescence in media and lysate. It is unclear how many of the selected factors are actually coming from the TRIP analysis.*

Our Response: The list of genes included within the siRNA screen, as well as the results were previously included, and are now included in **Appendix Fig S2**. We have further provided the information requested by Reviewer #2 within **Dataset EV5** indicating whether a gene was included in the siRNA screen due to its identification within our previous proteomics dataset (*Wright et al, 2021.*), the proteomics dataset presented here, or based upon primary literature. We added a comment in the text:

“Moreover, we were interested in identifying factors whose modulation may act to rescue mutant Tg secretion. HEK293 cells were engineered to stably express nanoluciferase-tagged Tg constructs (Tg-NLuc) and screened against 167 Tg interactors and related PN components (see **Dataset EV5** for the list of genes).”

Reviewer #2, Comment #7: *Only a small number of the 167 selected genes shows an effect on Tg abundance/secretion. How do the authors explain this result? Would we not expect that Tg interactors, especially those from the TRIP method which interact with the newly synthesized are more enriched for functionally relevant genes.*

Our Response: The proteostasis network contains genes and proteins of high redundancy in structure and function, and many single-gene knockdowns are likely insufficient to have a large impact on Tg abundance or secretion. In fact, these results are in line with what we would have expected when designing these experiments. Our goal here was to identify the key players that control Tg protein quality control.

We explain the proteostasis network redundancy in the manuscript:

“The functional implications of protein-protein interactions can be difficult to deduce, especially in the case of PQC mechanisms containing several layers of redundancy across stress response pathways, paralogs, and multiple unique proteins sharing similar functions (Wright & Plate, 2021; Bludau & Aebersold, 2020; Karagöz *et al*, 2019; Braakman & Hebert, 2013).”

Reviewer #2, Comment #8: *The authors initially performed the screen in HEK293 cells and as a second step wanted to validate the hits from the HEK cells in more relevant Fisher rat thyroid cells. Indeed they could show that knockdown of NAPA increased WT Tg in lysate and decreased WT Tg secretion. Furthermore, they further validated genes to modulate mutant Tg lysate and media abundance. The authors should perform a rescue experiment to demonstrate that the observed phenotype can be reversed through re-introduction of NAPA.*

Our Response: We have now performed the requested *NAPA* complementation experiments and provided the data within **Fig EV 7I**. Overexpression of a human, siRNA-resistant *NAPA* construct partially reversed the increase in WT Tg lysate retention. These results further support the identification of *NAPA* as a pro-trafficking factor for WT Tg. We updated the manuscript text to include these data as follows:

“To understand if these results were directly attributable to *NAPA* function, we performed complementation experiments where FRT cells treated with *NAPA* siRNAs were co-transfected with a human *NAPA* plasmid. WT Tg lysate abundance decreased when *NAPA* expression was complemented, confirming that the observed retention phenotype could be attributed to *NAPA* silencing (**Fig EV7I**). These results established that *NAPA* acts as a pro-secretion factor for WT Tg.”

Reviewer #2, Comment #9: *One hit from this analysis was the ER-phagy receptor TEX264, while TEX264 was not identified in the TRIP data, is selectively increased the C1264R secretion, but not wt and the other Tg mutant. Following Co-IP data however revealed some interaction between the C1264R and to a lesser extent the A2234D mutant. How do the authors explain that TEX264 was missed in the TRIP dataset?*

Our Response: The TRIP samples are of much lower protein abundance compared to globally purified samples used for the Co-IP analysis. While the interaction is seen with the globally purified Co-IP samples, this interaction is likely much more difficult to capture with the low abundance, time-resolved samples that are acquired through the TRIP workflow, especially if this interaction is transient or requires the coordination of other accessory proteins as has been detailed in the literature and discussed within the manuscript presented here:

“While A2234D and C1264R Tg were preferentially enriched with TEX264 compared to WT, it remains unclear what other accessory proteins may be necessary for the recognition of TEX264 clients (Chino *et al*, 2019; An *et al*, 2019). Furthermore, TEX264 function in both protein degradation and DNA damage repair further complicates siRNA-based investigations (Fielden *et al.*, 2022). Further investigation is needed to fully elucidate 1) if Tg degradation takes place via ER-phagy and 2) by which mechanisms this targeting is mediated.”

Minor comments:

Reviewer #2, Comment #10: *The workflow needs to be described clearer. For example, it should be better explained why the authors selected a two-stage enrichment strategy, I assume that the first based on the Flag affinity tag is to purify the protein of interest and the second step based on the*

incorporation and functionalization of the unnatural amino acids to enrich for the newly synthesized fraction at specific time points after protein synthesis? These are critical steps for the method but the rationals are not well explained, neither in the text nor the figures captures all these steps of the method very clearly, which makes it really difficult for the reader to understand the individual steps of the method. Moreover, the structures in Figure 1 workflow are not clearly labeled, so that it is confusing which part represents which protein/molecule.

Our Response: Thank you for this feedback. We have updated **Fig 1** to provide more detail to provide more clarity for the readers. Furthermore, we have edited the text to more clearly describe the workflow:

“To develop the time-resolved interactome profiling method, we envisioned a two-stage enrichment strategy utilizing epitope-tagged immunoprecipitation coupled with pulsed biorthogonal unnatural amino acid labeling and functionalization (**Fig 1A**). Cells can be pulse labeled with homopropargylglycine (Hpg) to synchronize newly synthesized populations of protein. After pulsed labeling with Hpg, samples can then be collected across time points throughout a chase period (**Fig 1A**, Box 1) (Kiick *et al*, 2001; Beatty *et al*, 2006). The Hpg alkyne incorporated into the newly synthesized population of protein can be conjugated to biotin using copper-catalyzed alkyne-azide cycloaddition (CuAAC) (**Fig 1A**, Box 2). Subsequently, the first stage of the enrichment strategy can take place where the client protein of interest is globally captured and enriched using epitope-tagged immunoprecipitation, followed by elution (**Fig 1A**, Box 3). The second enrichment step can then utilize a biotin-streptavidin pulldown to capture the Hpg pulse-labeled, and CuAAC conjugated population, enriching samples into time-resolved fractions (**Fig 1A**, Box 4) (Li *et al*, 2020; Thompson *et al*, 2019).”

Reviewer #2, Comment #11: *Except for the general workflow shown in Figure 1, a more detailed workflow showing the experimental steps, such as the sample fractions with the following steps could be added so that the design of the method is clearer. Also the style of the workflows including Figure 1, Figure 2A, and Figure 3A are different. It would be helpful to make them the same style and make the Figure 2A as a zoom in or more detailed illustration on part of Figure 1.*

Our Response: Thank you for this feedback. In addition to updating **Fig 1**, we also expanded **Fig 2A** to more clearly outline the experimental steps in the TRIP workflow. Assuming the term “style” used here is in reference to color pallets and figure schematics used, these have been updated to ensure they are agreeable aesthetically across manuscript figures.

Reviewer #2, Comment #12: *A summary of proteomics results of time course labeling after all enrichment steps, including the total number of identified proteins at different conditions and control would be helpful for having an overview impression on the proteomics results*

Our Response: We have included an updated **Dataset EV1** that provides a summary of proteomics data included which runs given proteins were identified in, % of TMT channels quantified, % of Hpg Pulse channels quantified, and generally number of proteins quantified across runs for each construct.

Reviewer #2, Comment #13: *In Figure 2B, the WB for PDIA4 in the Biotin PD elution is missing. Why was the PDIA4 interaction missing for the time course analysis, but the interaction was captured in the initial test for Wt Tg (Figure 1D). Additionally, in this panel the Rhodamine Probe Gel shows inconsistencies at the time points 1.5 - 3h. Does this mean that the labeling did not work well for these conditions? As we would expect a consistent Rhodamine Probe signal at every time point.*

Our Response: Please also see our response to **Reviewer #1, comments 3 & 11**. **Fig 1D** features continuous Hpg labeling for 4 hours to ensure that most intracellular Tg is labeled for this proof-of-concept experiment for the two-stage enrichment strategy. **Fig 2B** features a shorter 60 minute pulse of Hpg labeling, prior to the full chase period and two-stage enrichment strategy. PDIA4 interactions were detectable throughout **Fig 1D** because those measurements captured a larger population of labeled Tg, whereas in **Fig 2B** Tg bait protein amounts were much smaller after the two-stage enrichment procedure to capture the time-synchronized population.

The Rhodamine/TAMRA Probe Gel in **Fig 2B** does not have inconsistencies in Tg abundance, but highlights the fact that pulse labeled WT Tg is being secreted or degraded in FRT cells. As you would expect as time continues during the chase period, intracellular WT Tg signal decreases as secretion and degradation take place. Constant Rhodamine/TAMRA probe signal would not be expected here. Consistent with this, the C1264R Tg signal remains more stable for the initial time course. This is expected as the C1264R Tg variant is retained intracellular undergoing increased interactions the proteostasis network. We have removed the PDIA4 panel for WT Tg because there was no signal above the detection limit. This is now explained as follows:

“For WT Tg, interactions with HSPA5 peaked within the first 30 minutes of the chase period and rapidly declined, in line with previous observations, but PDIA4 interactions were not detectable by western blot analysis (**Fig 2B**) (Menon *et al*, 2007; Kim & Arvan, 1995).”

Reviewer #2, Comment #14: *In Figure 2, why was there no WB results for the A2234D? In Figure 2D and 2E, at which time point are the changes significant compared to WT?*

Our Response: We did not perform the WB experiments with A2234D. We used WT and C1264R Tg in our proof of concept experiments via WB and decided to move forward with analyzing A2234D Tg by LC-MS/MS. Please see our response above to **Reviewer #2, comment 3** for information on the statistical analysis.

Reviewer #2, Comment #15: *All figure legends should indicate how many biological replicates were performed for each experiment represented in the figure.*

Our Response: We have updated the figure captions to include this information where applicable.

Reviewer #2, Comment #16: *The heatmaps shown in Figure 3, Figure 3 - Figure Supplement 3, and Figure 7 are in the current form incomprehensible. The heatmaps depict the relative enrichment vs the control sample, which was scaled between 1 and -1. The color coding with 5 different colors from 1 to -1 is very confusing and should be changed to just two colors, one for positive and one for negative relative enrichment. I would also suggest changing the visualization of the heatmap showing the wt and mutants side by side, instead of stacked on top of each other for each individual protein.*

Our Response: Thank you for this feedback, and we apologize for the confusion. We adjusted our data analysis approach by removing previous negative enrichment values. As these served only as “background” within the dataset, they did not carry much meaning. The TRIP enrichment is now scaled from 0 to 1, where a value of 1 represents the time point at which the enrichment is greatest, while 0 represents the background intensity in the (-) Hpg control sample. The associated figures have been updated accordingly, and we feel they are now more comprehensible and aesthetically pleasing. We opted to keep the *Viridis* color scheme in the heatmap to allow for more nuanced differentiation of the enrichment values.

Reviewer #2, Comment #17: *The data analysis method for generating relative enrichment shown in the heatmap is not explained. This should be described in the method section for a better understanding of the data analysis.*

Our Response: We have edited the methods section as follows to better explain the analysis:

“For time resolved analysis, data were processed in R with custom scripts. Briefly, TMT abundances across chase samples were normalized to Tg TMT abundance as described previously and compared to (-) Hpg samples for enrichment analysis (Wright et al, 2021). For relative enrichment analysis, the means of log2 interaction differences were scaled to values from 0 to 1, where a value of 1 represented the time point at which the enrichment reached the maximum, and 0 represented the background intensity in the (-) Hpg channel. Negative log2 enrichment values were set to 0 as the enrichment fell below the background.”

Reviewer #2, Comment #18: *There are no legends of flowcharts in Figure 2A and Figure 3A and it is difficult to understand which are the key components in the complex and what are the differences among different periods of labeling.*

Our Response: We have now consolidated **Fig 2A** and **Fig 3A** into a single panel found in **Fig 2A**, which is significantly reorganized to better explain the TRIP workflow. The caption has additionally been updated to highlight key steps within the workflow with numbering to allow readers to follow and visualize the steps more easily. The figure caption now reads as follows:

“(A) Workflow for TRIP protocol utilizing western blot or mass spectrometric analysis of time-resolved interactomes. (1) Cells are pulse-labeled with Hpg (200µM final concentration) for 1 hr, chased in regular media for specified time points, and cross-linked with DSP (0.5mM) for 10 minutes to capture transient proteostasis network interactions; (2) Lysates are functionalized with a TAMRA-Azide-PEG-Desthiobiotin probe using copper CuAAC Click reaction; (3) Lysates undergo the first stage of the enrichment strategy where the Tg-FT is globally captured and enriched using immunoprecipitation; (4) Eluted Tg-FT populations from the global immunoprecipitation undergo biotin-streptavidin pulldown to capture the pulse Hpg-labeled, and CuAAC conjugated population of Tg-FT, enriching samples into time-resolved fractions; (5) Time-resolved fraction may then undergo western blot analysis or (6) quantitative liquid chromatography – tandem mass spectrometry (LC-MS/MS) analysis with tandem mass tag (TMTpro) multiplexing or analysis. The (-) Hpg control channel is used to identify enriched interactors and a (-) Biotin pulldown channel to act as a booster (or carrier).”

Reviewer #2, Comment #19: *Why did only one of the VCP inhibitors (ML-240) exhibit a phenotype in Tg abundance and secretion, but not the other VCP inhibitors?*

Our Response: Please also see our response to **Reviewer #3, comment 2** below. This could be due to a number of reasons, but we added a brief discussion on the mechanisms of action for the inhibitors that may at least partially explain the differences in phenotype seen with the VCP inhibitors. We updated the text as follows:

“ML-240 and CB-5083 are ATP-competitive inhibitors that preferentially target the D2 domain of VCP subunits, whereas NMS-873 is a non-ATP-competitive allosteric inhibitor which binds at the D1-D2 interface of VCP subunits (Chou et al, 2013, 2014; Anderson et al, 2015; le Moigne et al, 2017; Tang et al, 2019). ML-240 and NMS-873 have been shown to decrease both proteasomal degradation and autophagy, in line with VCP playing a role in both processes (Chou et al, 2013, 2014; Her et al, 2016). Conversely, while CB-5083 is known to decrease proteasomal degradation it has been shown to increase autophagy. (Anderson et al, 2015; le Moigne et al, 2017; Tang et al, 2019).”

Reviewer #2 (Significance (Required)):

Reviewer #2, Comment #20: The authors **describe a novel and elegant method to map time resolved protein interactions of newly synthesized proteins**, which allows monitoring of proteins regulating protein quality control.

Authors describe it as a general method, however, they only demonstrate the applicability to one protein and do not systematically evaluate the quantitative nature of their approach by determining quantitative reproducibility, which would be necessary to be able to claim that this is a method with broad applicability.

Given my expertise in quantitative proteomics, I can mainly comment on the technological aspects of the proteomics part of the manuscript, but do not feel qualified to evaluate the significance of this study in terms of novel biology. Nevertheless, **it feels that there is a stronger emphasis on the biology in the current form of the manuscript which will raise interest of scientists with a focus on protein quality control and Tg biology.**

Reviewer #3 (Evidence, reproducibility and clarity (Required)):

Provide a short summary of the findings and key conclusions (including methodology and model system(s) where appropriate). Please place your comments about significance in section 2.

In this manuscript, the authors describe their efforts to develop a methodology for determining time-resolved protein-protein interactions using quantitative mass spectrometry. With TRIP (time-resolved interactome profiling), they combine a pulsed bio-orthogonal unnatural amino acid labelling (homopropargylglycine, Hpg), CuAAC conjugation and biotin-streptavidin pull-downs to enrich at different timepoints and time-resolve by combining TMT labelling and LC-MS/MS (Figure 1). This technique is then applied to the maturation of the secreted WT and mutant thyroglobulin (Tg-WT, Tg-C1264R, Tg-A2234D) expressed in HEK293 and rat thyroid cells (FRT) and linked to hyperthyroidism. There, they identify a collection of ER resident proteins involved in protein folding/processing (e.g. chaperones, redox, glycans, hydroxylation) as well as degradation (e.g. autophagy, ERAD/proteasomes) (Fig. 2). Here the authors effectively use pulse-labelled form of TRIPs to highlight the different interactions formed with Tg-WT vs. Tg-mutants during biogenesis and secretion (or retention). The analysis found ~200 new interactions compared to previous studies along with about 40% of those identified previously. Differences in interactions were observed for mutants, which shown extended interaction with chaperones and redox processing pathways. While many interactions appeared as might be expected, the identification of membrane protein processing elements (e.g. EMC, PAT) was puzzling and raised some questions about the specificity within the protocol. Mutants enriched for CANX CALR and UGGT, suggesting prolonged association with glyco-processing factors. Interaction of C1264R with the ER-phagy factors CCPG1 and RTN3 was greater than WT. The authors note that their interaction correlated with that of EMC1 & 4, but it is not clear why that might be.

With interactors in hand, the authors complemented the TRIP protocol with siRNA KD of identified factors, to investigate any changes to secreted vs intracellular Tg upon loss. KD of NAPA (a-SNAP) and LMAN1 increased WT lysate (intracellular) Tg but not mutants. NAPA also reduced Tg-WT secretion. In contrast, KD of NAPA increased A2234D secretion while LEPRE1 increased C1264R (but not A2234D or WT), suggesting mutants have differential processing paths and requirements. KD of VCP increased secretion of both mutants. Some ER-phagy receptors were found among interactors (e.g. RTN3 in Tg-C1264R only) but often their KD had no impact on secretion (CCPG1, SEC62, FAM134B). NAMA observations were recapitulated in thyroid derived cell line (FRT). KD of TEX264 and VCP increased Tg-C1264 secretion while RTN3 KD in FRTs decreased Tg-C1264 secretion. This was in contrast to data from HEK293s for reasons that are not clear. Co-IP with TEX264 enriched for all Tg forms but more so for C1264R and A2234D - motivating the authors to propose selective targeting of Tg to TEX264 and the consideration of ER-phagy as a "major" degradative pathway during Tg processing.

Given the observations with siRNAs to VCP, the authors next use a selection of VCP inhibitors to ask whether secretion can be rescued upon pharmacological impairment of the AAA ATPase. They observed that ML-240, but interestingly not the more conventionally used CB-5083 or NMS-873, increased secretion of Tg-C1264R but not lysate. Inhibitors increased lysate but decreased the secreted fraction for Tg-WT (Fig 7). Finally, the authors used TRIP again in ML-240 treated Tg-C1264R expressing cells to look for changes to interactome with treatment - observed decreases to glycan and chaperone interactions, CANX and UGGT1, decreased interaction with DNAJB11 and C10, like that of WT. There was no apparent change to the UPR, although activation was not directly measured.

Major comments:

Reviewer #3, Comment #1: Are the key conclusions convincing? **The TRIP methodology appears to be quite robust and should be a powerful strategy for this field and others going forward.** The drawback will be the length of pulse required will limit the number/type of proteins to be monitored to ones with longer t1/2's. There were interesting interactions found with Tg and the mutants linked to hyperthyroidism, but cut and dry differences did not appear as obvious, even though strong "trends" appear to be present. The path from identifying interactors in a time-resolved manner to then following them up with targeted KD does provides some clarity, which is important.

Our Response: We thank Reviewer #3 for their time in reviewing our manuscript and providing this positive feedback. We have enhanced our analysis of the TRIP data to more clearly highlight difference in time profiles between WT and mutant variants. Please see our response to **Reviewer #2, comment 1 & 3**. We also highlight the limitations of the time resolution in the discussion (see also **Reviewer #2, comment 6**):

“To address this, we utilized a labeling time of 1 hr which allows us to generate a large enough labeled population of Tg-FT for TRIP analysis, but some early interactions are likely missed within the TRIP workflow. In the case of mutant Tg, performing the TRIP analysis for much longer chase periods (6-8 hrs) may provide insightful details to the iterative binding process of PN components that is thought to facilitate protein retention within the secretory pathway.”

We have addressed all further comments below.

Reviewer #3, Comment #2: Should the authors qualify some of their claims as preliminary or speculative, or remove them altogether? The data regarding VCP silencing and pharmacological impairment appear clear but leave some questions outstanding in this reviewer's opinion. The lack of effect with the 2 highly selective inhibitors suggests that the underlying mechanism for switching fate of intracellularly retained Tg-C1264R towards secreted forms is not at all clear. ML-240 is an early derivative of DBeQ and reportedly impairs both ERAD and autophagic pathways, similarly to DBeQ. The differences between the VCP inhibitors' mechanism of action were not discussed, but perhaps should be elaborated upon, particularly in the matter of how ERAD and ER-phagy pathways might be being differentially affected. At the risk of asking for too many additional experiments, this reviewer would just prefer to see this fleshed out in a bit more detail.

Our response: We agree with Reviewer #3 that the underlying mechanism for switching fate of the intracellular retained Tg-C1264R towards secreted forms remains unclear. We have added additional text to discuss further the details surrounding the inhibitors used and the general manner in which ERAD and ER-phagy pathways can be affected. This added text reads as follows:

“ML-240 and CB-5083 are ATP-competitive inhibitors that preferentially target the D2 domain of VCP subunits, whereas NMS-873 is a non-ATP-competitive allosteric inhibitor which binds at the D1-D2 interface of VCP subunits (Chou *et al*, 2013, 2014; Anderson *et al*, 2015; le Moigne *et al*, 2017; Tang *et al*, 2019). ML-240 and NMS-873 have been shown to decrease both proteasomal degradation and autophagy, in line with VCP playing a role in both processes (Chou *et al*, 2013, 2014; Her *et al*, 2016). Conversely, while CB-5083 is known to decrease proteasomal degradation it has been shown to increase autophagy. (Anderson *et al*, 2015; le Moigne *et al*, 2017; Tang *et al*, 2019).”

“As we discovered that pharmacological VCP inhibition with ML-240 can rescue C1264R Tg secretion yet is detrimental for WT Tg processing, it is unclear whether VCP may exhibit distinct functions for WT and mutant Tg PQC. Finally, as ML-240 is shown to block both the proteasomal and autophagic functions of VCP it is unclear which of these pathways may be playing a role in the rescue of C1264R, or detrimental WT processing (Chou *et al*, 2013, 2014).”

Reviewer #3, Comment #3: Q1. The degree (if any) of Tg-C1264 aggregation during and/or detergent solubility do not appear to have been considered as a potential source of the increase in released secreted material (Figure 4, 5). Do Tg mutants partition into RIPA-insoluble fractions at all? That is to say.. is the total population of synthesized Tg being considered? A full accounting? Could the authors address this and if biochemical extraction data (via urea or high SDS) is available, include it to answer this concern.

Our response: The transient aggregation of Tg has been investigated in some detail previously (Kim *et al*, 1992, 1993). The transient aggregates have the ability to partition into RIPA-insoluble fractions. Of note, these aggregates are shown to be made up, at least in part, of mixed disulfide linkages requiring reducing agent to fully resolubilize. With that being said, these aggregates represent a minority of the overall Tg population. In our prior manuscript (Wright, *et al*. 2021), we quantified the RIPA-insoluble fraction found in the pellet (see Supplemental Info Fig. 5). As the majority of Tg remains soluble during processing it should be able to be captured via our TRIP methodology. That is to say, we are capturing most of the Tg that is available for analysis while understanding that some smaller population of Tg remains in RIPA-insoluble fractions.

Reviewer #3, Comment #4: Q2. Along the same lines, what does Tg-WT and mutant expression look like by microscopy? Is Tg-WT uniformly distributed while Tg-mutants appear in puncta... more aggregated - perhaps reflecting the increased engagement of chaperones and redox machinery?

Changes in the pattern of Tg-C1264R mutant (e.g. w/ VCP KD or inhibition) would add additional support for the authors interpretation of improved secretion. If this data is at hand, including it might be worth consideration.

Our response: Thank you for this suggestion. The subcellular localization of Tg and any changes from proteostasis modulation is an ongoing area of follow up work in our lab. We have some preliminary results that the localization for WT and C1264R Tg indeed differs. However, given that this manuscript is already dense in information, we opted to reserve this data for a future manuscript where we plan to further elucidate the targeting mechanism of mutant Tg to VCP or TEX264. We direct the reviewer to work published by Zhang et al, 2022, (<https://doi.org/10.1016/j.jbc.2022.102066>) showing a staunch difference of WT vs mutant Tg in the localization from intracellular to a secreted population in rat tissue. While most all WT Tg is found in the follicular lumen (secreted), mutant Tg heavily co-localizes with the ER resident chaperone BiP. While this paper does not go into detail on the differences in subcellular localization, it further highlights the drastic changes in Tg processing and how these manifest in distinct differences in localization within tissue.

Reviewer #3, Comment #5: Q3. Does the level of Tg mutant expression in the FRT clones impact the profiles obtained by TRIP? (Figure 3). This is a question of gauging the relative saturation of QC machinery and how that might impact profiles from TRIP. Were clones expressing at different levels tested? Perhaps a brief discussion of this.

Our response: We do not foresee an impact from level of Tg expression on the profiles obtained by TRIP. We were able to identify distinct profiles because we processed the data and normalized it based on the relative Tg amount. For example, while WT and A2234D Tg are expressed at similar levels intracellularly, we were able to identify distinct differences in the interaction profiles across the two constructs. When developing FRT clones, we selected those that were expressed at similar levels and, therefore, did not have the capability to directly test differences, if any, in observed profiles that may be the result of different expression levels of the same Tg construct. Furthermore, Tg can make up 50% of all protein content within thyroid tissue (Di Jeso & Arvan, 2016). As such, thyroid cells are adept at maintaining the balance of QC machinery to process thyroid. Therefore, we do not anticipate that the amount of Tg expressed in TRIP experiments would have a significant impact on the profiles that we were able to observe.

Reviewer #3, Comment #6: Q4. For Figure 3, the hour-long labelling period seems a bit long, compared with 3 hr of chase. Perhaps this reviewer missed this but how long does Tg take to mature and/or mutants to misfold and degrade? Is there any possibility to shorten this so that the profiles of labelled Tg could be more synchronized? If not, perhaps this could just be discussed.

Our response: While the 1-hour labeling period may seem long, we had to balance the labeling time to 1) label a large enough population of Tg for it to remain detectable throughout the chase period, and 2) keep the chase period long enough to capture the large majority of Tg processing. In our hands we found that by 4 hours WT Tg was ~63% secreted, with ~25% retained intracellularly (**Fig EV7H**). Conversely, we found that C1264R remains very stable over this period with most protein being retaining intracellularly and little degradation taking place (**Fig EV9A**). Hence, we opted for the overall ~4 hour total for sample processing (1 Hr pulse labeling + 3 hour chase period for time point collections). Literature suggest that WT Tg takes ~2 hours to be processed within the ER and reach the medial golgi. This is exemplified by the EndoH resistant population that appears at this ~2 hour time point (Menon et al. *JBC*. 2007). Please also see our response to **Reviewer #1, comment 6**. We updated the text as follows:

“We pulse labeled WT Tg FRT cells with Hpg for 1 hr, followed by a 3 hr chase in regular media capturing time points in 30-minute intervals and analyzing via western blot or TMTpro LC-MS/MS (**Fig 2A**). Our previous study indicated that ~70% of WT Tg-FT was secreted after 4 hours, while approximately 50% of A2234D and 15% of C1264R was degraded after the same time period (Wright *et al*, 2021). Therefore, we reasoned that a 3-hr chase period would be a enough time to capture the majority of Tg interactions throughout processing, secretion, cellular retention, and degradation, while still being able to capture an appreciable amount of sample for analysis.”

We anticipate that this labeling period can be decreased with future iterations of this methodology. This will also be bolstered by the continued improvements that come about within quantitative proteomics in increased instrument sensitivity and improved sample preparation methods that have the ability to decrease sample loss.

We explain the labeling timeline and limitations further in the discussion:

“To address this, we utilized a labeling time of 1 hr which allows us to generate a large enough labeled population of Tg-FT for TRIP analysis, but some early interactions are likely missed within the TRIP workflow. In the case of mutant Tg, performing the TRIP analysis for much longer chase periods (6-8 hrs) may provide insightful details to the iterative binding process of PN components that is thought to facilitate protein retention within the secretory pathway.”

Reviewer #3, Comment #7: Q5. It is curious that only ML-240 and not other well characterized inhibitors of VCP/p97, has an effect, as both are used far more often than ML-240. The authors do not really address this in detail but does it suggest that the ML-240 effect on VCP/p97 could be affecting different pathways, given the nature of this compound. Is this compound acting on Tg-C1264R maturation at the level of translation or post-translationally? If the latter, through what means?

Our Response: We thank Reviewer #3 for appreciating this surprising finding. We were similarly curious as to how, or why ML-240 was able to elicit this effect compared to other VCP inhibitors. We elaborated in the manuscript text on these compounds and on how the ERAD and ERphagy pathways, utilizing VCP, may be differentially regulated (See response to **Reviewer #3, Comment 2**). While speculative, we believe that ML-240 acts on C1264R Tg maturation post-translationally. This is given by the fact that ML-240 does not seem to affect the translational velocity of C1264R Tg, as **Fig EV9A** shows similar levels of 35S-labeled C1264R in DMSO or ML-240 treated cells. It may be the case that acute treatment with ML-240 alters the folding vs degradation balance of the ER proteostasis network in such a way that some population of C1264R that is usually degraded is able to be secreted. Another Tg mutation G2320R was shown to be degraded via the proteasome in PLCCL3 thyrocytes, as MG-132 treatment slowed mutant Tg degradation (Menon *et al. JBC*. 2007), although G2320R degradation was not be exclusively proteasomal. The L2284P Tg mutation exemplified similar results to G2340R where MG-132 slowed degradation. Furthermore, L2284P Tg was not affected by autophagic/lysosomal inhibitors chloroquine and E64 (Tokunaga *et al. JBC*. 2000), suggesting ERAD more exclusively degrades L2284P. It is unclear which degradation pathway, ERAD or ER-phagy, may be the predominate pathway for C1264R Tg degradation. Furthermore, we do not exclude the possibility that both may be at play and affected by treatment with ML-240.

We utilized our HEK293 Tg-NLuc cells and screened other proteasomal and lysosomal inhibitors bafilomycin and bortezomib. Neither of these compounds were able to rescue A2234D or C1264R

secretion, highlighting that the effect is specific to ML-240 treatment. This new data is now shown in **Fig EV10A,B** and described in the text:

“To understand whether this rescue in secretion was uniquely linked to VCP inhibition or could be more broadly attributed to blocking Tg degradation, we tested the proteasomal inhibitor bortezomib, and lysosomal inhibitor bafilomycin. Bafilomycin increased WT Tg lysate abundance, and bortezomib significantly increased A2234D lysate abundance, consistent with a role of these degradation processes in Tg PQC (**Fig EV10A**). When monitoring Tg-NLuc media abundance, neither bafilomycin nor bortezomib significantly altered WT, A2234D, or C1264R abundance (**Fig. EV10B**), confirming that general inhibition of proteasomal or lysosomal degradation does with rescue mutant Tg secretion.”

Reviewer #3, Comment #8: Q6. Continuing from Q5.. At what point and where is VCP/p97 able to affect mutant Tg processing? In line 317, the authors seem to correlate increased VCP association with mutants to their increased secretion. It is not clear how this would result, as engagement with VCP would be in a compartment different to that which supports trafficking and secretion. Could the authors expand on how this might come about. This is also relevant to the ML-240 data in Figure 7. Moreover, VCP is associated with ERAD (as is HerpUD1) rather than ER-phagy and at least in the siRNA raw data, there are also effects from Derlin3 and FAF2 KDs.. both ERAD factors. Some clarity here would be appreciated.

Our Response: This line of discussion in the text was meant to suggest that, since VCP showed a higher enrichment for mutant Tg, particularly C1264R, it would make sense that inhibiting VCP would have a larger effect on mutant Tg processing as compared to WT Tg. As we saw with the siRNA screening data, suppression of VCP resulted in increased C1264R secretion, while not affecting WT Tg processing. This passage was not intended to suggest that increased VCP association with mutant Tg found within the TRIP dataset was the *reason* for rescued secretion. These are two different sets of experiments and environments in which these data are captured. We were simply looking for the opportunity to bridge the findings from the two sets of experiments to a single discussion point. Of note, we understand that VCP is associated with ERAD *and* acts to regulate autophagy. Given that core autophagy machinery is relevant for both bulk autophagy and ER-phagy, we did not want to rule out the fact that VCP inhibition via ML-240 *could* affect autophagic flux in these experiments (Chou et al. *Chemmedchem*. 2013; Khaminets et al. *Nature*. 2015; Hill et al. *Nat. Chem. Bio*. 2021.)

It is great that the reviewer also noted that DERL3 and FAF2 knockdown increased C1264R Tg secretion. Since these ERAD factors did not reach the defined threshold in the screen, we did not include further discussion, but this data remains available in **Appendix Fig S3**. We have updated the manuscript text to clarify the previous points we aimed to make. The text now reads as follows:

“VCP silencing exclusively affecting mutant Tg corroborates our TRIP dataset, and suggest a more prominent role for VCP in mutant Tg PQC compared to WT. VCP interactions were sparse for WT Tg while they remained more steady throughout the chase period for the mutants (**Fig 3H,K**).”

Reviewer #3, Comment #9: Q7. There does not appear to be a direct demonstration of Tg-C1264R turnover by ER-phagy (via TEX264). Given the inconsistency with it not being detected by TRIP, while another receptor RTN3 was, but has not impact on Tg-C1264R secretion, perhaps including that data would go some way to demonstrating a fate of ER-phagy (at least partly) for this mutant.

Our response: We performed follow-up experiments to test interactions with Tg and the wider panel of ER-phagy receptors. We transiently expressed FLAG-tagged CCPG1, RTN3L, and TEX264 in HEK293

cells stably expressing Tg-NLuc and performed FLAG IPs followed by western blot analysis. We found that WT and C1264R Tg were enriched, albeit modestly, in the RTN3L Co-IP compared to control samples expressing GFP. Additionally, we found that WT, A2234D, and C1264R Tg were all enriched with CCPG1 compared to control samples expressing GFP. CCPG1 was found to be a C1264R Tg interactor within our mass spectrometry datasets, along with RTN3. We have now integrated these data into the manuscript as **Fig EV8**, and updated the manuscript text as follows:

“Additionally, we monitored Tg enrichment with ER-phagy receptors CCPG1 and RTN3 via Western blot as both were found to be C1264R Tg interactors within our TRIP dataset. RTN3L is found to be the only RTN3 isoform involved in ER turnover via ER-phagy (Grumati et al, 2017). WT and C1264R Tg-NLuc were modestly enriched with RTN3L compared to control samples expressing GFP. Conversely, we found that all Tg variants exhibited modest interactions with CCPG1 compared to control samples expressing GFP, although less than with TEX264 (**Fig EV8**).

Together, these data suggest that TEX264, CCPG1, or RTN3L engage with Tg during processing, and CH-associated Tg mutants may be selectively targeted to TEX264. Furthermore, ER-phagy may be considered as a degradative pathway in Tg processing, as other studies have mainly focused on Tg degradation through ERAD (Tokunaga *et al*, 2000; Menon *et al*, 2007).”

Whether the TEX264 recruitment of mutant Tg leads to degradation remains to be tested. When we monitored C1264R Tg degradation by pulse-chase assay (**Fig. EV9A**), only a small fraction (< 20%) gets degraded by 4 hr. This observation raises the question whether ER-phagy is involved in this degradation or whether TEX264 assumes an alternative function, for example in intracellular sequestration. While ongoing work in the lab is addressing this question, we believe this is better reserved for a follow-up manuscript.

Reviewer #3, Comment #10: Q9. The authors provide data that the UPR was not induced by ML-240 at 3hrs (10 μ M) (Figure 7, supplemental 1). This is in stark contrast to the results of Chou et al (2013) which the authors reference, reporting that ML-240 induced ATF4 and CHOP by 2 hrs at concentrations lower than used here (albeit a different cell type). While not exclusively UPR, could the authors address the potential activation of the integrated stress response (eIF2 α phosphorylation, ATF4 and CHOP) in the FRT cells due to ML-240 treatment? If present, is there some link that could this provide an explanation for increased Tg-C1264R secretion? [Basal PERK/UPR activation with mutants.]

Our Response: Thank you for bringing up this important point. As the reviewer acknowledges, the difference in UPR activation could stem from the different cell lines. Additionally, we measured activation via qPCR, whereas Chou et al. measured via immunoblot. We would like to point out that while we did not observe the upregulation of HSPA5 or ASNS (markers of ATF6 and PERK/ISR activation, respectively) in the presence of short ML-240 treatment (2-3 hr), we did observe the upregulation of DNAJB9 (a marker of IRE1/XBP1s activation).

To address Reviewer #3's point, we performed further experiments monitoring the potential activation of the ISR in FRT cells due to ML-240 treatment. We treated C1264R Tg-FT FRT cells with ML-240 (10 μ M) for 2 hours, and monitored eIF2 α phosphorylation via immunoblot. Indeed, we observed that ML-240 induced eIF2 α phosphorylation compared to cells treated with DMSO. Tunicamycin (1mg/mL) was used a positive control, and showed similar results to ML-240. We have integrated these results into the manuscript, available in **Fig EV10C**.

However, we would like to point out that all of these markers represent signs of early UPR inductions. Importantly, our results that HSPA5 transcript levels are not induced suggest that there is only very modest upregulation of ER chaperone levels occurring. Typically, the ER proteostasis network

remodeling requires a longer time than the acute 2-4 hr treatment with ML-240. We have updated the manuscript text as follows:

“Finally, we monitored activation of the unfolded protein response (UPR) in the presence of ML-240 in FRT cells expressing C1264R Tg-FT. Phosphorylation of eIF2 α , an activation marker for the PERK arm of the UPR, was induced within 2 hr of ML-240 treatment (**Fig EV10C**). We further investigated the induction of UPR targets via qRT-PCR. HSPA5 and ASNS transcripts, markers of ATF6 and PERK UPR activation respectively, remained unchanged or slightly decreased after 3 hr treatment with ML-240 in C1264R Tg cells (**Fig EV10D**). Only DNAJB9 transcript expression showed a significant increase in both WT Tg and C2164R Tg FRT cells (**Fig EV10D**). Moreover, ML-240 did not significantly alter cell viability after 3 hr, as measured by propidium iodide staining (**Fig EV10E**). Overall, these results highlight that the short ML-240 treatment induces early UPR markers, but the selective rescue of C1264R Tg secretion via ML-240 treatment is unlikely the results of global remodeling of the ER PN due to UPR activation.”

Reviewer #3, Comment #11: Are the suggested experiments realistic in terms of time and resources? It would help if you could add an estimated cost and time investment for substantial experiments. Any of the suggested experiments above all use reagents reported in the manuscript and so would presumably incur minimal cost and hopefully time. This reviewer is sympathetic to time and financial constraints and so discussion of the issue could suffice.

Our response: We have addressed follow-up experiments whenever possible or provided further discussion details where applicable. We are appreciative of Reviewer #3's sympathy for the time and financial constraints that go into this work and addressing manuscript revisions. Unfortunately, the 1st and 2nd authors both left the lab immediately after the reviews were received. Hence, many of the experiments had to be addressed by other lab members joining the project, which took considerably longer than anticipated. We apologize for the long delay with our revisions.

Reviewer #3, Comment #12: Are the data and the methods presented in such a way that they can be reproduced? Yes. The methodology is explained in detail.

Our Response: Thank you.

Reviewer #3, Comment #13: Are the experiments adequately replicated and statistical analysis adequate? Yes. Relevant information is either in the figure legends or is provided in the source data.

Our Response: Thank you.

Minor comments:

Reviewer #3, Comment #14: Are prior studies referenced appropriately? The references are generally appropriate, with a few exceptions of more general references used

Our Response: Thank you.

Reviewer #3, Comment #15: Are the text and figures clear and accurate? The text is clearly written, and the figures are clear.

Our Response: Thank you.

Reviewer #3, Comment #16: Do you have suggestions that would help the authors improve the presentation of their data and conclusions? A summary figure comparing the changing profiles of WT and C1264R and the factors implicated for them could be helpful.

Our Response: We opted not to include a summary figure because the paper and figures area already dense in information.

Reviewer #3, Comment #17: Perhaps include common nomenclature for proteins as well (e.g. HSP5A - BiP, HSP90B1 - Grp94, etc..)

Our Response: We updated the manuscript throughout to reference common nomenclature or other protein names where applicable at their first mention.

Reviewer #3, Comment #18: Line 317 - our is misspelled

Our Response: Thank you. We have made this correction.

Reviewer #3, Comment #19: Figure 4 - Supplemental Figure 1 - Legend has text referring to panels J and K, but Figure only goes up to F.

Our Response: Thank you. This was an error in references to Figure panel lettering and we have since corrected this. Please note that this Figure is now **Fig EV6**.

Reviewer #3 (Significance (Required)):

Reviewer #3, Comment #20:

- Describe the nature and significance of the advance (e.g. conceptual, technical, clinical) for the field.
- Place the work in the context of the existing literature (provide references, where appropriate).
Protein-protein interactions are often used to illustrate complexes and functionality, but these provide only snapshots, rather than "movies". There are many datasets out there exploring P-P interactions, but most if not all lack any temporal resolution for the interactions they report. The TRIP method described approaches this from the dynamic perspective - identifying the transient interactions formed by folding nascent chains with proteins that aid in their maturation and trafficking, or degradation. This represents an important technical advance in our ability to dynamically monitor protein interactions. The use of Tg mutants is valuable and perhaps this will lead to new perspectives on how to rescue it or other pathophysiological mutants with loss of function phenotypes.

- State what audience might be interested in and influenced by the reported findings.

This work should appeal to a broad audience within cell biology, particularly as the TRIP technique is attempting to address a fundamental question - what interactions form during the biogenesis/lifetime of a protein. Moreover, the effort to try to understand the different interactions formed with pathologically relevant mutant proteins as a strategy to try to rescue functionality, is a valuable exercise of this approach.

- Define your field of expertise with a few keywords to help the authors contextualize your point of view. Indicate if there are any parts of the paper that you do not have sufficient expertise to evaluate.

ER quality control

Our Response: We thank reviewer #3 for this positive endorsement.

Reviewer #4 (Evidence, reproducibility and clarity (Required)):

Summary

In this manuscript, Wright et al. developed an approach (termed TRIP) that allowed to map the temporal changes in the interaction landscape of a newly synthesized protein of interest. Using their TRIP approach, the authors found that the extensive interactions of thyroglobulin (Tg) with the proteostasis network (PN) during its passage through the secretory pathway were profoundly altered in response to disease-causing mutations (e.g. C1264R). The authors cross-validated their findings with a focus RNAi screen monitoring the cellular and secreted abundance of Tg variants upon deletion of PN components. In subsequent experiments the authors focused on two hits, VCP and TEX264, for which they confirmed their inhibitory effect on the secretion of Tg C1264R. Importantly, the authors found that TEX264 increasingly interacts with the Tg mutant and that pharmacological inhibition of VCP yielded the same phenotype than depletion of VCP. Overall, Wright and colleagues **established an elegant method to map protein interaction in a time-resolved manner and demonstrated its value by the analysis of disease-related Tg mutants**. Hence, this work has the potential to serve as a rich resource for Tg-related research and as a powerful new tool to examine protein interactions. However, several concerns remain.

Our response: Thank you to reviewer #4 for their valuable feedback and positive assessment. We addressed all comments in detail below.

Major points:

Reviewer #4, Comment #1: Overall, the TRIP workflow is quite difficult to understand at a first glance - even for a reader with a background in proteomics, biochemistry and cell biology. The authors may want to improve the description of the TRIP methodology and explain in more detail what the individual components and steps are good for. Along the same line, from the main text and the figure legend it was not clear that Tg was actually Flag-tagged. However, without this information it is difficult to follow the workflow. While Figure 1A is certainly helpful, the bulky graphics are deflecting the reader's attention. A more schematic version might be more informative.

Our Response: Thank you for this feedback, which was also mirrored by **Reviewer #2 (comment 10)**. We have made significant updates to clarify **Fig 1** to provide more detail and eliminate some of unnecessary bulky graphics. We also expanded the schematic for the TRIP workflow in Fig 2A and we aligned all symbols used. Furthermore, we have edited the text to describe the workflow more clearly:

“To develop the time-resolved interactome profiling method, we envisioned a two-stage enrichment strategy utilizing epitope-tagged immunoprecipitation coupled with pulsed biorthogonal unnatural amino acid labeling and functionalization (**Fig 1A**). Cells can be pulse labeled with homopropargylglycine (Hpg) to synchronize newly synthesized populations of protein. After pulsed labeling with Hpg, samples can then be collected

across time points throughout a chase period (**Fig 1A**, Box 1) (Kiick *et al*, 2001; Beatty *et al*, 2006). The Hpg alkyne incorporated into the newly synthesized population of protein can be conjugated to biotin using copper-catalyzed alkyne-azide cycloaddition (CuAAC) (**Fig 1A**, Box 2). Subsequently, the first stage of the enrichment strategy can take place where the client protein of interest is globally captured and enriched using epitope-tagged immunoprecipitation, followed by elution (**Fig 1A**, Box 3). The second enrichment step can then utilize a biotin-streptavidin pulldown to capture the Hpg pulse-labeled, and CuAAC conjugated population, enriching samples into time-resolved fractions (**Fig 1A**, Box 4) (Li *et al*, 2020; Thompson *et al*, 2019)."

Additionally, we have improved text to very clearly state that for the TRIP experiments Tg is FLAG-tagged and this epitope tag is required for the two-stage enrichment strategy. As one small example:

"Thyroglobulin was chosen as the model secretory client protein. We generated isogenic Fischer rat thyroid cells (FRT) cells that stably expressed FLAG-tagged Tg (Tg-FT), including WT or mutant variants (A2234D and C1264R) (**Fig EV1**)"

"Furthermore, the C-terminal FLAG-tag and Hpg labeling are necessary for this two-stage enrichment strategy, and DSP crosslinking is necessary to capture these interactions after stringent wash steps (**Fig 1D**, **Fig EV2**)."

Reviewer #4, Comment #2: To what extent do the difference in protein abundance between Tg WT and Tg C1264R contribute to the increase binding of their interactors (e.g., HSP5 and PDIA4). The authors should perform a TRIP coupled immunoblot analysis where WT and Mutant are loaded side-by-side on the SDS-PAGE.

Our Response: As **Reviewer #3 (comment 5)** had a similar inquiry, we provide the same response as listed above:

We do not foresee an impact from level of Tg expression on the profiles obtained by TRIP. We were able to identify distinct profiles because we processed the data and normalized it based on the relative Tg amount. For example, while WT and A2234D Tg are expressed at similar levels intracellularly, we were able to identify distinct differences in the interaction profiles across the two constructs. When developing FRT clones, we selected those that were expressed at similar levels and, therefore, did not have the capability to directly test differences, if any, in observed profiles that may be the result of different expression levels of the same Tg construct. Furthermore, Tg can make up 50% of all protein content within thyroid tissue (Di Jeso & Arvan, 2016). As such, thyroid cells are adept at maintaining the balance of QC machinery to process thyroid. Therefore, we do not anticipate that the amount of Tg expressed in TRIP experiments would have a significant impact on the profiles that we were able to observe.

Reviewer #4, Comment #3: While the RNAi screen was done with pooled siRNA, it is not clear what was used for the RNAi validation experiments shown in Figure 5. This should be done by individual siRNA and not the same pooled reagents as used for the screen.

Our Response: Similarly, pooled siRNAs were initially utilized for the data shown in **Figure 5**. The RNAi screen utilized siRNAs optimized for human cells, where as those found for **Figure 5** were for rat cells. For the revisions, we performed control experiments with individual siRNAs, which are now shown in **Fig EV7J,K**. While we did not find that any one single siRNA recapitulated the full phenotype, we did find that several single siRNAs for VCP and TEX264 at least partially restored the observed phenotype of increased C1264R Tg secretion. This result is expected given that we reasoned the

siRNAs are likely providing an additive effect contributing to the observed phenotypes. We provided these single siRNA control experiments in **Fig EV7J,K**, and updated the manuscript text as follows:

“Several individual VCP and TEX264 siRNAs were able to partially recapitulate these increased secretion phenotype on C1264R Tg-FT, confirming that the effect is mediated by the respective gene silencing (**Fig EV7J,K**).”

Reviewer #4, Comment #4: In Figure 5A it is not clear which band was used to quantify the effect of NAPA reduction. Also, this analysis lacks normalization to an unrelated protein or loading control. Moreover, the authors should also examine the effect of the siRNA targets shown in Figure 5C for Tg WT and not only the mutant.

Our Response: The uppermost band in **Fig 5A** was used for quantification. We added a red asterisk similar to that found in **Fig 5C** to denote this lower back in the lysate panel(s) as a non-specific background band found within the Western blot. These data are the result of immunoprecipitations of both cell lysate and medium content, as such there is no applicable loading control that can be used within the western blots. For experiments, cell amounts were normalized by seeding and subsequently culturing the same amount of cells, as denoted within the Materials and Methods – FRT siRNA validation studies section of the manuscript. Furthermore, there are no loading controls that are easily utilized for analyzing cell culture medium. We have further clarified the **Fig 5** caption to provide clearer experimental detail:

“(A and B) Western blot analysis (A) and quantification (B) of WT Tg-FT secretion from FRT cells transfected with select siRNAs hits from initial screening data set. Red asterisk denotes a non-specific background band within the western blot. Cells were transfected with 25nM siRNAs for 36 hrs, media exchanged and conditions for 4 hrs, Tg-FT was immunoprecipitated from lysate and media samples, and Tg-FT amounts were analyzed via immunoblotting. N = 6.

(C and D) Western blot analysis (C) and quantification (D) of C1264R Tg-FT secretion from FRT cells transfected with select siRNA hits from the initial screening data set. Red asterisk denotes a non-specific background band within the western blot. Cells were transfected with 25nM siRNAs for 36 hrs, media exchanged and conditions for 8 hrs, Tg-FT was immunoprecipitated from lysate and media samples, and Tg-FT amounts were analyzed via immunoblotting. All statistical testing performed using an unpaired student's t-test with Welch's correction. *p<0.05, **p<0.005, ***p<0.0005. N = 5 (one RTN3 sample excluded due to sample handling error).”

Finally, as the siRNA targets shown in **Fig 5C** were shown to be hits exclusively for C1264R Tg-FT we did not believe it was necessary to follow-up on these with WT Tg-FT. Similarly, we did not follow-up on hits that were exclusive to WT Tg-FT with C1264R and A2234D Tg-FT.

Reviewer #4, Comment #5: The authors should also test for the binding of RTN3 to Tg WT and mutant - in particular in comparison to TEX264. This would be important in the context that only RTN3 but not TEX264 was detected in the TRIP approach. Do the authors also detect VCP and LC3B in their pulldowns?

Our response: Please also see **Reviewer #3, comment 9**, who made a similar point. We performed follow-up experiments to test interactions with Tg and the wider panel of ER-phagy receptors. We transiently expressed FLAG-tagged CCPG1, RTN3L, and TEX264 in HEK293 cells stably expressing Tg-NLuc and performed FLAG IPs followed by western blot analysis. We found that WT and C1264R Tg were enriched, albeit modestly, in the RTN3L Co-IP compared to control samples expressing GFP. Additionally, we found that WT, A2234D, and C1264R Tg were all enriched with

CCPG1 compared to control samples expressing GFP. CCPG1 was found to be a C1264R Tg interactor within our mass spectrometry datasets, along with RTN3. We have now integrated these data into the manuscript as **Fig EV8**, and updated the manuscript text as follows:

“Additionally, we monitored Tg enrichment with ER-phagy receptors CCPG1 and RTN3 via Western blot as both were found to be C1264R Tg interactors within our TRIP dataset. RTN3L is found to be the only RTN3 isoform involved in ER turnover via ER-phagy (Grumati et al, 2017). WT and C1264R Tg-NLuc were modestly enriched with RTN3L compared to control samples expressing GFP. Conversely, we found that all Tg variants exhibited modest interactions with CCPG1 compared to control samples expressing GFP, although less than with TEX264 (**Fig EV8**).

Together, these data suggest that TEX264, CCPG1, or RTN3L engage with Tg during processing, and CH-associated Tg mutants may be selectively targeted to TEX264. Furthermore, ER-phagy may be considered as a degradative pathway in Tg processing, as other studies have mainly focused on Tg degradation through ERAD (Tokunaga *et al*, 2000; Menon *et al*, 2007).”

Regarding VCP, we can detect it routinely in our AP-MS experiment as presented previously (Wright et al. 2021), and here in **Fig 3, Appendix Fig S1**. However, we have not been able to detect interactions via western blot, which may be attributed to the increased sensitivity that LC-MS offers. We have not probed for LC3 interactions via western blot as we did not detect it by LC-MS either, but we identified several lysosomal and other autophagy-related components previously (Wright et al. 2021), and here shown in **Appendix Fig S1** and **Fig EV5C**.

Reviewer #4, Comment #6: The effect of TEX264 depletion on Tg secretion should be confirmed by TEX263 KO experiments. Do the authors observe a similar increase in secreted Tg C1264R in BafA1- or SAR405-treated cells? Moreover, the authors should show that Tg C1264R is actually delivered to lysosomes using biochemical assays such as LysolP or colocalization experiments.

Our response: To address this concern, we generated stable TEX264 knockout FRT cell lines by CRISPR, and probed several clones for their impact on Tg secretion. We included this data below in **Review Fig 1**. As can be seen, TEX264 knockout did not recapitulate the increase in C1264R Tg secretion observed with transient siRNA knockout. While disappointing, these results are not necessarily surprising, considering that prolonged TEX264 knockout may lead the cell to adapt compensation mechanisms by modulating other proteostasis factors and/or autophagy machinery.

Review Fig 1: CRISPR Knockout of TEX264 in C1264R-Tg FRT Cell Line.

(A) Western blot of cells with stable expression of C1264R-Tg showing three cell lines with TEX264 knockout (KO). Cellular levels represented in lysate samples and secreted Tg shown in media samples. (B) Quantification of western blot data, represented as mean \pm SEM, $n = 6$. Media and lysate levels of C1264R-Tg are comparable between no knockout and knockout cell lines.

We performed experiments utilizing the autophagy inhibitor Bafilomycin A1, and have now included these results with the manuscript available in **Fig EV10A,B**. We found that BafA1 treatment led to the accumulation of WT Tg in the lysate but not for the C1264R Tg. We updated the manuscript text to accompany these data as follows:

“To understand whether this rescue in secretion was uniquely linked to VCP inhibition or could be more broadly attributed to blocking Tg degradation, we tested the proteasomal inhibitor bortezomib, and lysosomal inhibitor bafilomycin. Bafilomycin increased WT Tg lysate abundance, and bortezomib significantly increased A2234D lysate abundance, consistent with a role of these degradation processes in Tg PQC (**Fig EV10A**). When monitoring Tg-NLuc media abundance, neither bafilomycin nor bortezomib significantly altered WT, A2234D, or C1264R abundance (**Fig. EV10B**), confirming that general inhibition of proteasomal or lysosomal degradation does with rescue mutant Tg secretion.”

These results raise the possibility that the mutant Tg interaction with TEX264 may not lead to active autophagic degradation of mutant Tg. This is also consistent with the slow degradation of C1264R Tg observed in the pulse-chase experiment in **Fig EV9A**. Whether the TEX264 recruitment of mutant Tg leads to degradation or assumes an alternative function, for example, intracellular sequestration, remains to be tested. Importantly, we have refrained from making claims in the manuscript that C1264R Tg is delivered to the lysosome but have presented data showing that it interacts with ER-phagy-related components and have further speculated on the possibility how autophagy could play a role in Tg processing.

Thank you for the LysolP suggestion. Ongoing work in the lab is addressing this question and experiments suggested by the reviewer, but this is better reserved for a follow-up manuscript.

Reviewer #4, Comment #7: Figure 7A and 7C lack loading controls. The quantification shown in Figure 7B and 7D should be normalized to this control. Since VCP activity is often coupled to the of the proteasome, the authors should check whether blocking the proteasome yields a similar effect than ML-240.

Our Response: Like Fig 5A discussed above (**Reviewer #4, comment 4**), these data are the result of immunoprecipitations from cell lysate and medium. As a result, there is not applicable loading control that can be used within the western blots. For experiments, cell amounts were normalized by seeding and subsequently culturing the same amount of cells, as denoted within the Materials and Methods – FRT siRNA validation studies section of the manuscript and Material and Methods – VCP pharmacological inhibition studies.

Regarding the effect of proteasome inhibition, we tested whether bortezomib treatment can increase C1264R Tg secretion. We found that bortezomib led to a small but significant increase in A2234D Tg accumulation in the lysate, but did not increase secretion of Tg for WT or any of the mutant variants. This new data is shown in **Fig EV10A,B**. We updated the text as follow:

“To understand whether this rescue in secretion was uniquely linked to VCP inhibition or could be more broadly attributed to blocking Tg degradation, we tested the proteasomal inhibitor bortezomib, and lysosomal inhibitor bafilomycin. Bafilomycin increased WT Tg lysate abundance, and bortezomib significantly increased A2234D lysate abundance, consistent with a role of these degradation processes in Tg PQC (**Fig EV10A**). When monitoring Tg-NLuc media abundance, neither bafilomycin nor bortezomib significantly altered WT, A2234D, or C1264R abundance (**Fig. EV10B**), confirming that general inhibition of proteasomal or lysosomal degradation does with rescue mutant Tg secretion.”

Reviewer #4, Comment #8: With regard to Figure 7 - Figure supplement 1: The authors should monitor the effect of ML-240 on Tg secretion such that WT and C1264R mutants are directly compared (side-by-side on the same immunoblot). Otherwise, it is difficult to claim that ML-240 rescues the secretion of the mutant.

Our response: The reviewer is referring to the ³⁵S pulse-chase experiments now shown in **Fig EV9**. We would like to clarify that these images are not immunoblots but autoradiographs. Even though the samples for WT and C1264R Tg were loaded onto separate gels, the gels were imaged at the same time and are therefore directly comparable. Regardless, the more meaningful information that can be gleaned from these experiments are the absolute **rates** of protein secretion and degradation and how they change in response to ML-240 treatment. The scale in the quantifications (0 – 100%) is the same and corresponds to the total amount of WT or C1264R Tg that is labeled with ³⁵S during the 30 min pulse. Importantly, we found that C1264R Tg-FT secretion is significantly increased in the presence of ML-240, changing from < 1% to about 10% over the course of 4 hours (**Fig EV9B, Fig 7E**). WT Tg secretion (in the absence of ML-240) reaches about 70 % secretion after 4 hours (**Fig EV9D, Fig 7F**). As these percentages represent the total fraction secreted over the same time frame, they are directly comparable. We clarified in the figure legend that the images represent autoradiographs.

Reviewer #4, Comment #9: How did ML-240 affect the ER-phagy components (in particular RTN3) in the TRIP analysis of Tg C1264R (Figure 7G-L)?

Our response: This is a great discussion point raised by reviewer #4. We have updated the manuscript text to discuss in more detail changes in interactions with degradation components, especially with proteasomal degradation machinery (**Fig 7M,N**). The manuscript text now reads as follows:

“The most striking interaction changes occurred with proteasomal degradation components, which remained steady until 1.5 hr, but then abruptly declined with ML-240 treatment at later time points (**Fig 7M,N**). This decline tracks with changes to the glycan processing machinery, highlighting that the coordination between N-glycosylation and diverting Tg away from ERAD may be a key to the rescue mechanism.”

Minor points:

Reviewer #4, Comment #10: The candidate labeling in Figure 3 - Figure supplement 2 and 3 is too small and unreadable. The authors should provide a higher resolution of these figures or increase the font.

Our response: These figures are now in the **Appendix** and we have edited this figure to provide higher resolution.

Reviewer #4 (Significance (Required)):

Please see above

27th Jun 2024

Manuscript Number: MSB-2024-12441-T

Title: Time-Resolved Interactome Profiling Deconvolutes Secretory Protein Quality Control Dynamics

Author: Madison Wright

Bibek Timalsina

Valeria Garcia Lopez

Jake Hermanson

Sarah Garcia

Lars Plate

Dear Dr. Plate,

Thank you for the submission of your revised manuscript to Molecular Systems Biology. First of all, please accept my apologies for the delay in getting back to you, which was due to the late arrival of reviewers' reports. We have now received the enclosed reports from the four reviewers, who agreed to re-assess it. As you will see, the reviewers are all supportive, and I am pleased to inform you that we will be able to accept your manuscript pending the following amendments:

1. Please address the remaining concerns from Reviewer #2 regarding data presentation.
 2. Please provide a .docx formatted version of the manuscript text (including legends for main figures, EV figures and tables, without figures).
 3. Please provide individual production quality figure files as .eps, .tif, .jpg (one file per figure).
 4. Please provide up to five key words.
 5. Data availability: we cannot find PXD035681. Please ensure the datasets are publicly available upon the acceptance of the manuscript.
 6. Remove the Author contributions section.
 7. There are 10 EV figures, 5 could remain EV, but the rest should be included in the Appendix PDF with nomenclature Appendix Figure Sx and figure legends below the corresponding figures; callouts should be updated accordingly. Please add a Table of Content with page numbers on the title page of Appendix.
 8. We updated our journal's competing interests policy in January 2022 and request authors to consider both actual and perceived competing interests. Please review the policy <https://www.embopress.org/competing-interests> and update your competing interests if necessary.
- Please use the heading "Disclosure statement and competing interests" and there should be only one section referring to all authors.
9. Funding: there is a mismatch between the grant numbers in manuscript text (1R35GM133552) and in the submission system (R35GM133552). This needs to be fixed.
 10. At EMBO Press we ask authors to provide source data for the main figures. Our source data coordinator will contact you to discuss which figure panels we would need source data for and will also provide you with helpful tips on how to upload and organize the files.
 11. All Materials and Methods need to be described in the main text using our 'Structured Methods' format, which is required for all research articles. According to this format, the Methods section includes a Reagents and Tools Table (listing key reagents, experimental models, software and relevant equipment and including their sources and relevant identifiers) followed by a Methods and Protocols section describing the methods using a step-by-step protocol format. The aim is to facilitate adoption of the methodologies across labs. More information on how to adhere to this format as well as a downloadable template (.docx) for the Reagents and Tools Table can be found in our author guidelines:
<https://www.embopress.org/page/journal/17444292/authorguide#structuredmethods>.

12. Please provide a "standfirst text" summarizing the study in one or two sentences (approximately 250 characters, including space), three to four "bullet points" highlighting the main findings and a "synopsis image" (550px width and 400-600 px height, PNG format) to highlight the paper on our homepage.

Here are a couple of examples:

<https://www.embopress.org/doi/10.15252/msb.20199356>

<https://www.embopress.org/doi/10.15252/msb.20209475>

<https://www.embopress.org/doi/10.15252/msb.209495>

13. When you resubmit your manuscript, please download our CHECKLIST (<https://www.embopress.org/pb-assets/embo-site/EMBO%20Press%20Author%20Checklist-1642513524327.xlsx>) and include the completed form in your submission.

Please note that the Author Checklist will be published alongside the paper as part of the transparent process (<https://www.embopress.org/page/journal/17444292/authorguide#transparentprocess>).

Click on the link below to submit your revised paper.

Link Not Available

Kind regards,
Jingyi

Jingyi Hou, PhD
Scientific Editor
Molecular Systems Biology

If you do choose to resubmit, please click on the link below to submit the revision online before 27th Jul 2024.

Link Not Available

IMPORTANT: When you send your revision, we will require the following items:

1. the manuscript text in LaTeX, RTF or MS Word format
2. a letter with a detailed description of the changes made in response to the referees. Please specify clearly the exact places in the text (pages and paragraphs) where each change has been made in response to each specific comment given
3. three to four 'bullet points' highlighting the main findings of your study
4. a 'standfirst text' summarizing in two sentences the study (approx. 250 characters)
6. a "thumbnail image" (width=211 x height=157 pixels, jpeg format), which can be used as 'visual title' to highlight your paper on our homepage.
7. Please include an author contributions statement after the Acknowledgements section (see <https://www.nature.com/msb/authors/index.html#Submission>)
8. When assembling figures, please refer to our figure preparation guideline in order to ensure proper formatting and readability in print as well as on screen:
<https://bit.ly/EMBOPressFigurePreparationGuideline>
See also figure legend guidelines: <https://www.embopress.org/page/journal/17444292/authorguide#figureformat>
9. Include a Reagents and Tools Table as part of the Methods section, which can be downloaded from our author guidelines (<https://www.embopress.org/page/journal/17444292/authorguide#structuredmethods>)

*** PLEASE NOTE *** As part of the EMBO Publications transparent editorial process initiative (see our Editorial at <https://www.nature.com/msb/journal/v6/n1/full/msb201072.html>), Molecular Systems Biology will publish online a Review Process File to accompany accepted manuscripts. When preparing your letter of response, please be aware that in the event of acceptance, your cover letter/point-by-point document will be included as part of this File, which will be available to the scientific community. More information about this initiative is available in our Instructions to Authors. If you have any questions about this initiative, please contact the editorial office (msb@embo.org).

Reviewer #1:

The authors have submitted a substantially improved and much more clear manuscript that addresses many concerns, and the originality and significance of the science was already high. I enthusiastically support publication.

Reviewer #2:

In the revised version of the manuscript "Time-Resolved Interactome Profiling Deconvolutes Secretory Protein Quality Control Dynamics" the authors address the majority of the comments well. However, I have a few suggestions regarding the representation of the data:

I am still struggling with the presentation of the data in Figure 3 and Figure 7. There are a lot of different colors in these figures and it is just not intuitive to understand whether the colors are related or not, but I do think it could be addressed more readily:

For the heatmaps shown in Figure 3 and Figure 7 the authors select different colors as shade behind the protein names. As these are dark colors it makes the protein name less readable and I am wondering if the color shading has anything to do with the colors selected in 3F. A simple solution would just be to remove the color shading behind the protein names as it seems to me that the colors have no specific meaning and that it is only used to symbolize different proteins functions, however the individual heatmaps have anyways titles indicating this, so the color shading seems redundant and rather confuses.

I am struggling with the representation in figure 3F. It looks like the temporal profiles of the WT and the C1264R mutant were clustered individually, is this correct? Does cluster 1 have any relation to cluster 1' because the temporal profiles for the wt and mutant look completely different. Also there are too many colors representing clusters for the wt and mutant, which is confusing. Did the authors try to simply generate a heatmap which compares the temporal profile of the wt and the C1264R mutant next to each other and try to cluster these profiles across wt and C1264R mutant together, as shown in this recent paper:

<https://www.nature.com/articles/s41589-024-01588-3>. I am wondering if this would be a more effective way of representing the unbiased comparison of wt and mutant. I just feel that this would be more intuitive to look at.

Reviewer #3:

I am happy that the authors have made sufficient effort to address my comments.

The authors have addressed all of my concerns. They have significantly improved the manuscript and provided more information.

Reviewer #4:

In this revised manuscript, the authors have sufficiently addressed my comments and concerns from the initial submission. This reviewer appreciates the difficulties and extended time to complete the revisions in light staff departures and feels that the work now would not benefit from any additional data or revisions (beyond grammar/spelling). TRIPs is an important technique that should be able to be applied to a variety of systems and represents an important step forward for time-resolved monitoring of protein-protein interactions.

Rev_Com_number: RC-2022-01691
New_manu_number: MSB-2024-12441-T
Corr_author: Plate
Title: Time-Resolved Interactome Profiling Deconvolutes Secretory Protein Quality Control Dynamics

Our Response to Editor and Reviewer Comments

Editor comments:

1. *Please address the remaining concerns from Reviewer #2 regarding data presentation.*

Our response: Please see below for our response to reviewer #2.

2. *Please provide a .docx formatted version of the manuscript text (including legends for main figures, EV figures and tables, without figures).*

Our response: We have uploaded a .docx file without the figures and all figure legends.

3. *Please provide individual production quality figure files as .eps, .tif, .jpg (one file per figure).*

Our response: We uploaded high quality figure files for all main and EV figures.

4. *Please provide up to five key words.*

Our response: We added the following key words: Temporal proteomics; Thyroglobulin, Hypothyroidism; Proteostasis; Bioorthogonal protein labeling

5. *Data availability: we cannot find PXD035681. Please ensure the datasets are publicly available upon the acceptance of the manuscript.*

Our response: We will make the PRIDE data submission available once we have a doi for the paper.

6. *Remove the Author contributions section.*

Our response: This is removed.

7. *There are 10 EV figures, 5 could remain EV, but the rest should be included in the Appendix PDF with nomenclature Appendix Figure Sx and figure legends below the corresponding figures; callouts should be updated accordingly. Please add a Table of Content with page numbers on the title page of Appendix.*

Our response: We reduced the number of EV figures to five and moved the remaining figures to the Appendix. We added a table of content to the title page of the Appendix.

8. *We updated our journal's competing interests policy in January 2022 and request authors to consider both actual and perceived competing interests. Please review the*

policy <https://www.embopress.org/competing-interests> and update your competing interests if necessary.

Please use the heading "Disclosure statement and competing interests" and there should be only one section referring to all authors.

Our response: We condensed and updated the statement to declare that none of the authors have competing interests.

9. Funding: there is a mismatch between the grant numbers in manuscript text (1R35GM133552) and in the submission system (R35GM133552). This needs to be fixed.

Our response: We updated the grant number in the manuscript text to R35GM133552.

10. At EMBO Press we ask authors to provide source data for the main figures. Our source data coordinator will contact you to discuss which figure panels we would need source data for and will also provide you with helpful tips on how to upload and organize the files.

Our response: We uploaded a zip archive with the source data for figures requested from the editor.

11. All Materials and Methods need to be described in the main text using our 'Structured Methods' format, which is required for all research articles. According to this format, the Methods section includes a Reagents and Tools Table (listing key reagents, experimental models, software and relevant equipment and including their sources and relevant identifiers) followed by a Methods and Protocols section describing the methods using a step-by-step protocol format. The aim is to facilitate adoption of the methodologies across labs. More information on how to adhere to this format as well as a downloadable template (.docx) for the Reagents and Tools Table can be found in our author guidelines: <https://www.embopress.org/page/journal/17444292/authorguide#structuredmethods>.

An example of a Method paper with Structured Methods can be found here: <https://www.embopress.org/doi/10.15252/msb.20178071>.

Our response: We already included a resource table separately from the methods and protocols in our submitted version. We have now reformatted the table to conform to the new EMBO Structured Methods format.

12. Please provide a "standfirst text" summarizing the study in one or two sentences (approximately 250 characters, including space), three to four "bullet points" highlighting the main findings and a "synopsis image" (550px width and 400-600 px height, PNG format) to highlight the paper on our homepage.

Our response: We provided the “standfirst text” and bullet points (see below). We also included a synopsis image in the submission.

A new temporal proteomics technique reveals the protein interaction dynamics of folding and secretion pathways. The method clarifies how mutant forms of the pro-hormone thyroglobulin result in congenital hypothyroidism by elucidating the defects in protein quality control.

Main Findings:

- TRIP combines pulsed unnatural amino acid labeling, biotin-streptavidin enrichment, and TMTpro-multiplexed proteomics to study time-resolved protein interactions.
 - Secretion-defective thyroglobulin mutants exhibit prolonged interaction dynamics with chaperone, disulfide exchange, and degradation pathway interactions.
 - Depletion of degradation factors VCP and TEX264 rescues mutant prohormone secretion.
 - VCP inhibition shifts temporal protein interactions to rescue mutant thyroglobulin secretion.
-
- TRIP combines pulsed unnatural amino acid labeling, biotin-streptavidin enrichment, and TMTpro-proteomics to study time-resolved protein interactions.
 - Two secretion-defective thyroglobulin mutants exhibit prolonged chaperone, disulfide exchange, and degradation pathway interactions.
 - Depleting VCP or TEX264 rescues mutant thyroglobulin secretion.
 - VCP inhibition shifts temporal protein interactions to rescue mutant prohormone secretion.

13. When you resubmit your manuscript, please download our CHECKLIST (<https://www.embopress.org/pb-assets/embo-site/EMBO%20Press%20Author%20Checklist-1642513524327.xlsx>) and include the completed form in your submission.

**Please note* that the Author Checklist will be published alongside the paper as part of the transparent process*

(<https://www.embopress.org/page/journal/17444292/authorguide#transparentprocess>).

Our response: We completed the author checklist and uploaded it along with the submission.

Reviewer Comments

Reviewer #1:

The authors have submitted a substantially improved and much more clear manuscript that addresses many concerns, and the originality and significance of the science was already high. I enthusiastically support publication.

Our response: Thank you for this enthusiastic endorsement.

Reviewer #2:

In the revised version of the manuscript "Time-Resolved Interactome Profiling Deconvolutes Secretory Protein Quality Control Dynamics" the authors address the majority of the comments well. However, I have a few suggestions regarding the representation of the data:

Our response: Thank you to reviewer 2 for taking the time to review our manuscript again. We are pleased to hear that most comments were addressed well. We have addressed the remaining feedback as outlined below.

I am still struggling with the presentation of the data in Figure 3 and Figure 7. There are a lot of different colors in these figures and it is just not intuitive to understand whether the colors are related or not, but I do think it could be addressed more readily: For the heatmaps shown in Figure 3 and Figure 7 the authors select different colors as shade behind the protein names. As these are dark colors it makes the protein name less readable and I am wondering if the color shading has anything to do with the colors selected in 3F. A simple solution would just be to remove the color shading behind the protein names as it seems to me that the colors have no specific meaning and that it is only used to symbolize different proteins functions, however the individual heatmaps have anyways titles indicating this, so the color shading seems redundant and rather confuses.

Our response: The reviewer is correct that the different color shadings for the heatmaps were meant for the different pathways. They are consistent with Appendix Fig S4, which shows the full dataset. However, we agree that the shading is unnecessary in Fig. 3, as the heatmaps are grouped by pathways with clear titles. We have removed the color shadings to hopefully make the protein names more discernible.

I am struggling with the representation in figure 3F. It looks like the temporal profiles of the WT and the C1264R mutant were clustered individually, is this correct? Does cluster 1 have any relation to cluster 1' because the temporal profiles for the wt and mutant look completely different. Also there are too many colors representing clusters for the wt and mutant, which is confusing. Did the authors try to simply generate a heatmap which compares the temporal profile of the wt and the C1264R mutant next to each other and try to cluster these profiles across wt and C1264R mutant together, as shown in this

recent paper: <https://www.nature.com/articles/s41589-024-01588-3>. I am wondering if this would be a more effective way of representing the unbiased comparison of wt and mutant. I just feel that this would be more intuitive to look at.

Our response: Correct. In Figure 3F, the temporal profiles for WT and C1264R were clustered individually. The respective clusters, e.g., cluster 1 (WT) and 1' (C1264R), are not necessarily related, but the alluvial plot in the center clearly shows which proteins “move” from one temporal cluster to another. We carried out this clustering separately because we did not want to make any a priori assumptions about how WT and mutant Tg temporal profiles would be related to one another. However, we appreciate the reviewer's feedback about trying to cluster the WT and C1264R Tg temporal profiles together. We now carried out this analysis and included a new heatmap in **EV Fig. 2C** of the WT and C1264R temporal profiles side-by-side. We were gratified to see that this analysis recapitulated many of the same groupings and shifts that we observed in the individual clustering analysis.

Reviewer #3:

I am happy that the authors have made sufficient effort to address my comments. The authors have addressed all of my concerns. They have significantly improved the manuscript and provided more information.

Our response: Thank you for this feedback and for taking the time to review our manuscript again.

Reviewer #4:

In this revised manuscript, the authors have sufficiently addressed my comments and concerns from the initial submission. This reviewer appreciates the difficulties and extended time to complete the revisions in light of staff departures and feels that the work now would not benefit from any additional data or revisions (beyond grammar/spelling). TRIPs is an important technique that should be able to be applied to a variety of systems and represents an important step forward for time-resolved monitoring of protein-protein interactions.

Our response: Thank you for this positive assessment and for reviewing our manuscript again.

22nd Jul 2024

Manuscript number: MSB-2024-12441R

Title: Time-Resolved Interactome Profiling Deconvolutes Secretory Protein Quality Control Dynamics

Dear Lars,

Thank you again for sending us your revised manuscript. We are now satisfied with the modifications made and I am pleased to inform you that your paper has been accepted for publication.

Kind regards,
Jingyi

Jingyi Hou, PhD
Scientific Editor
Molecular Systems Biology

Rev_Com_number: RC-2022-01691
New_manu_number: MSB-2024-12441R
Corr_author: Plate
Title: Time-Resolved Interactome Profiling Deconvolutes Secretory Protein Quality Control Dynamics